# Learning Imperfect Information Extensive-form Games with Last-iterate Convergence under Bandit Feedback

## Abstract

We study learning the approximate Nash equilibrium (NE) policy profile in two-player zero-sum imperfect information extensive-form games (IIEFGs) with last-iterate convergence. The algorithms in previous works studying this problem either require full-information feedback or only have asymptotic convergence rates. In contrast, we study IIEFGs in the formulation of partially observable Markov games (POMGs) with the perfect-recall assumption and bandit feedback, where the knowledge of the game is not known a priori and only the rewards of the experienced information set and action pairs are revealed to the learners in each episode. Our algorithm utilizes a negentropy regularizer weighted by a virtual transition over information set-action space. By carefully designing the virtual transition together with the leverage of the entropy regularization technique, we prove that our algorithm converges to the NE of IIEFGs with a provable finite-time convergence rate of $\widetilde{\mathcal{O}}(k^{-1/8})$ with high probability under bandit feedback, thus answering the second question of Fiegel et al. (2023) affirmatively.

## 1 Introduction

In imperfect information games (IIGs), players operate with limited visibility into the game's true state, necessitating strategic decision-making based on incomplete information. Notably, the concept of imperfect-information extensive-form games (IIEFGs), as introduced by Kuhn (1953), encapsulates both the intricacies of imperfect information and the sequential nature of players' moves. This framework aptly represents a broad spectrum of real-world scenarios, such as Poker (Heinrich et al., 2015; Moravčík et al., 2017; Brown & Sandholm, 2018), Bridge (Tian et al., 2020), Scotland Yard (Schmid et al., 2021), and Mahjong (Li et al., 2020; Kurita & Hoki, 2021; Fu et al., 2022). Extensive research has been devoted to identifying the (approximate) Nash equilibrium (NE) (Nash Jr, 1950) within IIEFGs. Assuming the condition of perfect recall, where players possess the memory of past events and their implications, various methodologies have been employed to tackle these games. These include linear programming approaches (Koller & Megiddo, 1992; Von Stengel, 1996; Koller et al., 1996), which leverage mathematical optimization under full game knowledge, first-order optimization techniques (Hoda et al., 2010; Kroer et al., 2015; 2018; Munos et al., 2020; Lee et al., 2021; Liu et al., 2022), which iteratively refine strategies via repeated playthroughs of the games, and counterfactual regret minimization algorithms (Zinkevich et al., 2007; Lanctot et al., 2009; Johanson et al., 2012; Tammelin, 2014; Schmid et al., 2019; Burch et al., 2019; Liu et al., 2022), which adaptively adjust strategies based on counterfactual outcomes.

In practical scenarios, IIEFGs might involve large information set and action spaces, thwarting the application of linear programming approaches for *computing* the NE in IIEFGs. In this realm, the NE in IIEFGs is typically *learned* from random samples gathered through iterative playthroughs of the game, by Monte-Carlo counterfactual regret minimization (CFR) methods (Lanctot et al., 2009; Farina et al., 2020; Farina & Sandholm, 2021) or online mirror descent (OMD) and follow-the-regularized-leader (FTRL) frameworks (Farina et al., 2021; Kozuno et al., 2021; Bai et al., 2022; Fiegel et al., 2023). Notably, Bai et al. (2022) devise an OMD-based approach incorporating "balanced exploration policies" to learn an $\varepsilon$-approximate NE with sample complexity of $\widetilde{\mathcal{O}}\left(H^3(XA + YB)/\varepsilon^2\right)$, where $H$ is the horizon length, $X$, $Y$ are the sizes of the information set space for the max- and min-player, and $A$ and $B$ are the sizes of the action space for the max- and

min-player. This upper bound is information-theoretically optimal with respect to all parameters except $H$, up to logarithmic factors. Building upon Bai et al. (2022), Fiegel et al. (2023) make further strides, refining the upper bound to $\widetilde{\mathcal{O}}\left(H(XA + YB)/\varepsilon^2\right)$ by harnessing FTRL with "balanced transitions", achieving (nearly) optimal learning of IIEFGs in all parameters.

Despite the (nearly) optimal leaning of the $\varepsilon$-NE in IIEFGs by Bai et al. (2022); Fiegel et al. (2023), the algorithms in these works require to average all the policies generated during the running of the algorithms, so as to obtain the final policy profile with $\varepsilon$-NE guarantee. This is typically termed as the *average-iterate convergence*. However, in IIEFGs with large information set and action spaces, such an average operation over policy sets usually induces substantial storage and computation overhead. In cases when the policies in the games are approximated by nonlinear function approximation (*e.g.*, neural networks), which has achieved great empirical success in recent years (Moravčík et al., 2017; Brown & Sandholm, 2018), computing the averaged policy even might be not feasible due to the nonlinearity of such function approximations. This motivates the studies of the learning algorithms with the *last-iterate convergence* guarantee of games including IIEFGs (Lin et al., 2020; Wei et al., 2021a;a; Lee et al., 2021; Cai et al., 2022; Abe et al., 2023; Feng et al., 2023; Cen et al., 2023; Liu et al., 2023). Specifically, Lee et al. (2021); Liu et al. (2023) establish algorithms for learning IIEFGs with last-iterate convergence rate of $\widetilde{\mathcal{O}}(1/k)$. However, the algorithms of Lee et al. (2021); Liu et al. (2023) require full-information feedback when learning IIEFGs, and thus can not be directly applied in practical cases when the knowledge of the games is not known a priori. The above considerations naturally motivate the following question:

*Can we achieve last-iterate convergence for learning IIEFGs with bandit feedback?*

Indeed, the same question has also been raised by Fiegel et al. (2023). In this work, we answer this question affirmatively. The main contributions of our work are summarized as follows:

- We propose the first algorithm that learns the approximate NE of IIEFGs with provable last-iterate convergence in the bandit feedback setting. In contrast with the vanilla negentropy regularizer (Lee et al., 2021) and the dilated negentropy regularizer (Lee et al., 2021; Liu et al., 2023) used by previous works to achieve the last-iterate convergence for IIEFGs with full-information feedback, our algorithm leverages the negentropy regularizer weighted by a virtual transition over infoset-action space to regularize the game. Via constructing the loss estimator regularized by such virtual transition weighted negentropy, our algorithm avoids directly regularizing the sequence-form representation of policies and results in a desirable contraction of the KL-divergence between probability measures over the information set-action space, instead of only obtaining the KL-divergence between the sequence-form representation of policies (see Section 4.1 and Section 5.1 for details). Besides, our algorithm does not require any communication or coordination between the two players and is model-free, without requiring the knowledge of the underlying state transition probabilities and the reward functions.

- To efficiently bound the stability term in the one-step analysis of OMD with Bregman divergence induced by the virtual transition weighted negentropy regularizer, we design a virtual transition over the information set-action space that maximizes the minimum visitation probability of all the information sets (see Section 4.1 for more elaboration on this). With such a virtual transition, we finally prove that our algorithm obtains the finite-time last-iterate convergence rate for learning IIEFGs in the bandit feedback setting of $\widetilde{\mathcal{O}}((X + Y)[(XA + YB)^{1/2} + (X + Y)^{1/4}H]k^{-1/8})$ with high probability (for large enough $k$), where $H$ is the horizon length, $X$ and $Y$ are the size of the information set spaces of the max- and min-player, $A$ and $B$ are the size of the action spaces of the max- and min-player, and $k$ is the number of episodes. The methodology of our algorithm's analysis is inspired by the last-iterate convergence learning of the matrix games and the (fully observable) Markov games of Cai et al. (2023), but we provide a refined analysis specifically for IIEFGs to further sharpen the dependence on the parameters when deriving the final convergence rate (see Section 5.1 for details).

- When only obtaining the expected convergence rate is desired, our algorithm can generate a policy profile converging to the NE with a rate of $\widetilde{\mathcal{O}}((X + Y)[(X^2A + Y^2B)^{1/2} + (X + Y)^{1/4}H]k^{-1/6})$ in expectation. For the problem of learning the NE of IIEFGs in the bandit-feedback setting, we provide an $\Omega(\sqrt{XA + YB}k^{-1/2})$ lower bound of the last-iterate convergence rate.

## 2 RELATED WORKS

### 2.1 PARTIALLY OBSERVABLE MARKOV GAMES (POMGS)

With perfect information, learning Markov games (MGs) can be traced back to the seminal work of Littman & Szepesvári (1996) and has since garnered extensive research attention (Littman, 2001; Greenwald & Hall, 2003; Hu & Wellman, 2003; Hansen et al., 2013; Sidford et al., 2018; Lagoudakis & Parr, 2002; Pérolat et al., 2015; Fan et al., 2020; Jia et al., 2019; Cui & Yang, 2021; Zhang et al., 2021; Bai & Jin, 2020; Liu et al., 2021; Zhou et al., 2021; Song et al., 2022; Li et al., 2022; Xiong et al., 2022; Wang et al., 2023; Cui et al., 2023). In scenarios where only imperfect information is available yet the complete knowledge of the game (transitions and rewards) is known, existing research can be categorized into three primary streams. The first stream leverages sequence-form representation of policies to recast the problem as a linear program (Koller & Megiddo, 1992; Von Stengel, 1996; Koller et al., 1996). The second stream translates the problem into a minimax optimization problem and explores first-order algorithms, as exemplified in (Hoda et al., 2010; Kroer et al., 2015; 2018; Munos et al., 2020; Lee et al., 2021; Liu et al., 2022). Lastly, the third stream addresses the problem through CFR, minimizing counterfactual regrets locally within each information set (Zinkevich et al., 2007; Lanctot et al., 2009; Johanson et al., 2012; Tammelin, 2014; Schmid et al., 2019; Burch et al., 2019; Liu et al., 2022).

In the realm where the knowledge of the game is either unknown or only partially accessible, the Monte-Carlo CFR algorithm introduced by Lanctot et al. (2009) pioneers the achievement of the first $\varepsilon$-NE result. This framework has been further generalized and extended by Farina et al. (2020); Farina & Sandholm (2021). Additionally, another line of research focuses on integrating OMD and FTRL frameworks with importance-weighted loss estimators (Farina et al., 2021; Kozuno et al., 2021; Bai et al., 2022; Fiegel et al., 2023). Remarkably, Bai et al. (2022) achieve an $\varepsilon$-approximate NE with sample complexity of $\widetilde{\mathcal{O}}\left(H^3(XA + YB)/\varepsilon^2\right)$ by employing a "balanced" dilated KL-divergence as the distance metric. Building upon this concept, Fiegel et al. (2023) utilize "balanced transitions" and attain a (nearly) optimal sample complexity of $\widetilde{\mathcal{O}}\left(H(XA + YB)/\varepsilon^2\right)$, which matches the information-theoretic lower bound up to logarithmic factors. However, we note that all the algorithms in existing works studying POMGs with bandit feedback only have *average-iterate convergence* guarantees, while we aim to establish the *last-iterate convergence* guarantee.

### 2.2 LAST-ITERATE CONVERGENCE LEARNING IN GAMES

With full-information feedback, learning in games with last-iterate convergence guarantee has been investigated in strongly monotone games (Mokhtari et al., 2020; Jordan et al., 2024), monotone games (Golowich et al., 2020; Cai et al., 2022; Gorbunov et al., 2022; Cai & Zheng, 2023), Markov games (Cen et al., 2021; 2023), and IIEFGs (Lee et al., 2021; Liu et al., 2023; Bernasconi et al., 2024).

Recently, motivated by the fact that it might be restrictive to require full knowledge of the (noisy) gradient as in the full-information feedback setting, a growing body of works has studied learning in games with last-iterate convergence guarantee in the bandit feedback setting including strongly monotone games (Bravo et al., 2018; Lin et al., 2021) (Bravo et al., 2018; Hsieh et al., 2019; Lin et al., 2021; Drusvyatskiy et al., 2022; Huang & Hu, 2023), matrix games (Cai et al., 2023) and Markov games (Wei et al., 2021b; Chen et al., 2022; Cai et al., 2023). However, the algorithm of Wei et al. (2021b) needs coordinated updates and some prior knowledge of the game, and the algorithm of Chen et al. (2022) requires the players to inform the opponent about the entropy of their own policies. Amongst these works, Cai et al. (2023) remove all the coupling requirements, achieving last-iterate convergences of $\widetilde{\mathcal{O}}(k^{-1/8})$ for matrix games and of $\widetilde{\mathcal{O}}(k^{-1/9+\varepsilon})$ for any $\varepsilon > 0$ for irreducible Markov games. We note that all existing works study fully-observable Markov games, while we aim to establish uncoupled algorithms for learning IIEFGs in the formulation of partially-observable Makov games, without requiring the knowledge of the games.

## 3 PRELIMINARIES

For ease of exposition, we consider IIEFGs in the formulation of POMGs and introduce the preliminaries of them in this section, following previous works (Kozuno et al., 2021; Bai et al., 2022).

**Partially Observable Markov Games**    We study episodic, finite-horizon, two-player zero-sum POMGs, denoted by $\mathrm{POMG}(\mathcal{S}, \mathcal{X}, \mathcal{Y}, \mathcal{A}, \mathcal{B}, H, \mathbb{P}, r)$, in which

- $H$ is the horizon length;

- $\mathcal{S} = \bigcup_{h \in [H]} \mathcal{S}_h$ is the finite state space with $\mathcal{S}_h$ as the state space at step $h$. $S = \sum_{h=1}^{H} S_h$ is the size of $\mathcal{S}$ where $|\mathcal{S}_h| = S_h, \forall h \in [H]$;

- $\mathcal{X} = \bigcup_{h \in [H]} \mathcal{X}_h$ is the finite space of information sets (short for *infosets* in the following) for the max-player, where $\mathcal{X}_h = \{x(s) : s \in \mathcal{S}_h\}$ is the set of the infosets at step $h$ with $x : \mathcal{S} \to \mathcal{X}$ as the emission function. $X = \sum_{h=1}^{H} X_h$ is the size of $\mathcal{X}$ with $|\mathcal{X}_h| = X_h$. The finite space of infosets $\mathcal{Y} = \bigcup_{h \in [H]} \mathcal{Y}_h$ for the min-player and its size are defined analogously;

- $\mathcal{A}$ with $|\mathcal{A}| = A$ and $\mathcal{B}$ with $|\mathcal{B}| = B$ are the finite action spaces for the max-player and min-player, respectively;

- $\mathbb{P} = \{p_0(\cdot) \in \Delta_{\mathcal{S}_1}\} \bigcup \{p_h(\cdot|s_h, a_h, b_h) \in \Delta_{\mathcal{S}_{h+1}}\}_{(s_h, a_h, b_h) \in \mathcal{S}_h \times \mathcal{A} \times \mathcal{B}, h \in [H-1]}$ are the state transition probabilities, where $p_0(\cdot)$ is the probability distribution of initial states, $p_h(s_{h+1}|s_h, a_h, b_h)$ is the probability of transitioning to the next state $s_{h+1}$ conditioned on $(s_h, a_h, b_h)$ at step $h$, and $\Delta_{\mathcal{S}_h}$ denotes the probability simplex over $\mathcal{S}_h$;

- $r = \{r_h(s_h, a_h, b_h) \in [0, 1]\}_{(s_h, a_h, b_h) \in \mathcal{S}_h \times \mathcal{A} \times \mathcal{B}, h \in [H]}$ are the (randomized) reward functions with $\bar{r}_h(s_h, a_h, b_h)$ as mean for each $r_h(s_h, a_h, b_h)$.

**Learning Protocol**    Define the max-player's stochastic policy as $\mu = \{\mu_h\}_{h \in [H]}$, where $\mu_h^k : \mathcal{X}_h \to \Delta_{\mathcal{A}}$ denotes the policy at step $h$ during episode $k$. The set of all such policies for the max-player is denoted by $\Pi_{\max}$. Analogously, the min-player's stochastic policy is specified as $\nu = \{\nu_h\}_{h \in [H]}$, with $\nu_h^k : \mathcal{Y}_h \to \Delta_{\mathcal{B}}$ being the policy at step $h$ during episode $k$, and the set of all min-player policies is denoted by $\Pi_{\min}$. The game proceeds in a finite number of episodes. At the commencement of episode $k$, the max-player selects a stochastic policy $\mu^k \in \Pi_{\max}$, while the min-player chooses $\nu^k \in \Pi_{\min}$. Meanwhile, an initial state $s_1^k$ is sampled from the distribution $p_0(\cdot)$ by the environment. During each step $h$ within an episode, the max-player and min-player observe their respective infosets $x_h^k := x(s_h^k)$ and $y_h^k := y(s_h^k)$, but they do not directly observe the underlying state $s_h^k$. Given $x_h^k$, the max-player samples and executes an action $a_h^k \sim \mu_h^k(\cdot|x_h^k)$, while the min-player concurrently takes an action $b_h^k \sim \nu_h^k(\cdot|y_h^k)$. Upon taking these actions, the max-player and min-player receive rewards $r_h^k := r_h(s_h^k, a_h^k, b_h^k)$ and $-r_h^k$, respectively. Subsequently, the game transitions to the next state $s_{h+1}^t \sim p_h(\cdot|s_h^k, a_h^k, b_h^k)$. The $k$-th episode will terminate after actions $a_H^k$ and $b_H^k$ are taken conditioned on $x_H^k$ and $y_H^k$.

**Perfect Recall and Tree Structure**    Following prior works (Kozuno et al., 2021; Bai et al., 2022; Fiegel et al., 2023), we assume that the POMGs adhere to the *tree structure* and the *perfect recall* condition, as defined by Kuhn (1953). Explicitly, the tree structure signifies that for any step $h = 2, \ldots, H$ and state $s_h \in \mathcal{S}_h$, there exists a *unique* path $(s_1, a_1, b_1, \ldots, s_{h-1}, a_{h-1}, b_{h-1})$ culminating in $s_h$. The perfect recall condition, meanwhile, is fulfilled for both players, implying that for any $h = 2, \ldots, H$ and any infoset $x_h \in \mathcal{X}_h$ of the max-player (analogously for the min-player), there exists a *unique* history $(x_1, a_1, \ldots, x_{h-1}, a_{h-1})$ leading to $x_h$. Furthermore, we introduce the notation $C_{h'}(x_h, a_h) \subset \mathcal{X}_{h'}$ to represent the set of descendants of the infoset-action pair $(x_h, a_h)$ at step $h' \geqslant h$. Also, we define $C_{h'}(x_h) := \bigcup_{a_h \in \mathcal{A}} C_{h'}(x_h, a_h)$ as the union of descendants across all actions at $x_h$, and for convenience, let $C(x_h, a_h) := C_{h+1}(x_h, a_h)$ signify the immediate descendants at the subsequent step.

**Sequence-form Representations**    For any pair of product policies $(\mu, \nu)$, the tree structure and the perfect recall condition facilitate the *sequence-form representation* of the reaching probability for the state-action tuple $(s_h, a_h, b_h)$:

$$\mathbb{P}^{\mu,\nu}(s_h, a_h, b_h) = p_{1:h}(s_h)\mu_{1:h}(x(s_h), a_h)\nu_{1:h}(y(s_h), b_h), \qquad (1)$$

where $p_{1:h}(s_h) = p_0(s_1) \prod_{h'=1}^{h-1} p_{h'}(s_{h'+1}|s_{h'}, a_{h'}, b_{h'})$ denotes the sequence-form transition probability, and $\mu_{1:h}(x_h, a_h) := \prod_{h'=1}^{h} \mu_{h'}(a_{h'}|x_{h'})$ and $\nu_{1:h}(y_h, b_h) := \prod_{h'=1}^{h} \nu_{h'}(b_{h'}|y_{h'})$ represent the sequence-form policies of the max- and min player, respectively. Under the sequence-form representation, we adopt a slight abuse of notation for $\mu$ and $\nu$ by interpreting them as $\mu = \{\mu_{1:h}\}_{h \in [H]}$

and $\nu = \{\nu_{1:h}\}_{h \in [H]}$.[1] Furthermore, it is clear that $\Pi_{\max}$ constitutes a convex compact subspace of $\mathbb{R}^{XA}$ that adheres to the constraints $\mu_{1:h}(x_h, a_h) \geqslant 0$ and $\sum_{a_h \in \mathcal{A}} \mu_{1:h}(x_h, a_h) = \mu_{1:h-1}(x_{h-1}, a_{h-1})$, where $(x_{h-1}, a_{h-1})$ is such that $x_h \in C(x_{h-1}, a_{h-1})$ (with the understanding that $\mu_{1:0}(x_0, a_0) = 1$ as a base case).

**Learning Objective**  In this work, we consider the learning objective of finding an approximate NE of the POMG. Specifically, for any $\varepsilon \geqslant 0$, an $\varepsilon$-approximate NE is a pair of product policy $(\mu, \nu)$ satisfying

$$\mathrm{NEGap}(\mu, \nu) \coloneqq \max_{\mu^\dagger \in \Pi_{\max}} V^{\mu^\dagger, \nu} - \min_{\nu^\dagger \in \Pi_{\min}} V^{\mu, \nu^\dagger} \leqslant \varepsilon \,, \tag{2}$$

where $V^{\mu, \nu} = \mathbb{E}_{\mu, \nu}\left[\sum_{h=1}^H r_h(s_h, a_h, b_h)\right]$ the value function of $(\mu, \nu)$ with the expectation taken over the randomness of the product policy pair $(\mu, \nu)$ and the environment. It is known that using regret to NE conversion, an approximate NE can be obtained by averaging all the policies $\{\mu\}_{k=1}^K$ of the max-player generated by an algorithm with sublinear regret (similarly for the min-player) to obtain the average policy pair $(\bar{\mu}, \bar{\nu})$ (see, *e.g.*, Theorem 1 of Kozuno et al. (2021)). This is the so-called *average-iterate convergence* of learning NE. In this work, we are interested in finding the $\varepsilon$-NE with the (finite-time) *last-iterate convergence* guarantee; that is, the algorithm is required to generate an approximate NE policy profile $(\mu^k, \nu^k)$ such that $\mathrm{NEGap}(\mu^k, \nu^k) \leqslant \varepsilon_k$ in each episode for finite-time $k$.

**Information Available to the Players**  In this work, we consider learning POMGs in the bandit feedback setting, where in each episode $k$, the max-player only observes her experienced trajectory $(x_1^k, a_1^k, r_1^k, \ldots, x_H^k, a_H^k, r_H^k)$ of infosets, actions, and rewards, but not the underlying states or the opponent's infosets and actions. Additionally, the max-player does not have knowledge about the policies adopted by the min-player and also can not receive any information from the min-player and vice versa. Besides, there is no shared randomness between both players; that is, the algorithms of both players need to be fully uncoupled from each other.

**Additional Notations**  We slightly abuse the notation to view $x_h$ as the set $\{s \in \mathcal{S}_h : x(s) = x_h\}$, when writing $s \in x_h$. Given sequence-form representations, for any $\mu \in \Pi_{\max}$ and a sequence of functions $f = (f_h)_{h \in [H]}$ with $f_h : \mathcal{X}_h \times \mathcal{A} \to \mathbb{R}$, we define $\langle \mu, f \rangle \coloneqq \sum_{h \in [H], (x_h, a_h) \in \mathcal{X}_h \times \mathcal{A}} \mu_{1:h}(x_h, a_h) f_h(x_h, a_h)$. We denote by $\mathcal{F}^k$ the $\sigma$-algebra generated by the random variables $\{(s_h^t, a_h^t, b_h^t, r_h^t)\}_{h \in [H], t \in [k]}$. For brevity, we abbreviate the conditional expectation $\mathbb{E}[\cdot \mid \mathcal{F}^k]$ as $\mathbb{E}^k[\cdot]$. Throughout this paper, the notation $\widetilde{\mathcal{O}}(\cdot)$ suppresses all logarithmic factors.

## 4 ALGORITHM

In this section, we introduce the proposed algorithm, detailed in Algorithm 1.

### 4.1 FROM SEQUENCE-FORM REPRESENTATIONS TO PROBABILITY MEASURES OVER INFOSET-ACTION SPACE

With sequence-form representations, we first reformulate the IIEFG into the following bilinear game:

$$f(\mu, \nu) = \mu^\top \boldsymbol{G} \nu \,, \tag{4}$$

where $\boldsymbol{G} \in \mathbb{R}^{XA \times YB}$ is the loss matrix with $\boldsymbol{G}[(x_h, a_h), (y_h, b_h)] = \sum_{s_h \in x_h \cap y_h} p_{1:h}(s_h)(1 - r_h(s_h, a_h, b_h))$. In this manner, the learning objective is equivalent to finding $(\mu, \nu)$ such that $\mathrm{NEGap}(\mu, \nu) = \sup_{\mu^\dagger \in \Pi_{\max}, \nu^\dagger \in \Pi_{\min}} f(\mu, \nu^\dagger) - f(\mu^\dagger, \nu) \leqslant \varepsilon$. At a high level, we apply the entropy regularizing technique to perturb the bilinear form of the game, as defined in Eq. (4), into a strongly convex-strongly concave structure, ensuring convergence to both the NE of the perturbed game (and thus the NE of the original game in Eq. (4)). This approach

---

[1]The sequence-form representation of policies is defined in a top-down manner and is equivalent to the "treeplex" space of policies defined in a bottom-up manner (see, *e.g.*, Lee et al. (2021)).

---

**Algorithm 1** OMD with Virtual Transition Weighted Negentropy Regularization (max-player)

1: **Input:** $\eta_k = k^{-\alpha_\eta}, \gamma_k = k^{-\alpha_\gamma}, \varepsilon_k = k^{-\alpha_\varepsilon}$.
2: **Initialize:** $\mu_1(a_h|x_h) = \frac{1}{A}, \forall(x_h, a_h) \in \mathcal{X}_h \times \mathcal{A}, \forall h \in [H]$. Set $p^x$ computed by Algorithm 2.
3: **for** $k = 1, \cdots,$ **do**
4:    **for** $h = 1, \cdots, H$ **do**
5:       Observes $x_h^k$, executes $a_h^k \sim \mu_h^k(\cdot|x_h^k)$ and receives $r_h^k$.
6:       For all $(x_h, a_h) \in \mathcal{X}_h \times \mathcal{A}$, set entropy regularized loss estimator as

$$\widehat{\ell}_h^k(x_h, a_h) = \frac{\mathbb{I}_h^k\{x_h, a_h\}}{\mu_{1:h}^k(x_h, a_h) + \gamma_k}(1 - r_h^k) + \varepsilon_k \cdot p_{1:h}^x(x_n) \log[p_{1:h}^x \cdot \mu_{1:h}^k](x_h, a_h).$$

7:    **end for**
8:    Update policy

$$\mu^{k+1} = \arg\min_{\mu \in \Pi_{\max}^{k+1}} \eta_k \langle \mu, \widehat{\ell}^k \rangle + D_\psi(\mu, \mu^k), \tag{3}$$

     where $\Pi_{\max}^{k+1} = \{\mu \in \Pi_{\max} : \mu(a_h|x_h) \geqslant \frac{1}{A(k+1)}, \forall(x_h, a_h) \in \mathcal{X}_h \times \mathcal{A}, \forall h \in [H]\}$.
9: **end for**

---

**Algorithm 2** Computing virtual transition $p^x$ (max-player)

1: **Input:** Game tree structure of $\mathcal{X} \times \mathcal{A}$.
2: **Initialization:** Sequence-form representation of virtual transition $q \in \mathbb{R}^X$; array of maximized number of descendant infoset $c \in \mathbb{R}^X, d \in \mathbb{R}^{XA}$. For all $x_H$ in $\mathcal{X}_H$, set $c[x_H] = 1$.
3: **for** $h = H - 1$ to $1$ **do**
4:    **for** $x_h$ in $\mathcal{X}_h$ **do**
5:       **for** $a_h$ in $\mathcal{A}$ **do**
6:          Compute $d[x_h, a_h] = \sum_{x_{h+1} \in C(x_h, a_h)} c[x_{h+1}]$.
7:       **end for**
8:       Compute $c[x_h] = \max_{a \in \mathcal{A}} d[x_h, a]$.
9:    **end for**
10: **end for**
11: **for** $x_1$ in $\mathcal{X}_1$ **do**
12:    Compute $q_{1:1}(x_1) = \frac{c[x_1]}{\sum_{x_1 \in \mathcal{X}_1} c[x_1]}$.
13: **end for**
14: **for** $h = 1$ to $H - 1$ **do**
15:    **for** $x_h, a_h$ in $\mathcal{X}_h \times \mathcal{A}$ **do**
16:       **for** $x_{h+1}$ in $C(x_h, a_h)$ **do**
17:          Compute $q_{1:h+1}(x_{h+1}) = q_{1:h}(x_h) \cdot \frac{c[x_{h+1}]}{\sum_{x_{h+1} \in C(x_h, a_h)} c[x_{h+1}]}$.
18:       **end for**
19:    **end for**
20: **end for**
21: **return** $q$.

---

builds upon previous research that has explored last-iterate convergence learning in Markov games with full-information feedback (Cen et al., 2021; Chen et al., 2022; Cen et al., 2023), matrix games and Markov games with bandit feedback (Cai et al., 2023), and IIEFGs with full-information feedback (Liu et al., 2023). Specifically, we consider the following perturbed game as a surrogate:

$$f_k(\mu, \nu) = \mu^\top G \nu + \varepsilon_k \psi(\mu) - \varepsilon_k \psi(\nu), \tag{5}$$

where $\psi$ is some strongly convex regularizer to be used in OMD and $\varepsilon_k > 0$ serves as the knob to control the strength of the entropy regularization in episode $k$. Intuitively, due to the strongly convex-strongly concave property of the perturbed game, one is able to find the approximate NE of it with last-iterate convergence using OMD. On the other hand, by gradually decreasing $\varepsilon_k$ to be moderately small, the approximate NE of the perturbed game in Eq. (5) will also serve as an approximate NE of the original game in Eq. (4).

The crucial aspect lies in selecting an appropriate regularizer $\psi$. Initially, the first candidate that might come to mind is the utilization of the vanilla negentropy regularizer $\psi(\mu) = \sum_{h,x_h,a_h} \mu_{1:h}(x_h, a_h) \log \mu_{1:h}(x_h, a_h)$, which has been utilized to achieve the last-iterate convergence for IIEFGs with full-information feedback (Lee et al., 2021) and matrix games, the special case of IIEFGs, with bandit feedback (Cai et al., 2023). However, in IIEFGs with bandit feedback, though using the vanilla negentropy regularizer results in a convergence of the Bregman divergence, it is generally hard to control the NE gap since it directly regularizes the sequence-form representation policies. The other natural approach is considering using the dilated negentropy $\psi(\mu) = \sum_{h,x_h,a_h} \mu_{1:h}(x_h, a_h) \log\left(\frac{\mu_{1:h}(x_h,a_h)}{\mu_{1:h}(x_h)}\right)$ (Kroer et al., 2015; Kozuno et al., 2021). Indeed, the dilated negentropy has also been used to achieve the last-iterate convergence of the IIEFGs with full-information feedback (Lee et al., 2021; Liu et al., 2023; Bernasconi et al., 2024). However, in contrast with the full-information feedback setting, leveraging the entropy regularization technique to obtain the finite-time convergence guarantee in the bandit feedback setting requires the probability of selecting each action $a_h$ given each infoset $x_h$ being lower bounded to prevent the stability term in the analysis of OMD from being prohibitively largely. This essentially requires constraining the optimization of OMD onto a subset of the entire space of the sequence-form representations of policies $\Pi_{\max}$. Nevertheless, this will also make the stability term of OMD using the dilated negentropy in conjunction with the regularization technique hard to control, as bounding the stability term of the OMD with dilated negentropy critically relies upon its closed-form update solution (see, *e.g.*, Lemma 7 of Kozuno et al. (2021)), which no longer holds in the case where the policy update of OMD is constrained onto a subset of $\Pi_{\max}$.

To cope with the above difficulties, we instead consider using the negentropy regularizer weighted by a kind of *virtual transition* $p^x$ over the infoset-action space $\mathcal{X} \times \mathcal{A}$:

$$\psi_{p^x}(\mu) = \sum_{h,x_h,a_h} p^x_{1:h}(x_h)\mu_{1:h}(x_h, a_h) \log\left(p^x_{1:h}(x_h)\mu_{1:h}(x_h, a_h)\right),$$

where $p^x_h(\cdot|x_h, a_h) \in \Delta_{C(x_h,a_h)}$ is a transition probability over $\mathcal{X}_h \times \mathcal{A} \times \mathcal{X}_{h+1}$ and $p^x_{1:h}(x_h) = p^x_0(x_1) \prod_{h'=1}^{h-1} p^x_{h'}(x_{h'+1}|x_{h'}, a_{h'})$ is its sequence-form representation. Note that $p^x_h(x_{h+1}|x(s_h), a_h)$ is not necessarily to be the true transition probability $\mathbb{P}^{\mu^k,\nu^k}(x_{h+1}|x(s_h), a_h) = \sum_{s_{h+1}\in x_{h+1}, b_h \in \mathcal{B}} p(s_{h+1}|s_h, a_h, b_h)\nu^k(b_h|y(s_h))$ experienced by the max-player in episode $k$. Also, notice that $\psi_{p^x}(\cdot)$ is dependent on the chosen virtual transition $p^x$ and we drop the dependence in the subscript of $\psi_{p^x}(\cdot)$ on $p^x$ when the context is clear for brevity. We remark that similar ideas leveraging negentropy weighted by the transition over infoset-action space have also been exploited by Bai et al. (2022); Fiegel et al. (2023). However, we would like to underscore that the design of our virtual transition $p^\star$ over infoset-action space is different from those of Bai et al. (2022); Fiegel et al. (2023) and we aim to establish the last-iterate convergence of IIEFGs while they can only guarantee the average-iterate convergence, necessitating different theoretical analysis. Besides, one can see that the constructed virtual transition $p^x$ is well-defined by the perfect recall condition and $p^x_{1:h} \cdot \mu_{1:h}$ with $[p^x_{1:h} \cdot \mu_{1:h}](x_h, a_h) = p^x_{1:h}(x_h)\mu_{1:h}(x_h, a_h)$ is a probability measure over the infoset-action space $\mathcal{X}_h \times \mathcal{A}$ at step $h$. Therefore, we actually regularize the probability measures over $\mathcal{X}_h \times \mathcal{A}$ instead of directly regularizing the sequence-form representation $\mu$, which tackles the difficulties of using the vanilla negentropy and the dilated negentropy as mentioned above. The other nice property of virtual transition weighted negentropy is that $D_\psi(\mu_1, \mu_2) = \mathrm{KL}(p^x\mu_1, p^x\mu_2)$, facilitating bounding the final NE gap as we shall see in Section 5.1.

With regularizer $\psi$ specified, the derivative of $f_k(\mu, \nu)$ w.r.t. $\mu(x_h, a_h)$ is $\frac{\partial f_k(\mu,\nu)}{\partial \mu_{1:h}(x_h,a_h)} = \boldsymbol{G}\nu\left[(x_h, a_h)\right] + \varepsilon_k \cdot p^x_{1:h}(x_n)\left[\log[p^x_{1:h} \cdot \mu_{1:h}](x_h, a_h) + 1\right]$. Since $[p^x_{1:h} \cdot \mu_{1:h}] \in \Delta_{\mathcal{X}_h \times \mathcal{A}}$ for any $\mu$, the constant 1 in the above display does not affect the optimization of OMD. On the other hand, in the bandit feedback setting, an (optimistically biased) loss estimate $\frac{\mathbb{I}\{x_h, a_h\}}{\mu^k_{1:h}(x_h,a_h)+\gamma_k}(1 - r^k_h)$ of $\boldsymbol{G}\nu\left[(x_h, a_h)\right]$ in episode $k$ is constructed (Kozuno et al., 2021), where $\gamma_k > 0$ is the implicit exploration parameter (Neu, 2015). This specifies the final entropy regularized loss estimator used by Algorithm 1 on Line 6.

With the constructed loss estimator, Algorithm 1 then uses OMD to update policy. Since now the entropy regularized loss estimator is considered, the variance of the loss estimator will be prohibitively large if running OMD on the entire space of the sequence-form representations $\Pi_{\max}$, eventually leading to an unbounded stability term of OMD. Hence we constrain the feasible set of the OMD

as a subset $\Pi_{\max}^{k+1}$ of $\Pi_{\max}$, where each $\mu \in \Pi_{\max}^{k+1}$ satisfying $\mu(a_h|x_h)$ is lower bounded for all $(x_h, a_h) \in \mathcal{X}_h \times \mathcal{A}$ and $h \in [H]$ (Line 8).

## 4.2 Virtual Transition with Maximized Minimum Visitation Probability

As elaborated in Section 4.1, our Algorithm 1 leverages a virtual transition weighted negentropy to regularize the loss estimator and induce the Bregman divergence used in OMD. It remains to specify an appropriate virtual transition $p^x$. The upside of employing such virtual transition $p^x$ lies in that it implicitly helps to operate the update of OMD in the space of probability measures over infoset-action pairs instead of the sequence-form representations of policies. However, this also comes at the expense of enlarging the stability term of OMD. Specifically, upon applying the virtual transition to weight the negentropy, the stability term associated with OMD at each information set $x_h$ will be enlarged by (approximately) a multiplicative factor of $1/p^x(x_h)$. This enlargement arises intuitively from the fact that, at each $x_h$, the Bregman divergence induced by $\psi$ undergoes a downscaling, proportional to $p^x(x_h)$, thereby resulting in a relative increase in the stability term. Therefore, to ensure that the stability term is well-controlled, we design the following $p^x$ which maximizes the minimum visitation probability of all $x_h$ in its sequence-form representation:

$$p^x = \arg\max_{q \in \mathbb{P}^x} \min_{x_h \in \mathcal{X}_h, h \in [H]} q_{1:h}(x_h). \tag{6}$$

In the above display, we denote by $\mathbb{P}^x$ the set of all the valid virtual transitions over infoset-action space. We note that such a virtual transition $p^x$ can be efficiently computed by Algorithm 2 via backward dynamic programming.

**Computation** Due to the fact the update of OMD is now constrained onto a subset $\Pi_{\max}^k$ of the entire space $\Pi_{\max}$ of the sequence-form representation policies, the computation of Eq. (3) generally does not have a closed-form solution. We hereby provide an algorithm, which computes an approximate solution to Eq. (3), detailed in Algorithm 3. In particular, Algorithm 3 utilizes a Frank–Wolfe-type procedure to compute the update in Eq. (3). In particular, there will be $T$ iterations in Algorithm 3, and in each iteration $t$, the policy will be updated towards the direction that minimizes the gradient of the objective function w.r.t. policy $\mu^{(t-1)}$ in iteration $t-1$ by dynamic programming in Algorithm 4. We defer the details of Algorithm 3 and Algorithm 4 to Appendix F.

## 5 Analysis

In this section, we first present the upper bound of the last-iterate convergence rate of our Algorithm 1. Then the lower bound for the problem of learning IIEFGs with bandit feedback and last-iterate convergence guarantee will be provided.

### 5.1 Upper Bound of Last-iterate Convergence

**Theorem 5.1.** *If Algorithm 1 is adopted by both players, for any $k \geqslant 1$, with probability at least $1 - \widetilde{\mathcal{O}}(\delta)$, it holds that*

$$\mathrm{NEGap}(\mu^k, \nu^k) = \mathcal{O}\left( \left[ (XA + YB)^{\frac{1}{2}} k^{-\frac{1}{8}} + (XA + YB)^{\frac{1}{2}} H k^{-\frac{3}{8}} + \left(X^2 A + Y^2 B\right)^{\frac{1}{2}} k^{-\frac{1}{4}} + (X + Y)^{\frac{1}{4}} H k^{-\frac{1}{8}} \right] \right.$$

$$\left. \cdot (X + Y) \left( \log\left(XAk/\delta\right) + \log\left(YBk/\delta\right) \right) \log^{\frac{1}{2}}(k) + k^{-\frac{1}{8}} H (\ln(XA) + \ln(YB)) + (XAB + YBH)/k \right).$$

**Remark 5.2.** *Ignoring the poly-logarithmic terms and when $k$ is large enough (specifically, $k \geqslant \max\{H^4, (X^2A+Y^2B)^4/(XA+YB)^4, (XA+YB)^{8/7}/(X+Y)^{10/7}\}$), we have $\mathrm{NEGap}(\mu^k, \nu^k) = \widetilde{\mathcal{O}}((X + Y)[(XA + YB)^{1/2} + (X + Y)^{1/4}H]k^{-1/8})$. Besides, when only obtaining an expected last-iterate convergence rate is desired, our Algorithm 1 has an improved last-iterate convergence rate of $\widetilde{\mathcal{O}}((X+Y)[(X^2A+Y^2B)^{1/2} + (X+Y)^{1/4}H]k^{-1/6})$ in expectation, the details of which are deferred to Appendix C. Though the last-iterate convergence rate of our Algorithm 1 is inferior to the $\widetilde{\mathcal{O}}(1/k)$ convergence rate by Lee et al. (2021); Liu et al. (2023), we note that both their algorithms can only work in the full-information setting. Further, we remark that the algorithm of Lee et al. (2021) needs the assumption that the NE of the IIEFG considered is unique, and the algorithm of Liu et al. (2023) requires both players being controlled by a central controller, and thus the algorithm*

*of Liu et al. (2023) is not uncoupled. In contrast, our algorithm can work in the bandit feedback setting, is fully uncoupled between the two players, and can still guarantee a regret of order $\widetilde{\mathcal{O}}(k^{7/8})$ when the opponent of the max-player is an adversary. More importantly, we show in Section 5.2 that the lower bound of the convergence rate for learning IIEFGs with bandit feedback, last-iterate convergence guarantee, and uncoupled algorithms will be of order $\Omega(k^{-1/2})$ (for large enough k).*

**Proof Sketch of Theorem 5.1** We postpone the complete proof of Theorem 5.1 to Appendix B. Here we provide a proof sketch of it.

We denote by $\xi^{k,\star} := (\mu^{k,\star}, \nu^{k,\star})$ the unique NE in the regularized game $f_k$ in Eq. (5), where there is only a unique NE since $f_k$ is strongly convex in $\mu$ and strongly concave in $\nu$. We first show that in each episode $k$, the product policy $\xi^k := (\mu^k, \nu^k)$ generated by the algorithm will approach $\xi^{k,\star}$ close enough by showing that the Bregman divergence $D_\psi(\xi^{k,\star}, \xi^k)$ is an (approximate) contraction mapping. In particular, we show that

$$D_\psi\left(\xi^{k+1,\star}, \xi^{k+1}\right) \lesssim (1 - \eta_k \varepsilon_k) D_\psi\left(\xi^{k,\star}, \xi^k\right) + \eta_k^2 \left(X \underline{\tau}_k + Y \bar{\tau}_k\right) + \eta_k^2 \left(X^2 A + Y^2 B\right)$$
$$+ \eta_k \rho_k + \eta_k \sigma_k + \eta_k^2 \varepsilon_k^2 H^2 \left(X A + Y B\right) + \omega_k \,, \tag{7}$$

where we denote

$$\underline{\tau}_k = \frac{1}{X} \sum_{h, x_h, a_h} \frac{1}{p_{1:h}^x(x_h)} \left(\frac{\mathbb{I}_h^k\{x_h, a_h\}}{\mu_{1:h}^k(x_h, a_h) + \gamma_k} - 1\right),$$

$$\bar{\tau}_k = \frac{1}{Y} \sum_{h, y_h, b_h} \frac{1}{p_{1:h}^y(y_h)} \left(\frac{\mathbb{I}_h^k\{y_h, b_h\}}{\nu_{1:h}^k(y_h, b_h) + \gamma_k} - 1\right),$$

$$\rho_k = \sum_{h, x_h, a_h} \mu_{1:h}^k(x_h, a_h) \left[(\boldsymbol{G}\nu^k)\left[(x_h, a_h)\right] - \left(\frac{\mathbb{I}_h^k\{x_h, a_h\}}{\mu_{1:h}^k(x_h, a_h) + \gamma_k}\left(1 - r_h^k\right)\right)\right]$$

$$+ \sum_{h, y_h, b_h} \nu_{1:h}^k(y_h, b_h) \left[\left(1 - (\boldsymbol{G}^\top \mu^k)\left[(y_h, b_h)\right]\right) - \frac{\mathbb{I}_h^k\{y_h, b_h\}}{\nu_{1:h}^k(y_h, b_h) + \gamma_k} r_h^k\right],$$

$$\sigma_k = \sum_{h, x_h, a_h} \mu_{1:k}^{k,\star}(x_h, a_h) \left[(\boldsymbol{G}\nu^k)\left[(x_h, a_h)\right] - \left(\frac{\mathbb{I}_h^k\{x_h, a_h\}}{\mu_{1:h}^k(x_h, a_h) + \gamma}\left(1 - r_h^k\right)\right)\right]$$

$$+ \sum_{h, y_h, b_h} \nu_{1:h}^{k,\star}(y_h, b_h) \left[\frac{\mathbb{I}_h^k\{y_h, b_h\}}{\nu_{1:h}^k(y_h, b_h) + \gamma_k} r_h^k - \left(1 - (\boldsymbol{G}^\top \mu^k)\left[(y_h, b_h)\right]\right)\right],$$

$$\omega_k = D_\psi\left(\mu^{k+1,\star}, \mu^{k+1}\right) - D_\psi\left(\mu^{k,\star}, \mu^{k+1}\right) + D_\psi\left(\nu^{k+1,\star}, \nu^{k+1}\right) - D_\psi\left(\nu^{k,\star}, \nu^{k+1}\right).$$

Expanding the above recursion, we can bound $D_\psi\left(\xi^{k+1,\star}, \xi^{k+1}\right)$ as

$$D_\psi\left(\xi^{k+1,\star}, \xi^{k+1}\right) \lesssim \underbrace{\sum_{i=1}^k w_k^i \eta_i \rho_i}_{\textbf{Term 1}} + \underbrace{\sum_{i=1}^k w_k^i \eta_i \sigma_i}_{\textbf{Term 2}} + \underbrace{(XA + YB) H^2 \sum_{i=1}^k w_k^i (\eta_i \varepsilon_i)^2}_{\textbf{Term 3}}$$

$$+ \underbrace{\sum_{i=1}^k w_k^i \eta_i^2 \left(X \underline{\tau}_i + Y \bar{\tau}_i\right)}_{\textbf{Term 4}} + \underbrace{\sum_{i=1}^k w_k^i \eta_i^2 \left(X^2 A + Y^2 B\right)}_{\textbf{Term 5}} + \underbrace{\sum_{i=1}^k w_k^i \omega_i}_{\textbf{Term 6}}, \tag{8}$$

where $w_k^i = \prod_{j=i+1}^k (1 - \eta_j \varepsilon_j)$ is the contraction parameter. Then we bound each of the above terms in by Lemma B.4 - Lemma B.9 in Appendix B.2. Note that we follow a similar analysis scheme of Cai et al. (2023) to bound the last-iterate convergence of learning matrix games with bandit feedback. However, we also remark that the straightforward application of their analysis will not address our problem of learning IIEFGs with bandit feedback, since we leverage a different regularizer and a new virtual transition $p^x$ computed by Algorithm 2, which serves as a core ingredient of the analysis in deriving the contraction of Eq. (7) and bounding **Term 6**. Besides, compared with the analysis of Cai et al. (2023), the additional **Term 5** in Eq. (8) comes from the fact that we

establish a refined analysis in the case of IIEFGs to further sharpen the dependence on $X$ and $A$ (as well as $Y$ and $B$) of the final convergence rate.

Further, one can see that the NE policy profile $\xi^{k,\star}$ of the perturbed game in Eq. (5) is also an approximate NE of the original game in Eq. (4), enabling to bound $\mathrm{NEGap}(\xi^k)$ using $\mathrm{NEGap}(\xi^{k,\star})$ together with the distance between $\xi^k$ and $\xi^{k,\star}$ weighted by the virtual transitions as bellow:

$$\mathrm{NEGap}(\xi^k) \leqslant \mathrm{NEGap}(\xi^{k,\star}) + X \left\| p^x \left( \mu^k - \mu^{k,\star} \right) \right\|_1 + Y \left\| p^y \left( \nu^k - \nu^{k,\star} \right) \right\|_1, \qquad (9)$$

where $\mathrm{NEGap}(\xi^{k,\star})$ can be controlled by Lemma B.2. Due to the constructed virtual transition $p^x$ and $p^y$, the second and the third term in Eq. (9) are actually the $\ell_1$-norm of the difference between the probability measures over infoset-action spaces, which thus turns out to be bounded by $\mathcal{O}(\sqrt{\mathrm{KL}\left(p^x\mu^{k,\star}, p^x\mu^k\right)})$ and $\mathcal{O}(\sqrt{\mathrm{KL}\left(p^y\nu^{k,\star}, p^y\nu^k\right)})$ by Pinsker's inequality. Also, thanks to the virtual transition weighted negentropy $\psi$, one can see that $\mathrm{KL}\left(p^x\mu^{k,\star}, p^x\mu^k\right) = D_\psi(\mu^{k,\star}, \mu^k)$ (and similarly on the min-player side). Therefore, the proof can be concluded by substituting Eq. (8) into Eq. (9) and then using Lemma B.2 and Lemma B.4 - Lemma B.9.

### 5.2 LOWER BOUND OF LAST-ITERATE CONVERGENCE

**Theorem 5.3.** *For any algorithm* Alg *that both players adopt to generate policy profile* $(\mu^k, \nu^k)$ *and is uncoupled between both players, there exists an IIEFG instance such that the lower bound of the last-iterate convergence of learning this IIEFG in the bandit-feedback setting satisfies* $\mathrm{NEGap}(\mu^k, \nu^k) = \Omega(\sqrt{XA + YB}k^{-1/2})$, *when* $k \geqslant \max(XA, YB)$.

*Proof Sketch.* The idea of the proof is to leverage the fact that if an uncoupled algorithm can learn the NE of IIEFGs with a last-iterate convergence guarantee of $\widetilde{\Theta}(k^{-\alpha})$ ($\alpha \in [0, 1]$) in the bandit feedback setting, then it can be used to learn IIEFGs where the opponent is an adversary with a regret of order $\widetilde{\Theta}(k^{1-\alpha})$. Therefore, considering that the hardness of minimizing regret of IIEFGs with an adversarial opponent is equivalent to minimizing regret on a bandit problem with $AX$ arms (Bai et al., 2022; Fiegel et al., 2023), the proof of Theorem 5.3 can be completed by contradiction. $\qquad \square$

**Remark 5.4.** *Compared with the lower bound of the convergence rate above, the upper bound in Theorem 5.1 is loose by a factor of $\widetilde{\mathcal{O}}((X + Y)k^{3/8})$ (for large enough $X$, $Y$, $A$ and $B$). We believe one of the promising approaches to improve the upper bound of the convergence rate might be to consider using the optimistic OMD/FTRL, which utilizes accelerated techniques from the optimization perspective and is typically used to achieve the $\widetilde{\mathcal{O}}(1/k)$ convergence rate for learning IIEFGs with last-iterate convergence in the full-information setting. One of the main difficulties of using optimistic OMD/FTRL in conjunction with the regularization technique to achieve a faster last-iterate convergence rate of learning IIEFGs in the bandit feedback setting is that the loss estimator constructed in the bandit feedback setting (either unbiased or optimistically biased) to serve as a surrogate of the true loss would have undesirably large variance, making the stability of optimistic OMD/FTRL hard to be controlled even in the special case of learning matrix games. We leave the possible improvement of our convergence upper bound as our future study.*

## 6 CONCLUSTION

In this work, we make the first step to establishing the algorithm that learns an approximate NE of IIEFGs in the bandit feedback setting with finite-time last-iterate convergence. Our algorithm is fully uncoupled between the two players involved in the games and does not require any coordination, communication, or shared randomness between these players. We prove that our algorithm achieves the last-iterate convergence of order $\widetilde{\mathcal{O}}((X + Y)[(XA + YB)^{1/2} + (X + Y)^{1/4}H]k^{-1/8})$ with high probability and of order $\widetilde{\mathcal{O}}((X+Y)[(X^2A+Y^2B)^{1/2}+(X+Y)^{1/4}H]k^{-1/6})$ in expectation (for large enough $k$). Also, we provide the lower bound of order $\Omega(\sqrt{XA + YB}k^{-1/2})$ for learning IIEFGs with last-iterate convergence guarantee in the bandit feedback setting. An interesting problem might be closing the gap between the established convergence upper and lower bound, which still remains open in the special case of learning matrix games with the last-iterate convergence guarantee in the bandit feedback setting. We will leave the investigation of this for our future research endeavors.

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
