## A  MORE DISCUSSIONS ON VIRTUAL TRANSITION PROBABILITIES

### A.1  ILLUSTRATION ON THE FAILURE OF USING UNIFORM VIRTUAL TRANSITION

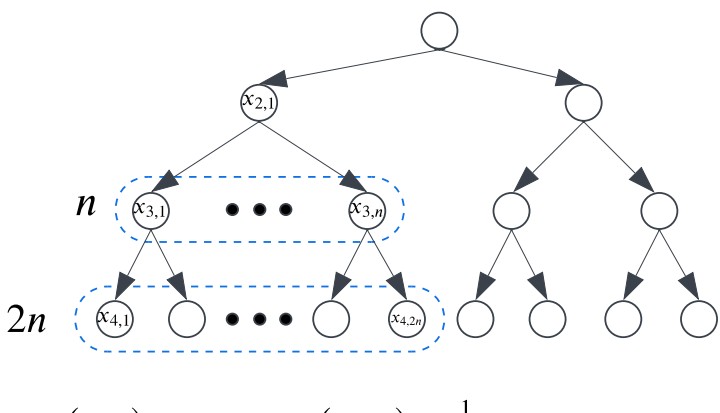

$$p_{1:H}(x_{4,1}) = \ldots = p_{1:H}(x_{4,2n}) = \frac{1}{4n}$$

Figure 1: An illustrative example where using uniform virtual transition $p$ fails to guarantee $\min_{x_h \in \mathcal{X}_h, h \in [H]} p_{1:h}(x_h) \geq 1/X$.

On the IIEFG instance shown in Figure 1, there is only one action $a$ and $H = 4$. Each infoset $x$ in the game tree of this instance satisfies $|C(x,a)| = 2$ except for infoset $x_{2,1}$, which is such that $|C(x_{2,1},a)| = n$ with some $n \geq 2$. Now suppose the uniform distribution $p$ is used as a virtual transition over infoset-action spaces. Then for all the descendants $\{x_{4,i}\}_{i=1}^{2n}$ on step $h = 4$ of infoset $x_{2,1}$, one can see that $p_{1:H}(x_{H,i}) = \frac{1}{2} \cdot \frac{1}{n} \cdot \frac{1}{2} = \frac{1}{4n}$, while there are only $X = 9 + 3n$ infosets in total. Thus, it will happen that $p_{1:H}(x_{H,i}) < \frac{1}{X}$ when $n > 9$.

Actually, one can easily construct an IIEFG instance such that $\min_{x_H \in \mathcal{X}_H} p_{1:H}(x_H) \leq \mathcal{O}(\frac{1}{n^m})$ and $X = \mathcal{O}(mn + c)$ with $c$ as a parameter that depends on $m$ but not $n$ for uniform virtual transition $p$. Therefore, when using uniform distribution $p$ as a virtual transition, $\max_{x_H \in \mathcal{X}_H} 1/p_{1:H}(x_H)$ might be prohibitively large and lead to a convergence rate with much worse dependence on $X$ than the virtual transition constructed in our Algorithm 2.

### A.2  BALANCED EFFECTS OF THE PROPOSED VIRTUAL TRANSITION PROBABILITY

**Lemma A.1.** *For any $h \in [H]$ and $x_h \in \mathcal{X}_h$, the constructed virtual transition $p^x$ guarantees that $1/p^x_{1:h}(x_h) \leq X$.*

*Proof.* Clearly, $p^x_{1:h}(\cdot)$ is minimized at $h = H$ for some $x_H \in \mathcal{X}_H$ by the definition of virtual transition. By the construction of $p^x_{1:h}(\cdot)$ in Algorithm 2, one can deduce that $\forall x_H \in \mathcal{X}_H$, it holds that (understanding $\{(x_h, a_h)\}_{h \in [H-1]}$ as the unique trajectory leading to $x_H$ below)

$$p^x_{1:H}(x_H) = q[x_H]$$

$$= q[x_{H-1}] \cdot \frac{c[x_H]}{\sum_{x'_H \in C(x_{H-1}, a_{H-1})} c[x'_H]}$$

$$= q[x_{H-2}] \cdot \frac{c[x_{H-1}]}{\sum_{x'_{H-1} \in C(x_{H-2}, a_{H-2})} c[x'_{H-1}]} \cdot \frac{c[x_H]}{\sum_{x'_H \in C(x_{H-1}, a_{H-1})} c[x'_H]}$$

$$= q[x_{H-2}] \cdot \frac{c[x_{H-1}]}{\sum_{x'_{H-1} \in C(x_{H-2}, a_{H-2})} c[x'_{H-1}]} \cdot \frac{c[x_H]}{d[x_{H-1}, a_{H-1}]}$$

$$\overset{(i)}{\geq} q[x_{H-2}] \cdot \frac{c[x_{H-1}]}{\sum_{x'_{H-1} \in C(x_{H-2}, a_{H-2})} c[x'_{H-1}]} \cdot \frac{c[x_H]}{c[x_{H-1}]}$$

$$= q\left[x_{H-2}\right] \cdot \frac{c\left[x_H\right]}{\sum_{x'_{H-1} \in C(x_{H-2}, a_{H-2})} c\left[x'_{H-1}\right]}$$

$$\geq \dots$$

$$\geq \frac{c\left[x_H\right]}{\sum_{x_1 \in \mathcal{X}_1} c\left[x_1\right]}$$

$$\geq \frac{c\left[x_H\right]}{X_H}$$

$$\geq \frac{c\left[x_H\right]}{X}$$

$$= \frac{1}{X},$$

where $c[\cdot]$, $q[\cdot]$, and $d[\cdot, \cdot]$ are defined in our Algorithm 2; and $(i)$ is due to $c[x_{H-1}] = \max_{a \in \mathcal{A}} d[x_{H-1}, a] \geq d[x_{H-1}, a_{H-1}]$. $\qquad\square$

The property shown in this lemma of our constructed virtual transition $p^x$ serves as a key ingredient in the analysis (say, when bounding our **Term 4** and when establishing the final convergence upper bound of the NE gap in the proof of Theorem 5.1) as we shall see.

# B   PROOF OF HIGH-PROBABILITY LAST-ITERATE CONVERGENCE RATE

**Lemma B.1** (One-step analysis of OMD with virtual transition weighted negentropy regularized loss)**.** *Let*

$$\mu' = \operatorname{argmin}_{\tilde{\mu} \in \Omega} \sum_{h, x_h, a_h} \tilde{\mu}_{1:h}\left(x_h, a_h\right)\left(\ell\left(x_h, a_h\right) + \varepsilon\left(x_h, a_h\right) p^x_{1:h}\left(x_h\right) \log\left(p^x_{1:h}\left(x_h\right) \mu_{1:h}\left(x_h, a_h\right)\right)\right)$$

$$+ \frac{1}{\eta} D_\psi(\tilde{\mu}, \mu),$$

*for some convex set* $\Omega \subseteq \Pi_{\max}, \ell \in \mathbb{R}^{XA}_{\geq 0}$, *and* $\varepsilon \in \left[0, \frac{1}{\eta}\right]^{XA}$. *Then* $\forall u \in \Omega$.

$$\langle \mu' - \mu, \ell + \varepsilon p \log p\mu \rangle$$

$$\leqslant \sum_{h, (x_h, a_h)} \left[\frac{\eta}{p^x_{1:h}(x_h)} \mu_{1:h}(x_h, a_h)\ell^2(x_h, a_h) + \eta\varepsilon^2(x_h, a_h) \log^2(p^x_{1:h}\mu_{1:h}(x_h, a_h))\right],$$

*where* $(\varepsilon p \log p\mu)[(x_h, a_h)] := \varepsilon p^x_{1:h}(x_h) \log\left(p^x_{1:h}(x_h)\mu_{1:h}(x_h, a_h)\right)$.

*Proof.* The common one-step analysis of OMD shows that

$$\langle \mu' - \mu, \ell + \varepsilon p \log p\mu \rangle \leqslant \frac{1}{\eta}\left(D_\psi(u, \mu) - D_\psi(u, \mu') - D_\psi(\mu', \mu)\right).$$

Then, to upper bound $\langle \mu - \mu', l + \varepsilon p \log p\mu \rangle - \frac{1}{\eta} D_\psi(\mu', \mu)$, notice that

$$\langle \mu - \mu', \ell + \varepsilon p \log p\mu \rangle - \frac{1}{\eta} D_\psi(\mu', \mu)$$

$$\leqslant \sup_{v \in \mathbb{R}^{XA}_{\geq 0}} \left(\langle \mu - v, \ell + \varepsilon p \log p\mu \rangle - \frac{1}{\eta} D_\psi(v, \mu)\right)$$

$$= \langle \mu, \ell + \varepsilon p \log p\mu \rangle - \inf_{v \in \mathbb{R}^{XA}_{\geq 0}} \left(\langle v, \ell + \varepsilon p \log p\mu \rangle + \frac{1}{\eta} D_\psi(v, \mu)\right).$$

Further, the first-order optimality condition $\ell + \varepsilon p \log p\mu + \frac{1}{\eta}(\nabla\psi(v) - \nabla\psi(\mu)) = 0$ implies that

$$\log \frac{v_{1:h}\left(x_h, a_h\right)}{\mu_{1:h}\left(x_h, a_h\right)} = -\frac{\eta}{p^x_{1:h}\left(x_h\right)} \left[\ell\left(x_h, a_h\right) + \varepsilon\left(x_h, a_h\right) p^x_{1:h}\left(x_h\right) \log\left(p^x_{1:h}\left(x_h\right) \mu_{1:h}\left(x_h, a_h\right)\right)\right].$$

Hence, one can see that

$$v_{1:h}\left(x_h, a_h\right)$$
$$=\mu_{1:h}\left(x_h, a_h\right) \exp\left(-\frac{\eta}{p_{1:h}^x\left(x_h\right)}\left[\ell\left(x_h, a_h\right)+\varepsilon\left(x_h, a_h\right) p_{1:h}^x\left(x_h\right) \log\left(p_{1:h}^x\left(x_h\right) \mu_{1:h}\left(x_h, a_h\right)\right)\right]\right).$$

(10)

Therefore, we have

$$\left\langle\mu-\mu', \ell+\varepsilon p \log(p\mu)\right\rangle-\frac{1}{\eta} D_\psi\left(\mu', \mu\right)$$

$$=\sum_{h,(x_h, a_h)}\left[\left(\mu_{1:h}\left(x_h, a_h\right)-v_{1:h}\left(x_h, a_h\right)\right)\left(\ell\left(x_h, a_h\right)+\varepsilon\left(x_h, a_h\right) p_{1:h}^x\left(x_h\right) \log\left(p_{1:h}^x\left(x_h\right) \mu_{1:h}\left(x_h, a_h\right)\right)\right)\right.$$

$$\left.-\frac{1}{\eta}\left(p_{1:h}^x(x_h) v_{1:h}(x_h, a_h) \log\frac{v_{1:h}(x_h, a_h)}{\mu_{1:h}(x_h)}-p_{1:h}^x(x_h)(v_{1:h}(x_h, a_h)-\mu_{1:h}(x_h, a_h))\right)\right]$$

$$=\sum_{h,(x_h, a_h)}\left[\left(\mu_{1:h}\left(x_h, a_h\right)\right)\left(\ell\left(x_h, a_h\right)+\varepsilon\left(x_h, a_h\right) p_{1:h}^x\left(x_h\right) \log\left(p_{1:h}^x\left(x_h\right) \mu_{1:h}\left(x_h, a_h\right)\right)\right)\right.$$

$$\left.+\frac{p_{1:h}^x(x_h)}{\eta}\left(\exp\left(-\frac{\eta}{p_{1:h}^x(x_h)}[\ell(x_h, a_h)+\varepsilon(x_h, a_h) p_{1:h}^x(x_h) \log(p_{1:h}^x(x_h) \mu_{1:h}(x_h, a_h))]\right)-1\right) \mu_{1:h}(x_h, a_h)\right]$$

$$=\sum_{h,(x_h, a_h)}\frac{p_{1:h}^x(x_h)}{\eta} \mu_{1:h}(x_h, a_h)\left[\frac{\eta}{p_{1:h}^x(x_h)}\left(\ell\left(x_h, a_h\right)+\varepsilon\left(x_h, a_h\right) p_{1:h}^x\left(x_h\right) \log\left(p_{1:h}^x\left(x_h\right) \mu_{1:h}\left(x_h, a_h\right)\right)\right)\right.$$

$$\left.+\left(\exp\left(-\frac{\eta}{p_{1:h}^x(x_h)}[\ell(x_h, a_h)+\varepsilon(x_h, a_h) p_{1:h}^x(x_h) \log(p_{1:h}^x(x_h) \mu_{1:h}(x_h, a_h))]\right)-1\right)\right]$$

$$\leqslant\sum_{h,(x_h, a_h)}\frac{\eta}{p_{1:h}^x(x_h)} \mu_{1:h}(x_h, a_h) \ell^2(x_h, a_h)$$

$$+\sum_{h,(x_h, a_h)}\frac{p_{1:h}^x(x_h)}{\eta}\left[\mu_{1:h}\left(x_h, a_h\right)\left(\ell\left(x_h, a_h\right)+\varepsilon\left(x_h, a_h\right) p_{1:h}^x\left(x_h\right) \log\left(p_{1:h}^x\left(x_h\right) \mu_{1:h}\left(x_h, a_h\right)\right)\right)\right.$$

$$+\mu_{1:h}(x_h, a_h) \exp\left(-\frac{\eta}{p_{1:h}^x(x_h)}[\ell(x_h, a_h)+\varepsilon(x_h, a_h) p_{1:h}^x(x_h) \log(p_{1:h}^x(x_h) \mu_{1:h}(x_h, a_h))]\right)$$

$$\left.-\mu_{1:h}(x_h, a_h) \exp\left(-\frac{\eta}{p_{1:h}^x(x_h)} \ell(x_h, a_h)\right)\right]$$

$$=\sum_{h,(x_h, a_h)}\frac{\eta}{p_{1:h}^x(x_h)} \mu_{1:h}(x_h, a_h) \ell^2(x_h, a_h)$$

$$+\sum_{h,(x_h, a_h)}\frac{1}{\eta}\left[\mu_{1:h}\left(x_h, a_h\right) \eta\varepsilon\left(x_h, a_h\right) p_{1:h}^x\left(x_h\right) \log\left(p_{1:h}^x\left(x_h\right) \mu_{1:h}\left(x_h, a_h\right)\right)\right.$$

$$\left.+\exp\left(-\frac{\eta}{p_{1:h}^x} \ell(x_h, a_h)\right)\left(\left(p_{1:h}^x \mu_{1:h}(x_h, a_h)\right)^{1-\eta\varepsilon(x_h, a_h)}-p_{1:h}^x \mu_{1:h}(x_h, a_h)\right)\right]$$

$$\leqslant\sum_{h,(x_h, a_h)}\frac{\eta}{p_{1:h}^x(x_h)} \mu_{1:h}(x_h, a_h) \ell^2(x_h, a_h)$$

$$+\sum_{h,(x_h, a_h)}\frac{1}{\eta}\left[\mu_{1:h}\left(x_h, a_h\right) \eta\varepsilon\left(x_h, a_h\right) p_{1:h}^x\left(x_h\right) \log\left(p_{1:h}^x\left(x_h\right) \mu_{1:h}\left(x_h, a_h\right)\right)\right.$$

$$\left.-\eta\varepsilon(x_h, a_h)(p_{1:h}^x(x_h) \mu_{1:h}(x_h, a_h))^{1-\eta\varepsilon(x_h, a_h)} \ln(p_{1:h}^x(x_h) \mu_{1:h}(x_h, a_h))\right]$$

$$\leqslant\sum_{h,(x_h, a_h)}\left[\frac{\eta}{p_{1:h}^x(x_h)} \mu_{1:h}(x_h, a_h) \ell^2(x_h, a_h)+\eta\varepsilon^2(x_h, a_h) \log^2(p_{1:h}^x(x_h) \mu_{1:h}(x_h, a_h))\right],$$

where in the second equality we substitute $v_{1:h}(x_h, a_h)$ with Eq. (10), in first inequality comes from the fact that $\frac{\eta}{p_{1:h}^x(x_h)} \ell\left(x_h, a_h\right) \leqslant\left(\eta\ell\left(x_h, a_h\right) / p_{1:h}^x(x_h)\right)^2-\exp(\eta\ell\left(x_h, a_h\right) / p_{1:h}^x(x_h))$ and the

forth equality follows from

$$\exp\left(-\frac{\eta}{p_{1:h}^x(x_h)}[\ell(x_h,a_h)+\varepsilon(x_h,a_h)p_{1:h}^x(x_h)\log(p_{1:h}^x(x_h)\mu_{1:h}(x_h,a_h))]\right)$$

$$=\exp\left(-\frac{\eta}{p_{1:h}^x(x_h)}\ell(x_h,a_h)\right)((p_{1:h}^x(x_h)\mu_{1:h}(x_h,a_h))^{1-\eta\varepsilon(x_h,a_h)}\,.$$

The last two inequalities can be derived by following calculations:

$$\exp\left(-\frac{\eta}{p_{1:h}^x(x_h)}\ell(x_h,a_h)\right)((p_{1:h}^x(x_h)\mu_{1:h}(x_h,a_h))^{1-\eta\varepsilon(x_h,a_h)}-p_{1:h}^x(x_h)\mu_{1:h}(x_h,a_h))$$

$$\leqslant(p_{1:h}^x(x_h)\mu_{1:h}(x_h,a_h))^{1-\eta\varepsilon(x_h,a_h)}-p_{1:h}^x(x_h)\mu_{1:h}(x_h,a_h)$$

$$\leqslant\eta\varepsilon(x_h,a_h)(p_{1:h}^x(x_h)\mu_{1:h}(x_h,a_h))^{1-\eta\varepsilon(x_h,a_h)}\ln(p_{1:h}^x(x_h)\mu_{1:h}(x_h,a_h))\,,$$

and

$$\mu_{1:h}(x_h,a_h)\,\eta\varepsilon(x_h,a_h)\,p_{1:h}^x(x_h)\log(p_{1:h}^x(x_h)\,\mu_{1:h}(x_h,a_h))$$

$$-\eta\varepsilon(x_h,a_h)(p_{1:h}^x(x_h)\mu_{1:h}(x_h,a_h))^{1-\eta\varepsilon(x_h,a_h)}\ln(p_{1:h}^x(x_h)\mu_{1:h}(x_h,a_h))$$

$$=-\eta\varepsilon(x_h,a_h)\log(p_{1:h}^x(x_h)\,\mu_{1:h}(x_h,a_h))((p_{1:h}^x(x_h)\,\mu_{1:h}(x_h,a_h))^{1-\eta\varepsilon(x_h,a_h)}-p_{1:h}^x(x_h)\mu_{1:h}(x_h,a_h))$$

$$\leqslant\eta^2\varepsilon^2(x_h,a_h)(\log^2(p_{1:h}^x(x_h)\,\mu_{1:h}(x_h,a_h))^{1-\eta\varepsilon(x_h,a_h)}$$

$$\leqslant\eta^2\varepsilon^2(x_h,a_h)(\log^2(p_{1:h}^x(x_h)\,\mu_{1:h}(x_h,a_h))\,.$$

$\square$

**Lemma B.2.** $\forall k\geqslant 1$, we have

$$\mathrm{NEGap}(\xi^{k,*})=\mathcal{O}\left(\varepsilon_k H(\ln(XA)+\ln(YB))+\frac{XAH}{k}+\frac{YBH}{k}\right)\,. \tag{11}$$

*Proof.* $\forall(\mu',\nu')\in\Pi_{\max}\times\Pi_{\min}$, we have

$$f\left(\mu^{k,\star},\nu'\right)-f\left(\mu',\nu^{k,\star}\right)$$

$$=f\left(\mu^{k,\star},\nu'\right)-f\left(\mu^{k,\star},\nu\right)+\left(\mu^{k,\star},\nu\right)-f\left(\mu,\nu^{k,\star}\right)+f\left(\mu,\nu^{k,\star}\right)-f\left(\mu',\nu^{k,\star}\right)\,.$$

First notice that $\forall(\mu,v)\in\Pi_{\max}^k\times\Pi_{\min}^k$,

$$f\left(\mu^{k,\star},\nu\right)-f\left(\mu,\nu^{k,\star}\right)$$

$$=f\left(\mu^{k,\star},\nu\right)-f_k\left(\mu^{k,\star},\nu\right)+f_k\left(\mu^{k,\star},\nu\right)-f_k\left(\mu,\nu^{k,\star}\right)+f_k\left(\mu,\nu^{k,\star}\right)-f\left(\mu,\nu^{k,\star}\right)$$

$$=-\left(\varepsilon_k\psi\left(\mu^{k,\star}\right)-\varepsilon_k\psi(\nu)\right)+\left(\varepsilon_k\psi(\mu)-\varepsilon_k\psi\left(\nu^{k,\star}\right)\right)$$

$$\leqslant-\varepsilon_k\psi\left(\mu^{k,\star}\right)-\varepsilon_k\psi\left(\nu^{k,\star}\right)$$

$$\leqslant\varepsilon_k H(\ln(XA)+\ln(YB))\,.$$

To bound $f\left(\mu^{k,\star},\nu'\right)-f\left(\mu^{k,\star},\nu\right)$, we have

$$f\left(\mu^{k,\star},\nu'\right)-f\left(\mu^{k,\star},\nu\right)$$

$$\leqslant\left\langle\nabla_\nu f\left(\mu^{k,\star},\nu\right),\nu'-\nu\right\rangle$$

$$\leqslant\|\nabla_\nu f\left(\mu^{k,\star},\nu\right)\|_1\|\nu'-\nu\|_\infty$$

$$\leqslant YB\left(1-\left(1-\frac{B-1}{Bk}\right)^H\right)$$

$$\leqslant YB\left(1-\left(1-\frac{B}{Bk}\right)^H\right)$$

$$\leqslant YB\left(1-\left(1-\frac{1}{k}\right)^H\right)$$

$$= \mathcal{O}\left(\frac{YBH}{k}\right).$$

Similarly, we have

$$f\left(\mu, \nu^{k,\star}\right) - f\left(\mu', \nu^{k,\star}\right) \leqslant \mathcal{O}\left(\frac{XAH}{k}\right).$$

Putting all the above together completes the proof. $\qquad\square$

## B.1 CONVERGENCE RATE OF THE CONTRACTION OF THE BREGMAN DIVERGENCE

**Lemma B.3** (Contraction on Bregman divergence).

$$D_\psi\left(\xi^{k+1,\star}, \xi^{k+1}\right) \leqslant \underbrace{\sum_{i=1}^k w_k^i \eta_i \rho_i}_{\textbf{Term 1}} + \underbrace{\sum_{i=1}^k w_k^i \eta_i \sigma_i}_{\textbf{Term 2}}$$

$$+ \underbrace{XA\left(\log X + H\log\left(Ak\right)\right)^2 \sum_{i=1}^k w_k^i\left(\eta_i\varepsilon_i\right)^2 + YB\left(\log Y + H\log\left(Bk\right)\right)^2 \sum_{i=1}^k w_k^i\left(\eta_i\varepsilon_i\right)^2}_{\textbf{Term 3}}$$

$$+ \underbrace{\sum_{i=1}^k w_k^i \eta_i^2\left(X\underline{\tau}_i + Y\bar{\tau}_i\right)}_{\textbf{Term 4}} + \underbrace{\sum_{i=1}^k w_k^i \eta_i^2\left(X^2A + Y^2B\right)}_{\textbf{Term 5}} + \underbrace{\sum_{i=1}^k w_k^i \omega_i}_{\textbf{Term 6}}.$$

*Proof.* Recall we denote $[p^x\mu](x_h, a_h) := p_{1:h}^x(x_h)\mu_{1:h}(x_h, a_h)$.

$$f_k(\mu^k, \nu^k) - f_k(\mu^{k,\star}, \nu^k) = \left(\mu^k - \mu^{k,\star}\right)^\top \boldsymbol{G}\nu^k + \varepsilon_k\left(\psi\left(\mu^k\right) - \psi\left(\mu^{k,\star}\right)\right).$$

For the first term in the above display, we have

$$\left(\mu^k - \mu^{k,\star}\right)^\top \boldsymbol{G}\nu^k$$

$$= \left(\mu^k - \mu^{k,\star}\right)^\top \left(\boldsymbol{G}\nu^k + g^k - g^k\right)$$

$$= \left(\mu^k - \mu^{k,\star}\right)^\top g^k + \left(\mu^k\right)^\top \left(\boldsymbol{G}\nu^k - g^k\right) - \left(\mu^{k,\star}\right)^\top \left(\boldsymbol{G}\nu^k - g^k\right)$$

$$= \left(\mu^k - \mu^{k,\star}\right)^\top g^k + \sum_{h,x_h,a_h} \mu_{1:k}^k\left(x_h, a_h\right)\left[\left(\boldsymbol{G}\nu^k\right)\left[(x_h, a_h)\right] - g^k\left[(x_h, a_h)\right]\right]$$

$$- \sum_{h,x_h,a_h} \mu_{1:h}^{k,\star}\left(x_n, a_n\right)\left[\left(\boldsymbol{G}\nu^k\right)\left[(x_h, a_h)\right] - g^k\left[(x_h, a_h)\right]\right]$$

$$= \left(\mu^k - \mu^{k,\star}\right)^\top g^k$$

$$+ \sum_{h,x_h,a_h} \mu_{1:h}^k\left(x_h, a_h\right)\left[\left(\boldsymbol{G}\nu^k\right)\left[(x_h, a_h)\right] - \left(\frac{\mathbb{I}_h^k\left\{x_h, a_h\right\}}{\mu_{1:h}^k\left(x_h, a_h\right) + \gamma_k}\left(1 - r_h^k\right) + \varepsilon_k p_{1:h}^x\left(x_h\right)\log\left[p^x\mu^k\right]\left(x_n, a_n\right)\right)\right]$$

$$- \sum_{h,x_h,a_h} \mu_{1:h}^{k,\star}\left(x_h, a_h\right)\left[\left(\boldsymbol{G}\nu^k\right)\left[(x_h, a_h)\right] - \left(\frac{\mathbb{I}_h^k\left\{x_h, a_h\right\}}{\mu_{1:h}^k\left(x_h, a_h\right) + \gamma_k}\left(1 - r_h^k\right) + \varepsilon_k p_{1:h}^x\left(x_h\right)\log\left[p^x\mu^k\right]\left(x_n, a_n\right)\right)\right].$$

For the second term, we have

$$\psi\left(\mu^k\right) - \psi\left(\mu^{k,\star}\right)$$

$$= \sum_{h,x_h,a_h} \left[p^x\mu^k\right]\left(x_h, a_h\right)\log\left[p^x\mu^k\right]\left(x_h, a_h\right)$$

$$- \sum_{h,x_h,a_h} \left[p^x\mu^{k,\star}\right]\left(x_h, a_h\right)\log\left[p^x\mu^{k,\star}\right]\left(x_h, a_h\right)$$

$$= \sum_{h,x_h,a_h} \left( \left[ p^x \mu^k \right] (x_h, a_h) - \left[ p^x \mu^{k,\star} \right] (x_h, a_h) \right) \log \left[ p^x \mu^k \right] (x_h, a_h)$$

$$- \sum_{h,x_h,a_h} \left[ p^x \mu^{k,\star} \right] (x_h, a_h) \left( \log \left[ p^x \mu^{k,\star} \right] (x_h, a_h) - \log \left[ p^x \mu^k \right] (x_h, a_h) \right)$$

$$= \sum_{h,x_h,a_h} \left( \left[ p^x \mu^k \right] (x_h, a_h) - \left[ p^x \mu^{k,\star} \right] (x_h, a_h) \right) \log \left[ p^x \mu^k \right] (x_h, a_h) - D_\psi(\mu^{k,\star}, \mu^k).$$

We then arrive at

$$f_k(\mu_k, v_k) - f_k(\mu_k^\star, v_k)$$

$$= \left( \mu^k - \mu^{k,\star} \right)^\top g^k + \underbrace{\sum_{h,x_h,a_h} \mu_{1:k}^k (x_h, a_h) \left[ (\boldsymbol{G}\nu^k) \left[ (x_h, a_h) \right] - \left( \frac{\mathbb{I}_h^k \{x_h, a_h\}}{\mu_{1:h}^k (x_h, a_h) + \gamma_k} \left( 1 - r_h^k \right) \right) \right]}_{=:\underline{\rho_k}}$$

$$- \underbrace{\sum_{h,x_h,a_h} \mu_{1:k}^{k,\star} (x_h, a_h) \left[ (\boldsymbol{G}\nu^k) \left[ (x_h, a_h) \right] - \left( \frac{\mathbb{I}_h^k \{x_h, a_h\}}{\mu_{1:h}^k (x_h, a_h) + \gamma_k} \left( 1 - r_h^k \right) \right) \right]}_{=:\underline{\sigma_k}} - \varepsilon_k D_\psi(\mu^{k,\star}, \mu^k)$$

$$\leqslant \frac{1}{\eta_k} \left( D_\psi(\mu^{k,\star}, \mu^k) - D_\psi(\mu^{k,\star}, \mu^{k+1}) \right) - \varepsilon_k D_\psi(\mu^{k,\star}, \mu^k) + \underline{\rho_k} + \underline{\sigma_k}$$

$$+ \sum_{h,x_h,a_h} \eta_k \left( \frac{1}{p_{1:h}^x(x_h)} \mu_{1:h}^k(x_h, a_h) \hat{\ell}_h^k(x_h, a_h)^2 + \varepsilon_k^2(x_h, a_h) \log^2(p_{1:h}^x(x_h)\mu_{1:h}^k(x_h, a_h)) \right)$$

$$\leqslant \frac{(1 - \eta_k \varepsilon_k) D_\psi \left( \mu^{k,\star}, \mu^k \right) - D_\psi \left( \mu^{k,\star}, \mu^{k+1} \right)}{\eta_k} + \underline{\rho_k} + \underline{\sigma_k}$$

$$+ \sum_{h,x_h,a_h} \eta_k \left( \frac{1}{p_{1:h}^x(x_h)} \frac{\mathbb{I}_h^k \{x_h, a_h\}}{\mu_{1:h}^k (x_h, a_h) + \gamma_k} + \varepsilon_k^2 \log^2 ( \underbrace{p_{1:h}^x(x_h)\mu_{1:h}^k(x_h, a_h)}_{m := \min_{h,(x_h,a_h)} p_{1:h}^x(x_h)\mu_{1:h}^k(x_h, a_h)} ) \right)$$

$$\leqslant \frac{(1 - \eta_k \varepsilon_k) D_\psi \left( \mu^{k,\star}, \mu^k \right) - D_\psi \left( \mu^{k,\star}, \mu^{k+1} \right)}{\eta_k} + \underline{\rho_k} + \underline{\sigma_k}$$

$$+ \underbrace{\eta_k \sum_{h,x_h,a_h} \frac{1}{p_{1:h}^x(x_h)} \left( \frac{\mathbb{I}_h^k \{x_h, a_h\}}{\mu_{1:h}^k (x_h, a_h) + \gamma_k} - 1 \right)}_{=:X\underline{\tau_k}} + \eta_k X^2 A + \eta_k X A \log^2 m$$

$$\leqslant \frac{(1 - \eta_k \varepsilon_k) D_\psi \left( \mu^{k,\star}, \mu^k \right) - D_\psi \left( \mu^{k,\star}, \mu^{k+1} \right)}{\eta_k} + \underline{\rho_k} + \underline{\sigma_k} + \eta_k X\underline{\tau_k} + \eta_k X^2 A + \eta_k X A \log^2 m.$$

Rearranging shows that

$$D_\psi \left( \mu^{k+1,\star}, \mu^{k+1} \right)$$

$$\leqslant (1 - \eta_k \varepsilon_k) D_\psi \left( \mu^{k,\star}, \mu^k \right) + \eta_k \left( f_k \left( \mu^{k,\star}, \nu_k \right) - f_k \left( \mu_k, \nu_k \right) \right)$$

$$+ \eta_k^2 X A \log^2 m + \eta_k^2 X A \underline{\tau_k} + \eta_k^2 X^2 A + \eta_k \underline{\rho_k} + \eta_k \underline{\sigma_k} + \underbrace{D_\psi \left( \mu^{k+1,\star}, \mu^{k+1} \right) - D_\psi \left( \mu^{k,\star}, \mu^{k+1} \right)}_{=:\underline{\omega_k}}.$$

Analogously, for the min-player, we have

$$D_\psi \left( \nu^{k+1,\star}, \nu^{k+1} \right)$$

$$\leqslant (1 - \eta_k \varepsilon_k) D_\psi \left( \nu^{k,\star}, \nu^k \right) + \eta_k \left( f_k \left( \mu^k, \nu^k \right) - f_k \left( \mu_k, \nu^{k,\star} \right) \right)$$

$$+ \eta_k^2 Y B \log^2 m + \eta_k^2 Y \bar{\tau}_k + \eta_k^2 Y^2 B + \eta_k \bar{\rho}_k + \eta_k \bar{\sigma}_k + \bar{w}_k,$$

where

$$\bar{\tau}_k := \frac{1}{Y} \sum_{h,y_h,b_h} \frac{1}{p_{1:h}^y(y_h)} \left( \frac{\mathbb{I}_h^k \{y_h, b_h\}}{\nu_{1:h}^k (y_h, b_h) + \gamma_k} - 1 \right)$$

$$\bar{\rho}_k := \sum_{h,y_h,b_h} \nu_{1:h}^k (y_h, b_h) \left[ \left( 1 - \left( \boldsymbol{G}^\top \mu^k \right) [(y_h, b_h)] \right) - \frac{\mathbb{I}_h^k \{y_h, b_h\} r_h^k}{\nu_{1:h}^k (y_h, b_h) + \gamma_{kk}} \right]$$

$$\bar{\sigma}_k := \sum_{h,y_h,b_h} \nu_{1:h}^{k,\star} (y_h, b_h) \left[ \frac{\mathbb{I}_h^k \{y_h, b_h\} r_h^k}{\nu_{1:h}^k (y_h, b_h) + \gamma_{kk}} - \left( 1 - \left( \boldsymbol{G}^\top \mu^k \right) [(y_h, b_h)] \right) \right]$$

$$\bar{\omega}_k := D_\psi \left( \nu^{k+1,\star}, \nu^{k+1} \right) - D_\psi \left( \nu^{k,\star}, \nu^{k+1} \right) .$$

Combining both sides and noticing that $f_k \left( \mu^{k,\star}, \nu^k \right) - f_k \left( \mu^k, \nu^{k,\star} \right) \leqslant 0$, we have

$$D_\psi \left( \xi^{k+1,x}, \xi^{k+1} \right)$$
$$\leqslant (1 - \eta_k \varepsilon_k) D_\psi \left( \xi^{k,\star}, \xi^k \right) + \eta_k^2 \left( X \underline{\tau}_k + Y \bar{\tau}_k \right) + \eta_k^2 \left( X^2 A + Y^2 B \right) + \eta_k \rho_k + \eta_k \sigma_k + \omega_k$$
$$+ \eta_k^2 X A \varepsilon_k^2 \left( \log X + H \log (Ak) \right)^2 + \eta_k^2 Y B \varepsilon_k^2 \left( \log Y + H \log (Bk) \right)^2 .$$

Now expanding the recursion in the above display leads to

$$D_\psi \left( \xi^{k+1,\star}, \xi^{k+1} \right) \leqslant \underbrace{\sum_{i=1}^k w_k^i \eta_i \rho_i}_{\textbf{Term 1}} + \underbrace{\sum_{i=1}^k w_k^i \eta_i \sigma_i}_{\textbf{Term 2}}$$

$$+ \underbrace{X A \left( \log X + H \log (Ak) \right)^2 \sum_{i=1}^k w_k^i \left( \eta_i \varepsilon_i \right)^2 + Y B \left( \log Y + H \log (Bk) \right)^2 \sum_{i=1}^k w_k^i \left( \eta_i \varepsilon_i \right)^2}_{\textbf{Term 3}}$$

$$+ \underbrace{\sum_{i=1}^k w_k^i \eta_i^2 \left( X \underline{\tau}_i + Y \bar{\tau}_i \right)}_{\textbf{Term 4}} + \underbrace{\sum_{i=1}^k w_k^i \eta_i^2 \left( X^2 A + Y^2 B \right)}_{\textbf{Term 5}} + \underbrace{\sum_{i=1}^k w_k^i \omega_i}_{\textbf{Term 6}} ,$$

where $w_k^i = \prod_{j=i+1}^k (1 - \eta_j \varepsilon_j)$. $\qquad\square$

## B.2 BOUNDING CONTRACTION TERMS

**Lemma B.4** (Bounding **Term 1**).

$$\textbf{Term 1} \leqslant (XA + YB) \ln(k) k^{-\alpha_{\gamma_k} + \alpha_\varepsilon} + k^{-\frac{\alpha_k}{2} + \frac{\alpha_\varepsilon}{2}} \log \left( \frac{k^2}{\delta} \right) .$$

*Proof.* Recall

$$\textbf{Term 1} = \sum_{i=1}^k w_k^i \eta_i \rho_i = \sum_{i=1}^k w_k^i \eta_i \underline{\rho}_i + \sum_{i=1}^k w_k^i \eta_i \bar{\rho}_i .$$

To bound $\sum_{i=1}^k w_k^i \eta_i \underline{\rho}_i$, note that

$$\sum_{i=1}^k w_k^i \eta_i \underline{\rho}_i$$
$$= \sum_{i=1}^k w_k^i \eta_i \left\langle \mu^i, \ell^{i,x} - \hat{\ell}^{i,x} \right\rangle$$
$$= X A \sum_{i=1}^k w_k^i \eta_i \gamma_{ki} + H \sqrt{2 \sum_{i=1}^k \left( w_k^i \eta_i \right)^2 \log \frac{k^2}{\delta}}$$

$$\leqslant XA \sum_{i=1}^{k} \left[ i^{-\alpha_{\gamma_k} - \alpha_\eta} \prod_{j=i+1}^{k} \left(1 - j^{-\alpha_\eta - \alpha_\varepsilon}\right) \right] + \sqrt{\log\left(\frac{k^2}{\delta}\right) \sum_{i=1}^{k} \left[ i^{-2\alpha_\eta} \left( \prod_{j=i+1}^{k} \left(1 - j^{-\alpha_\gamma - \alpha_\varepsilon}\right) \right)^2 \right]}$$

$$\leqslant XA \sum_{i=1}^{k} \left[ i^{-\alpha_\gamma - \alpha_\eta} \prod_{j=i+1}^{k} \left(1 - j^{-\alpha_\eta - \alpha_\varepsilon}\right) \right] + \sqrt{\log\left(\frac{k^2}{\delta}\right) \sum_{i=1}^{k} \left[ i^{-2\alpha_\eta} \left( \prod_{j=i+1}^{k} \left(1 - j^{-\alpha_n - \alpha_\varepsilon}\right) \right) \right]}$$

$$\leqslant XA \ln(k) k^{-\alpha_\gamma + \alpha_\varepsilon} + \sqrt{\log\left(\frac{k^2}{\delta}\right) \ln(k) k^{-\alpha_\gamma + \alpha_\varepsilon}}$$

$$\leqslant XA \ln(k) k^{-\alpha_\gamma + \alpha_\varepsilon} + k^{-\frac{\alpha_\eta}{2} + \frac{\alpha_\varepsilon}{2}} \log\left(\frac{k^2}{\delta}\right),$$

where the second equality is given by Lemma B.13 and the third inequality comes from Lemma E.1.

Analogously, we have

$$\sum_{i=1}^{k} w_k^i \eta_i \bar{\rho}_i \leqslant YB \ln(k) k^{-\alpha_\gamma + \alpha_\varepsilon} + k^{-\frac{\alpha_k}{2} + \frac{\alpha_\varepsilon}{2}} \log\left(\frac{k^2}{\delta}\right).$$

Hence

$$\textbf{Term 1} \leqslant (XA + YB) \ln(k) k^{-\alpha_\gamma + \alpha_\varepsilon} + k^{-\frac{\alpha_k}{2} + \frac{\alpha_\varepsilon}{2}} \log\left(\frac{k^2}{\delta}\right).$$

$\square$

**Lemma B.5** (Bounding **Term 2**).

$$\textbf{Term 2} \leqslant k^{-\alpha_\eta + \alpha_{\gamma_k}} \log \frac{k^2}{\delta}.$$

*Proof.*

$$\textbf{Term 2} = \sum_{i=1}^{k} w_k^i \eta_i \sigma_i$$

$$= \sum_{i=1}^{k} w_k^i \eta_i \underline{\sigma}_i + \sum_{i=1}^{k} w_k^i \eta_i \bar{\sigma}_i$$

$$\leqslant \max_{1 \leqslant i \leqslant k} \frac{\eta_i w_k^i}{\gamma_{k_k}} \log \frac{k^2}{\delta} \quad \text{(with probability } 1 - \frac{k^2}{\delta}\text{)}$$

$$\leqslant k^{-\alpha_\eta + \alpha_{\gamma_k}} \log \frac{k^2}{\delta},$$

where the last inequality is due to Lemma E.2. $\square$

**Lemma B.6** (Bounding **Term 3**).

$$\textbf{Term 3} \leqslant \left( XA \left(\log X + H \log (Ak)^2\right) + YB \left(\log Y + H \log (Bk)\right)^2 \right) k^{-\alpha_\eta - \alpha_\varepsilon}.$$

*Proof.*

**Term 3**

$$= XA \left(\log X + H \log (Ak)\right)^2 \sum_{i=1}^{k} w_k^i (\eta_i \varepsilon_i)^2 + YB \left(\log Y + H \log (Bk)\right)^2 \sum_{i=1}^{k} w_k^i (\eta_i \varepsilon_i)^2$$

$$\leqslant \left( XA \left(\log X + H \log (Ak)\right)^2 + YB \left(\log Y + H \log (Bk)\right)^2 \right) k^{-2(\alpha_\eta + \alpha_\varepsilon) + \alpha_\eta + \alpha_\varepsilon}$$

$$= \left( XA \left(\log X + H \log (Ak)^2\right) + YB \left(\log Y + H \log (Bk)\right)^2 \right) k^{-\alpha_\eta - \alpha_\varepsilon},$$

where the inequality follows from Lemma E.1. $\square$

**Lemma B.7** (Bounding **Term 4**).
$$\textbf{Term 4} \leqslant k^{\alpha_{\gamma_k} - 2\alpha_\eta}(X + Y)\log\left(\frac{1}{\delta}\right).$$

*Proof.*

$$\textbf{Term 4}$$

$$= \sum_{i=1}^{k} w_k^i \eta_i^2 \left(X \underline{\tau}_i + Y \bar{\tau}_i\right)$$

$$= \sum_{i=1}^{k} w_k^i \eta_i^2 \left(X \cdot \frac{1}{X} \sum_{h, x_h, a_h} \frac{1}{p_{1:h}^x(x_h)} \left(\frac{\mathbb{I}_h^k\{x_h, a_h\}}{\mu_{1:h}^k(x_h, a_h) + \gamma_k} - 1\right)\right.$$

$$\left. + Y \cdot \frac{1}{Y} \sum_{h, y_h, b_h} \frac{1}{p_{1:h}^y(y_h)} \left(\frac{\mathbb{I}_h^k\{y_h, b_h\}}{\nu_{1:h}^k(y_h, b_h) + \gamma_k} - 1\right)\right)$$

$$\leqslant \max_{1 \leqslant i \leqslant k} \frac{w_k^i \eta_i^2 (X + Y)}{\gamma_k} \log(\frac{1}{\delta})$$

$$\leqslant k^{\alpha_\gamma - 2\alpha_\eta}(X + Y)\log\left(\frac{1}{\delta}\right),$$

where the first inequality follows from that $\frac{1}{X}\frac{1}{p_{1:h}^x(x_h)} \leqslant 1$ for all $(x_h, a_h)$ guaranteed by Lemma A.1 together with the use of Lemma B.15. $\qquad\square$

**Lemma B.8** (Bounding **Term 5**).
$$\textbf{Term 5} = \left(X^2 A + Y^2 B\right) k^{-\alpha_\eta + \alpha_\varepsilon}.$$

*Proof.*

$$\textbf{Term 5} = \sum_{i=1}^{k} w_k^i \eta_i^2 \left(X^2 A + Y^2 B\right)$$

$$\leqslant \left(X^2 A + Y^2 B\right) k^{-2\alpha_\eta + \alpha_\eta + \alpha_\varepsilon}$$

$$= \left(X^2 A + Y^2 B\right) k^{-\alpha_\eta + \alpha_\varepsilon},$$

where the inequality is given by Lemma E.1. $\qquad\square$

**Lemma B.9** (Bounding **Term 6**).
$$\textbf{Term 6} \leqslant (X + Y)^{\frac{1}{2}} \left(H \log\left(Ak\right) + H \log\left(Bk\right)\right) \cdot \log(k) k^{-\min\left\{1, \frac{3}{2} - \frac{\alpha_\varepsilon}{2}\right\} + \alpha_\eta + \alpha_\epsilon}.$$

*Proof.* To begin with, note that $\min_{(x_h, a_h) \in \mathcal{X}_h \times \mathcal{A}, h \in [H]} \mu_{1:h}^k(x_h, a_h) \geq \frac{1}{(Ak)^H}$ due to the definition of $\Pi_{\max}$ in Algorithm 1. Similarly, $\min_{(y_h, b_h) \in \mathcal{Y}_h \times \mathcal{B}, h \in [H]} \nu_{1:h}^k(y_h, b_h) \geq \frac{1}{(Bk)^H}$ holds for the min-player. Further,

$$\textbf{Term 6}$$

$$= \sum_{i=1}^{k} w_k^i \omega_i$$

$$\leqslant (X + Y)^{\frac{1}{2}} \log\left(\frac{1}{(Ak)^H}\right) \log\left(\frac{1}{(Bk)^H}\right) \sum_{i=1}^{k} w_k^i i^{-\min\left\{1, \frac{3}{2} - \frac{\alpha_\varepsilon}{2}\right\}}$$

$$\leqslant (X + Y)^{\frac{1}{2}} \log\left(\frac{1}{(Ak)^H}\right) \log\left(\frac{1}{(Bk)^H}\right) \log(k) k^{-\min\left\{1, \frac{3}{2} - \frac{\alpha_\varepsilon}{2}\right\} + \alpha_\eta + \alpha_\epsilon}$$

$$\leqslant (X + Y)^{\frac{1}{2}} \left(H \log\left(Ak\right) + H \log\left(Bk\right)\right) \log(k) k^{-\min\left\{1, \frac{3}{2} - \frac{\alpha_\varepsilon}{2}\right\} + \alpha_\eta + \alpha_\epsilon},$$

where the first inequality is due to Lemma B.10 and the second inequality comes from Lemma E.1. $\qquad\square$

## B.3 BOUNDING THE NE GAP OF $(\mu^{k,\star}, \nu^{k,\star})$

**Lemma B.10** (Bounding divergence difference).

$$|w_k| = \mathcal{O}\left(\frac{(X+Y)^{\frac{1}{2}}\left(\ln\left((Ak)^H\right) + \ln\left((Bk)^H\right)\right)^2}{k^{\min\left\{1,\frac{3}{2}-\frac{\alpha_\varepsilon}{2}\right\}}}\right).$$

*Proof.* Again, note that $\min_{(x_h,a_h)\in\mathcal{X}_h\times\mathcal{A}, h\in[H]} \mu_{1:h}^k(x_h, a_h) \geq \frac{1}{(Ak)^H}$ and $\min_{(y_h,b_h)\in\mathcal{Y}_h\times\mathcal{B}, h\in[H]} \nu_{1:h}^k(y_h, b_h) \geq \frac{1}{(Bk)^H}$. Therefore, it holds that

$$
\begin{aligned}
&|w_k| \\
&\leqslant \left|D_\psi\left(\mu^{k+1,\star}, \mu^{k+1}\right) - D_\psi\left(\mu^{k,\star}, \mu^{k+1}\right)\right| + \left|D_\psi\left(\nu^{k+1,\star}, \nu^{k-1}\right) - D_\psi\left(\nu^{k,\star}, \nu^{k+1}\right)\right| \\
&\leqslant \left(\ln\left((Ak)^H\right) + \ln\left((Bk)^H\right)\right)\left(\left\|p^x\mu^{k+1,\star} - p^x\mu^{k,\star}\right\|_1 + \left\|p^y\nu^{k+1,\star} - p^y\nu^{k,\star}\right\|_1\right) \\
&\leqslant \mathcal{O}\left(\frac{(X+Y)^{\frac{1}{2}}\left(\ln\left((Ak)^H\right) + \ln\left((Bk)^H\right)\right)^2}{k^{\min\left\{1,\frac{3}{2}-\frac{\alpha_\varepsilon}{2}\right\}}}\right),
\end{aligned}
$$

where the second inequality is due to Lemma B.11 and the last inequality comes from Lemma B.12. $\qquad\square$

**Lemma B.11** (Bounding divergence using $\ell_1$-norm). $\forall\mu,\mu^1,\mu^2 \in \Pi_{\max}^k$, *it holds that*

$$\left|D_\psi\left(\mu', \mu\right) - D_\psi\left(\mu^2, \mu\right)\right| \leqslant \mathcal{O}\left(\ln\left((Ak)^H\right)\left\|p^x\mu^1 - p^x\mu^2\right\|_1\right).$$

*Proof.*

$$
\begin{aligned}
&D_\psi\left(\mu', \mu\right) - D_\psi\left(\mu^2, \mu\right) \\
&= \sum_{h,(x_h,a_h)} p_{1:h}^x(x_h)\left(\mu_{1:h}^1(x_h,a_h)\log\frac{\mu_{1:h}^1(x_h,a_h)}{\mu_{1:h}(x_h\cdot a_h)} - \mu_{1:h}^2(x_h,a_h)\log\frac{\mu_{1:h}^2(x_h\cdot a_h)}{\mu_{1:h}(x_h,a_h)}\right) \\
&= \sum_{h,(x_h,a_h)} p_{1:h}^x(x_h)\left(\left(\mu_{1:h}^1(x_h,a_h) - \mu_{1:h}^2(x_h,a_h)\right)\log\frac{\mu_{1:h}^1(x_h,a_h)}{\mu_{1:h}(x_h\cdot a_h)}\right) \\
&\quad + \sum_{h,(x_h,a_h)} p_{1:h}^x(x_h)\mu_{1:h}^1(x_h,a_h)\left(\log\frac{\mu_{1:h}^1(x_h,a_h)}{\mu_{1:h}(x_h\cdot a_h)} - \log\frac{\mu_{1:h}^2(x_h,a_h)}{\mu_{1:h}(x_h\cdot a_h)}\right) \\
&\leqslant \mathcal{O}\left(\ln\left((Ak)^H\right)\left\|p^x\mu^1 - p^x\mu^2\right\|_1\right) - D_\psi(\mu^2, \mu^1) \\
&\leqslant \mathcal{O}\left(\ln\left((Ak)^H\right)\left\|p^x\mu^1 - p^x\mu^2\right\|_1\right).
\end{aligned}
$$

$\qquad\square$

**Lemma B.12** (Bounding $\ell_1$-norm of the difference between $\mu^{k,\star}$ and $\mu^{k+1,\star}$). *The $\ell_1$-norm of the difference between $\mu^{k,\star}$ and $\mu^{k+1,\star}$ satisfies*

$$\left\|p^z\xi^{k+1,\star} - p^z\xi^{k,\star}\right\|_1 = \mathcal{O}\left(\frac{(X+Y)^{\frac{1}{2}}\left(\ln\left((Ak)^H\right) + \ln\left((Bk)^H\right)\right)}{k^{\min\left\{1,\frac{3}{2}-\frac{\alpha_\varepsilon}{2}\right\}}}\right).$$

*Proof.* First note that, $\forall k, \forall(\mu,\nu) \in \Pi_{\max}^k \times \Pi_{\min}^k$, we have

$$
\begin{aligned}
&f_k\left(\mu, \nu^{k,\star}\right) - f_k\left(\mu^{k,\star}, \nu\right) \\
&= f_k\left(\mu, \nu^{k,\star}\right) - f_k\left(\mu^{k,\star}, \nu^{k,\star}\right) + f_k\left(\mu^{k,\star}, \nu^{k,\star}\right) - f_k\left(\mu^{k,\star}, \nu\right) \\
&\geqslant f_k\left(\mu, \nu^{k,\star}\right) - f_k\left(\mu^{k,\star}, \nu^{k,\star}\right) - \nabla_\mu f_k\left(\mu^{k,\star}, \nu^{k,\star}\right)^\top\left(\mu - \mu^{k,\star}\right) \\
&\quad - f_k\left(\mu^{k,\star}, \nu\right) - \left(-f_k\left(\mu^{k,\star}, \nu^{k,\star}\right)\right) - \left(-\nabla_\nu f_k\left(\mu^{k,\star}, \nu^{k,\star}\right)^\top\left(\nu - \nu^{k,\star}\right)\right) \\
&\geqslant \varepsilon_k D_\psi\left(\mu, \mu^{k,\star}\right) + \varepsilon_k D_\psi\left(\nu, \nu^{k,\star}\right)
\end{aligned}
$$

$$\begin{aligned}
&= \varepsilon_k \, \mathrm{KL}\left(p^x\mu, p^x\mu^{k,\star}\right) + \varepsilon_k \, \mathrm{KL}\left(p^y\nu, p^y\nu^{k,\star}\right) \\
&\geqslant \frac{1}{2}\varepsilon_k \left(\left\|p^x\mu - p^x\mu^{k,\star}\right\|_1^2 + \left\|p^y\nu - p^y\nu^{k,\star}\right\|_1^2\right) \\
&\geqslant \frac{1}{4}\varepsilon_k \left\|p^z\xi - p^z\xi^{k,\star}\right\|_1^2 \, .
\end{aligned}$$

Let $\mu^{k+1,\prime} = p_{k+1}\bar{\mu} + (1 - p_{k+1})\mu^{\star}_{k+1}$. Then $\forall h, (x_h, a_h)$,

$$\mu^{k+1,\prime}\left(a_h \mid x_h\right) \geqslant p_{k+1}\frac{1}{A} + (1 - p_{k+1})\frac{1}{A(k+1)^2} \geqslant \frac{1}{Ak^2} \, ,$$

which means that $\mu^{k+1,\prime} \in \Pi^k_{\max}$. Similarly, we define $\nu^{k+1,\prime}$, which is such that $\nu^{k+1,\prime} \in \Pi^k_{\min}$. By previous analysis, we have

$$f_k\left(\mu^{k+1,\prime}, \nu^{k,\star}\right) - f_k\left(\mu^{k,\star}, \nu^{k+1,\prime}\right) \geqslant \frac{1}{4}\varepsilon_k \left\|p^z\xi^{k+1,\prime} - p^z\xi^{k,\star}\right\|_1^2 \, . \tag{12}$$

On the other hand, since $\left(\mu^{k,\star}, \nu^{k,\star}\right) \in \Pi^{k+1}_{\max} \times \Pi^{k+1}_{\min}$, we have

$$f_{k+1}\left(\mu^{k,\star}, \nu^{k+1,\star}\right) - f_{k+1}\left(\mu^{k+1,\star}, \nu^{k,\star}\right) \geqslant \frac{1}{4}\varepsilon_{k+1} \left\|p^z\xi^{k,\star} - p^z\xi^{k+1,\star}\right\|_1^2 \, . \tag{13}$$

Combing both sides, we have

$$\begin{aligned}
&f_k\left(\mu^{k+1,\star}, \nu^{k,\star}\right) - f_k\left(\mu^{k,\star}, \nu^{k+1,\star}\right) \\
&= f_k\left(\mu^{k+1,\prime}, \nu^{k,\star}\right) - f_k\left(\mu^{k,\star}, \nu^{k+1,\prime}\right) + f_k\left(\mu^{k+1,\star}, \nu^{k,\star}\right) - f_k\left(\mu^{k+1,\prime}, \nu^{k,\star}\right) \\
&\quad + f_k\left(\mu^{k,\star}, \nu^{k+1,\prime}\right) - f_k\left(\mu^{k,\star}, \nu^{k+1,\star}\right) \\
&\geqslant \frac{1}{4}\varepsilon_k \left\|p^z\xi^{k+1,\prime} - p^z\xi^{k,\star}\right\|_1^2 + \left\langle\nabla_\mu f_k\left(\mu^{k+1,\prime}, \nu^{k,\star}\right), \mu^{k+1,\star} - \mu^{k+1,\star}\right\rangle \\
&\quad + \left\langle\nabla_\nu f_k\left(\mu^{k,\star}, \nu^{k+1,\prime}\right), \nu^{k+1,\prime} - \nu^{k+1,\star}\right\rangle \\
&\geqslant \frac{1}{4}\varepsilon_k \left\|p^z\xi^{k+1,\prime} - p^z\xi^{k,\star}\right\|_1^2 - \sup_{\mu \in \Pi^{k+1}_{\max}}\left\|\nabla_\mu f_k\left(\mu, \nu^{k,\star}\right)\right\|_\infty \left\|\mu^{k+1,\star} - \mu^{k+1,\prime}\right\|_1 \\
&\quad - \sup_{\nu \in \Pi^{k+1}_{\min}}\left\|\nabla_\nu f_k(\mu^{k,\star}, \nu)\right\|_\infty \left\|\nu^{k+1,\prime} - \nu^{k+1,\star}\right\|_1 \, .
\end{aligned}$$

Further using the fact that

$$\begin{aligned}
&\left\|\nabla_\mu f_k\left(\mu, \nu^{k,\star}\right)\right\|_\infty \\
&= \max_{h,(x_h,a_h)}\left|\boldsymbol{G}\nu^{k,\star}\left[(x_h, a_h)\right] + \varepsilon_k p^x_{1:h}(x_h)\log\left[p^x\mu\right]\left[(x_h, a_h)\right]\right| \\
&\leqslant \max_{h,(x_h,a_h)}\left|\boldsymbol{G}\nu^{k,\star}\left[(x_h, a_h)\right]\right| + \left|\varepsilon_k p^x_{1:h}(x_h)\log\left[p^x\mu\right]\left[(x_h, a_h)\right]\right| \\
&\leqslant 1 + k^{-\alpha\varepsilon}\left(\ln\left((Ak)^H\right) + \ln\left((Bk)^H\right)\right) = \mathcal{O}(1) \, ,
\end{aligned}$$

and

$$\begin{aligned}
\left\|\mu^{k+1,\star} - \mu^{k+1,\prime}\right\|_1 &= \left\|p_{k+1}\left(\bar{\mu} - \mu^{\star}_{k+1}\right)\right\|_1 \leqslant \left\|p_{k+1}\bar{\mu}\right\|_1 + \left\|p_{k+1}\mu^{\star}_{k+1}\right\|_1 \\
&\leqslant p_{k+1}2X = \mathcal{O}\left(\frac{X+Y}{k^2}\right) \, ,
\end{aligned}$$

one can deduce that

$$\begin{aligned}
&f_k\left(\mu^{k+1,\star}, \nu^{k,\star}\right) - f_k\left(\mu^{k,\star}, \nu^{k+1,\star}\right) \\
&\geqslant \frac{1}{8}\varepsilon_k\left\|p^z\xi^{k+1,\star} - p^z\xi^{k,\star}\right\|_1^2 - \frac{1}{4}\varepsilon_k\left\|p^z\xi^{k+1,\star} - p^z\xi^{k+1,\star}\right\|_1^2 - \mathcal{O}\left(\frac{X+Y}{k^3}\right) \\
&\geqslant \frac{1}{8}\varepsilon_k\left\|p^z\xi^{k+1,\star} - p^z\xi^{k,\star}\right\|_1^2 - \frac{1}{4}\varepsilon_k\left(2\left(\left\|p^x\mu^{k+1,\prime} - p^x\mu^{k+1,\star}\right\|_1^2 + \left\|p^y\nu^{k+1,\prime} - p^y\nu^{k+1,\star}\right\|_1^2\right)\right) \\
&\quad - \mathcal{O}\left(\frac{X+Y}{k^3}\right)
\end{aligned}$$

$$\geqslant \frac{1}{8}\varepsilon_k \left\| p^z \xi^{k+1,\star} - p^z \xi^{k,\star} \right\|_1^2 - \mathcal{O}\left(\frac{X+Y}{k^3}\right)$$

$$- \frac{1}{4}\varepsilon_k \left(4\left(\left\|p_{k+1}p^x \bar\mu\right\|_1^2 + \left\|p_{k+1}p^x \mu^{k+1,\star}\right\|_1^2\right) + 4\left(\left\|p_{k+1}p^y \bar\nu\right\|_1^2 + \left\|p_{k+1}p^y \nu^{k+1,\star}\right\|_1^2\right)\right)$$

$$= \frac{1}{8}\varepsilon_k \left\| p^z \xi^{k+1,\star} - p^z \xi^{k,\star} \right\|_1^2 - \mathcal{O}\left(\frac{X+Y}{k^3}\right) - \mathcal{O}\left(\frac{1}{k^6}\right)$$

$$\geqslant \frac{1}{8}\varepsilon_{k+1} \left\| p^z \xi^{k+1,\star} - p^z \xi^{k,\star} \right\|_1^2 - \mathcal{O}\left(\frac{X+Y}{k^3}\right).$$

Combining with Eq. (13), we have

$$\frac{3}{8}\varepsilon_{k+1}\|p^z \xi^{k+1,\star} - p^z \xi^{k,\star}\|_1^2$$

$$\leqslant f_{k+1}\left(\mu^{k,\star}, \nu^{k+1,\star}\right) - f_k\left(\mu^{k,\star}, \nu^{k+1,\star}\right) - f_{k+1}\left(\mu^{k,\star}, \nu^{k+1,\star}\right) + f_k\left(\mu^{k+1,\star}, \nu^{k,\star}\right) + \mathcal{O}\left(\frac{X+Y}{k^3}\right)$$

$$= \bar f_k\left(\mu^{k,\star}, \nu^{k+1,\star}\right) - \bar f_k\left(\mu^{k+1,\star}, \nu^{k,\star}\right) + \mathcal{O}\left(\frac{X+Y}{k^3}\right) \quad \left(\bar f_k(\mu, \nu) := f_{k+1}(\mu, \nu) - f_k(\mu, \nu)\right)$$

$$= \bar f_k\left(\mu^{k,\star}, \nu^{k+1,\star}\right) - \bar f_k\left(\mu^{k+1,\star}, \nu^{k+1,\star}\right) + \bar f_k\left(\mu^{k+1,\star}, \nu^{k+1,\star}\right) - \bar f_k\left(\mu^{k+1,\star}, \nu^{k,k}\right) + \mathcal{O}\left(\frac{X+Y}{k^3}\right)$$

$$\leqslant \left\langle \nabla_\mu \bar f_k\left(\mu^{k,\star}, \nu^{k+1,\star}\right), \mu^{k,\star} - \mu^{k+1,\star} \right\rangle + \left\langle \nabla_\nu \bar f_k\left(\mu^{k+1,\star}, \nu^{k,\star}\right), \nu^{k+1,\star} - \nu^{k,\star} \right\rangle + \mathcal{O}\left(\frac{X+Y}{k^3}\right)$$

$$= \left\langle \nabla_\mu \bar f_k\left(\mu^{k,\star}, \nu^{k+1,\star}\right)/p^x, p^x\left(\mu^{k,\star} - \mu^{k+1,\star}\right) \right\rangle + \left\langle \nabla_\nu \bar f_k\left(\mu^{k+1,\star}, \nu^{k,\star}\right)/p^y, p^y\left(\nu^{k+1,\star} - \nu^{k,\star}\right) \right\rangle + \mathcal{O}\left(\frac{X+Y}{k^3}\right)$$

$$\leqslant \left\|\nabla_\mu \bar f_k\left(\mu^{k,\star}, \nu^{k+1,\star}\right)/p^x\right\|_\infty \left\|p^x\left(\mu^{k,\star} - \mu^{k+1,\star}\right)\right\|_1 + \left\|\nabla_\nu \bar f_k\left(\mu^{k+1,\star}, \nu^{k,\star}\right)/p^y\right\|_\infty \left\|p^y\left(\nu^{k+1,\star} - \nu^{k,\star}\right)\right\|_1$$
$$+ \mathcal{O}\left(\frac{X+Y}{k^3}\right)$$

$$\leqslant \left(\sup_{\mu \in \Pi_{\max}^k} \left\|\nabla_\mu \bar f_k\left(\mu, \nu^{k+1,\star}\right)/p^x\right\|_\infty + \sup_{\nu \in \Pi_{\min}^k} \left\|\nabla_\nu \bar f_k\left(\mu^{k+1,\star}, \nu\right)/p^y\right\|_\infty\right) \left\|p^z \xi^{k+1,\star} - p^z \xi^{k,\star}\right\|_1 + \mathcal{O}\left(\frac{X+Y}{k^3}\right)$$

$$\leqslant \left(\sup_{\mu \in \Pi_{\max}^k} \max_{h,(x_h,a_h)} \left|(\varepsilon_k - \varepsilon_{k+1}) \log[p^x \mu][(x_h, a_h)]\right| + \sup_{\nu \in \Pi_{\min}^k} \max_{h,(y_h,b_h)} \left|(\varepsilon_k - \varepsilon_{k+1}) \log[p^y \nu][(y_h, b_h)]\right|\right)$$

$$\cdot \left\|p^z \xi^{k+1,\star} - p^z \xi^{k,\star}\right\|_1 + \mathcal{O}\left(\frac{X+Y}{k^3}\right)$$

$$= \mathcal{O}\left((\varepsilon_k - \varepsilon_{k+1})\left(\ln\left((Ak)^H\right) + \ln\left((Bk)^H\right)\right) \left\|p^z \xi^{k+1,\star} - p^z \xi^{k,\star}\right\|_1 + \frac{X+Y}{k^3}\right).$$

In what follows, we slightly abuse the notations by denoting $m_k = (Ak)^H (Bk)^H$. Solving the above equation leads to

$$\left\|p^z \xi^{k+1,\star} - p^z \xi^{k,\star}\right\|_1$$

$$\leqslant \frac{(\varepsilon_k - \varepsilon_{k+1}) \log(m_k) + \sqrt{(\varepsilon_k - \varepsilon_{k+1})^2 \log^2(m_k) + \varepsilon_{k+1}\frac{X+Y}{k^3}}}{\varepsilon_{k+1}}$$

$$\leqslant \frac{(\varepsilon_k - \varepsilon_{k+1})}{\varepsilon_{k+1}} \log(m_k) + \sqrt{\frac{X+Y}{\varepsilon_{k+1}k^3}}$$

$$\leqslant \frac{\log(m_k)}{k} + \sqrt{\frac{X+Y}{\varepsilon_{k+1}k^3}} = \mathcal{O}\left(\frac{(X+Y)^{\frac{1}{2}}\log(m_k)}{k^{\min\left\{1, \frac{3}{2} - \frac{\alpha}{2}\right\}}}\right).$$

In the last inequality of the above display, we use the fact that

$$\frac{(\varepsilon_k - \varepsilon_{k+1})}{\varepsilon_{k+1}} = \frac{k^{-\alpha_\epsilon}}{(k+1)^{-\alpha_\epsilon}} = (1 + \frac{1}{k})^{\alpha_\epsilon} - 1 = \mathcal{O}\left(\frac{\alpha_\epsilon}{k}\right),$$

by Taylor expansion. $\qquad\square$

**Lemma B.13.** *Let $\{c_i\}_{i=1}^k$ be fixed positive numbers. Then with probability at least $1 - \delta$, it holds that*

$$\sum_{i=1}^k c_i \left\langle \mu^i, \ell^{i,x} - \hat{\ell}^{i,x} \right\rangle \leqslant XA \sum_{i=1}^k c_i \gamma_{k_i} + H \sqrt{2 \sum_{i=1}^k c_i^2 \log \frac{1}{\delta}}\,.$$

*Proof.* To begin with, notice that

$$\sum_{i=1}^k c_i \left\langle \mu^i, \ell^{i,x} - \hat{\ell}^{i,x} \right\rangle = \sum_{i=1}^k c_i \left\langle \mu^i, \ell^{i,x} - \mathbb{E}_{i-1}\left[\hat{\ell}^{i,x}\right] \right\rangle + \sum_{i=1}^k c_i \left\langle \mu^i, \mathbb{E}_{i-1}\left[\hat{\ell}^{i,x}\right] - \hat{\ell}^{i,x} \right\rangle\,.$$

For the first part, we have

$$\sum_{i=1}^k c_i \left\langle \mu^i, \ell^{i,x} - \mathbb{E}_{i-1}\left[\hat{\ell}^{i,x}\right] \right\rangle$$

$$= \sum_{i=1}^k c_i \sum_{h,x_h,a_h} \mu_{1:h}^i(x_h, a_h)\, \ell_{[(x_h,a_h)]}^{i,x} \left(1 - \frac{\mu_{1:h}^i(x_h, a_h)}{\mu_{1:h}^i(x_h, a_h) + \gamma_{k_i}}\right)$$

$$\leqslant \sum_{i=1}^k c_i \gamma_{k_i} \sum_{h,x_h,a_h} \ell_{[(x_h,a_h)]}^{i,x}$$

$$\leqslant \sum_{i=1}^k c_i \gamma_{k_i} XA\,,$$

where the last inequality comes from $\ell[(x_h, a_h)]^{i,x} \leqslant 1$ for all $(x_h, a_h) \in \mathcal{X} \times \mathcal{A}$.

For the second part, taking $\delta = \exp\left(\frac{-\varepsilon^2}{2\sum_{i=1}^k c_i^2 H^2}\right)$, $\varepsilon = \sqrt{2\sum_{i=1}^k c_i^2 H^2 \log\left(\frac{1}{\delta}\right)}$ and using Azuma-Hoeffding inequality, it holds with probability at least $1 - \delta$ that

$$\sum_{i=1}^k c_i \left\langle \mu^i, \mathbb{E}_{i-1}\left[\hat{\ell}^{i,x}\right] - \hat{\ell}^{i,x} \right\rangle \leqslant \sqrt{2\sum_{i=1}^k c_i^2 H^2 \log\left(\frac{1}{\delta}\right)}\,.$$

The proof is concluded by combining the upper bounds of the two parts above. $\qquad\square$

**Lemma B.14.** *Let $\delta \in (0, 1)$ and $\{\gamma_{k_i}\}_{i=1}^k \in (0, +\infty)^k$. Fix $h \in [H]$. For any coefficient sequence $\{c^i\}_{i=1}^k$ s.t. $c^i \in \left[0, 2\gamma_k{}^i\right]^{XA}$ is $\mathcal{F}_{i-1}$ - measurable, with probability $1 - \delta$, we have*

$$\sum_{i=r}^k w_i \left\langle c_i, \hat{\ell}_i - \ell_i \right\rangle \leqslant \max_{1 \leqslant i \leqslant k} w_i \log \frac{1}{\delta}\,.$$

*Proof.* Define $w = \max_{1 \leqslant i \leqslant k} w_i$. Hence

$$w^i \hat{\ell}^i(x_h, a_h)$$

$$= \frac{w_i \mathbb{I}_{i,h}\{x_h, a_h\}\left(1 - r_h^i\right)}{\mu_{1:h}^i(x_h, a_h) + r_i}$$

$$\leqslant \frac{w_i \mathbb{I}_{i,h}\{x_h, a_h\}\left(1 - r_h^i\right)}{\mu_{i,h}^i(x_h, a_h) + r_i \frac{w_i\left(1 - r_h^i\right)\mathbb{I}_{i,h}\{x_h,a_h\}}{w}}$$

$$= \frac{w}{2\gamma_{k_i}} \frac{\frac{2\gamma_{k_i} w_i\left(1 - r_h^i\right)\mathbb{I}_{i,h}\{x_h,a_h\}}{w\mu_{i:h}^i(x_h,a_h)}}{1 + \frac{\gamma_{k_i} w_i\left(1 - r_h^i\right)\mathbb{I}_{i,h}\{x_h,a_h\}}{w\mu_{i:h}^i(x_h,a_h)}}$$

$$\leqslant \frac{w}{2\gamma_{k_i}} \log\left(1 + \frac{2\gamma_{k_i} w_i \left(1 - r_h^i\right) \mathbb{I}_{i,h}\{x_h, a_h\}}{w\mu_{i:h}^i(x_h, a_h)}\right).$$

Denote by $\hat{S}_h^i = \frac{w_i}{w}\left\langle c^i, \hat{\ell}_h^i\right\rangle$, $S_h^i = \frac{w_i}{w}\left\langle c^i, \ell_h^i\right\rangle$. Then

$$\mathbb{E}_{i-1}[\exp(\hat{S}^i)]$$

$$\leqslant \mathbb{E}_{i-1}\left[\exp\left(\sum_{(x_h, a_h)\in\mathcal{X}\times\mathcal{A}} \frac{c^i(x_h, a_h)}{2\gamma_{k_i}} \log\left(1 + \frac{2\gamma_{k_i} w_i \left(1 - r_h^i\right)\mathbb{I}_{i,h}\{x_h, a_h\}}{w\mu_{i:h}^i(x_h, a_h)}\right)\right)\right]$$

$$\leqslant \mathbb{E}_{i-1}\left[\prod_{(x_h, a_h)\in\mathcal{X}\times\mathcal{A}}\left(1 + \frac{c_i(x_h, a_h) w_i \left(1 - r_h^i\right)\mathbb{I}_{i,h}\{x_h, a_h\}}{w\mu_{i:h}^i(x_h, a_h)}\right)\right]$$

$$= \mathbb{E}_{i-1}\left[1 + \sum_{(x_h, a_h)\in\mathcal{X}\times\mathcal{A}} \frac{c^i\left(x_h, a_h\right) w_i \left(1 - r_h^i\right)\mathbb{I}_{i,h}\{x_h, a_h\}}{w\mu_{i:h}^i(x_h, a_h)}\right]$$

$$= 1 + S_h^i \leqslant \exp\left(S_h^i\right).$$

Finally, one can see that

$$\mathbb{E}\left[\sum_{i=1}^k \left(\hat{S}_h^i - S_h^i\right) \geqslant \log\frac{1}{\delta}\right]$$

$$= \mathbb{E}\left[\exp\left(\sum_{i=1}^k \left(\hat{S}_h^i - S_h^i\right)\right) \geqslant \frac{1}{\delta}\right]$$

$$\leqslant \delta\mathbb{E}\left[\exp\left(\sum_{i=1}^k \left(\hat{S}_h^i - S_h^i\right)\right)\right]$$

$$= \delta\mathbb{E}\left[\left[\mathbb{E}_{k-1}\left[\exp\left(\sum_{i=1}^k \left(\hat{S}_h^i - S_h^i\right)\right)\right]\right]\right]$$

$$= \delta\mathbb{E}\left[\exp\left(\sum_{i=1}^{k-1} \left(\hat{S}_h^i - S_h^i\right)\right)\left[\mathbb{E}_{k-1}\left[\exp\left(\hat{S}_h^k - S_h^k\right)\right]\right]\right] \leqslant \ldots \leqslant \delta.$$

$\square$

**Lemma B.15.** *Let $\{c_i\}_{i=1}^k$ be fixed positive numbers. Fix $h \in [H]$. Then $\forall$ sequence $\{q_i\}_{i=1}^k \in [0,1]^{XA}$ s.t. $q^i$ is $\mathcal{F}_{i-1}$ - measurable, with probability at least $1 - \delta$,*

$$\sum_{i=1}^k c_i \left\langle q_i, \hat{\ell}_h^i - \ell_h^i\right\rangle \leqslant \max_{1\leqslant i\leqslant k} \frac{c_i}{\gamma_{k_i}} \log\left(\frac{1}{\delta}\right).$$

*Proof.* Noticing that $\{\gamma_{k_i}\}_{i=1}^k$ is decreasing and $\|q^i\|_\infty \leqslant 1$, applying Lemma B.14, we arrive at

$$\sum_{i=1}^k c_i \left\langle q^i, \hat{\ell}_h^i - \ell_h^i\right\rangle = \sum_{i=1}^k \frac{c_i}{2\gamma_{k_i}} \left\langle 2\gamma_{k_i} q^i, \hat{\ell}_h^i - \ell_h^i\right\rangle \leqslant \max_{1\leqslant i\leqslant k} \frac{c_i}{\gamma_{k_i}} \log\left(\frac{1}{\delta}\right).$$

$\square$

## B.4 PROOF OF THEOREM 5.1

We are now ready to present the proof of our main result.

*Proof of Theorem 5.1.* Putting Lemma B.3, B.4, B.5, B.6, B.7, B.8, B.9 together, we have

$$D_\psi \left( \xi^{k+1,\star}, \xi^{k+1} \right)$$

$$= \mathcal{O} \left( (XA + YB) \ln(k) k^{-\alpha_{\gamma_k} + \alpha_\varepsilon} + k^{-\frac{\alpha_\eta}{2} + \frac{\alpha_\varepsilon}{2}} \log \left( \frac{k^2}{\delta} \right) + k^{-\alpha_\eta + \alpha_{\gamma_k}} \log \left( \frac{k^2}{\delta} \right) \right.$$

$$+ \left( XA \left( \log X + H \log (Ak)^2 + YB \left( \log Y + H \log (Bk) \right)^2 \right) k^{-\alpha_\eta - \alpha_\varepsilon} \right.$$

$$+ k^{\alpha_{\gamma_k} - 2\alpha_\eta} (X + Y) \log \left( \frac{1}{\delta} \right) + (X^2 A + Y^2 B) k^{-\alpha_\eta + \alpha_\epsilon}$$

$$+ (X + Y)^{\frac{1}{2}} \left( H \log (Ak) + H \log (Bk) \right) \left( \log X + H \log (k) + \log Y + H \log (Bk) \right)$$

$$\left. \cdot \log(k) k^{-\min\left\{ 1, \frac{3}{2} - \frac{\alpha_\varepsilon}{2} \right\} + \alpha_\eta + \alpha_\epsilon} \right)$$

$$= \mathcal{O} \left( \left[ k^{-\frac{1}{4}} (XA + YB) + k^{-\frac{1}{4}} + k^{-\frac{1}{4}} + (XA + YB) H^2 k^{-\frac{3}{4}} + (X + Y) k^{-\frac{7}{8}} + \left( X^2 A + Y^2 B \right) k^{-\frac{1}{2}} \right. \right.$$

$$\left. \left. + (X + Y)^{\frac{1}{2}} H^2 K^{-\frac{1}{4}} \right] \left( \log^2 (XAk/\delta) + \log^2 (YBk/\delta) \right) \log(K) \right) .$$

Moreover, note that

$$\text{NEGap} \left( \xi^k \right)$$

$$= \sup_{\mu \in \Pi_{\max}, \nu \in \Pi_{\min}} f \left( \mu^k, \nu \right) - f \left( \mu, \nu^k \right)$$

$$= f \left( \mu^{k,\star}, \nu \right) - f \left( \mu^{k,\star}, \nu \right) + f \left( \mu^k, \nu \right) - f \left( \mu, \nu^k \right) + f \left( \mu, \nu^{k,\star} \right) - f \left( \mu, \nu^{k,\star} \right)$$

$$\leqslant \text{NEGap} \left( \xi^{k,\star} \right) + \left( \mu^k - \mu^{k,\star} \right)^\top \boldsymbol{G} \nu + \mu^\top \boldsymbol{G} \left( \nu^{k,\star} - \nu^k \right)$$

$$\leqslant \text{NEGap} \left( \xi^{k,\star} \right) + \left\langle p^x \left( \mu^k - \mu^{k,\star} \right), \boldsymbol{G} \nu/p^x \right\rangle + \left\langle p^y \left( \nu^k - \nu^{k,\star} \right), \boldsymbol{G}^\top \mu/p^y \right\rangle$$

$$\leqslant \text{NEGap} \left( \xi^{k,\star} \right) + \left\| p^x \left( \mu^k - \mu^{k,\star} \right) \right\|_1 \left\| \boldsymbol{G} \nu/p^x \right\|_\infty + \left\| p^y \left( \nu^k - \nu^{k,\star} \right) \right\|_1 \left\| \boldsymbol{G}^\top \mu/p^y \right\|_\infty$$

$$\leqslant \text{NEGap} \left( \xi^{k,\star} \right) + X \left\| p^x \left( \mu^k - \mu^{k,\star} \right) \right\|_1 + Y \left\| p^y \left( \nu^k - \nu^{k,\star} \right) \right\|_1$$

$$\leqslant \varepsilon_k H \left( \ln(XA) + \ln(YB) \right) + \mathcal{O} \left( \frac{XAH}{k} + \frac{YBH}{k} \right)$$

$$+ \mathcal{O} \left( X \sqrt{\text{KL} \left( p^x \mu^{k,\star}, p^x \mu^k \right)} + Y \sqrt{\text{KL} \left( p^y \nu^{k,\star}, p^y \nu^k \right)} \right)$$

$$\leqslant \varepsilon_k H \left( \ln(XA) + \ln(YB) \right) + \mathcal{O} \left( \frac{XAH}{k} + \frac{YBH}{k} \right)$$

$$+ \mathcal{O} \left( (X + Y) \sqrt{\text{KL} \left( p^x \mu^{k,\star}, p^x \mu^k \right) + \text{KL} \left( p^y \nu^{k,\star}, p^y \nu^k \right)} \right)$$

$$\leqslant \varepsilon_k H \left( \ln(XA) + \ln(YB) \right) + \mathcal{O} \left( \frac{XAH}{k} + \frac{YBH}{k} \right) + \mathcal{O} \left( (X + Y) \sqrt{\text{KL} \left( p^z \xi^{k,\star}, p^z \xi^k \right)} \right)$$

$$\leqslant \varepsilon_k H \left( \ln(XA) + \ln(YB) \right) + \mathcal{O} \left( \frac{XAH}{k} + \frac{YBH}{k} \right) + \mathcal{O} \left( (X + Y) \sqrt{D_\psi \left( \xi^{k,\star}, \xi^k \right)} \right) ,$$

where $\boldsymbol{G} \nu/p^x \in \mathbb{R}^{XA}$ is defined as $(\boldsymbol{G} \nu/p^x)[(x_h, a_h)] = (\boldsymbol{G} \nu)[(x_h, a_h)]/p_{1:h}^x(x_h)$ and similarly for $\boldsymbol{G}^\top \mu/p^y$.

Therefore, we can see that

$$\text{NEGap}(\mu^k, \nu^k)$$

$$= \mathcal{O} \left( (X + Y) \left[ k^{-\frac{1}{8}} (XA + YB)^{\frac{1}{2}} + (XA + YB)^{\frac{1}{2}} H k^{-\frac{3}{8}} + \left( X^2 A + Y^2 B \right)^{\frac{1}{2}} k^{-\frac{1}{4}} + (X + Y)^{\frac{1}{4}} H K^{-\frac{1}{8}} \right] \right.$$

$$\left. \cdot \left( \log (XAk/\delta) + \log (YBk/\delta) \right) \log^{\frac{1}{2}}(k) + k^{-\frac{1}{8}} H (\ln(XA) + \ln(YB)) + \frac{XAB}{k} + \frac{YBH}{k} \right)$$

$$= \widetilde{\mathcal{O}} \left( (X + Y) \left[ k^{-\frac{1}{8}} (XA + YB)^{\frac{1}{2}} + (XA + YB)^{\frac{1}{2}} H K^{-\frac{3}{8}} + \left( X^2 A + Y^2 B \right)^{\frac{1}{2}} k^{-\frac{1}{4}} + (X + Y)^{\frac{1}{4}} H k^{-\frac{1}{8}} \right] \right.$$

$$+ \frac{(XAH + YBH)}{k} \Bigg)$$

$$= \widetilde{\mathcal{O}} \left( (X+Y)k^{-\frac{1}{8}} \left[ (XA + YB)^{\frac{1}{2}} + (X+Y)^{\frac{1}{4}} H \right] \right) ,$$

where the last equality holds when $k \geqslant \max\{H^4, {(X^2A + Y^2B)}^4/{(XA+YB)^4}, {(XA+YB)}^{8/7}/{(X+Y)}^{10/7}\}$.
□

## C  LAST-ITERATE CONVERGENCE RATE IN EXPECTATION

**Theorem C.1.** *With the same condition as in Theorem 5.1, Algorithm 1 guarantees that*

$$\mathbb{E}\left[ \mathrm{NEGap}(\mu^k, \nu^k) \right] = \widetilde{\mathcal{O}}\left( \left( (X+Y)^{\frac{1}{4}} H + \sqrt{(X^2A + Y^2B)} \right) k^{-\frac{1}{6}} \right) .$$

*Proof.* With the same arguments as in the proof of Theorem 5.1, we have

$$D_\psi\left( \xi^{k+1,x}, \xi^{k+1} \right)$$
$$\leqslant (1 - \eta_k \varepsilon_k) D_\psi\left( \xi^{k,\star}, \xi^k \right) + \eta_k^2 \left( X_{\underline{\tau}_k} + Y \bar{\tau}_k \right) + \eta_k^2 \left( X^2A + Y^2B \right) + \eta_k \rho_k + \eta_k \sigma_k + \omega_k$$
$$+ \eta_k^2 X A \varepsilon_k^2 \left( \log X + H \log (Ak) \right)^2 + \eta_k^2 Y B \varepsilon_k^2 \left( \log Y + H \log (Bk) \right)^2 .$$

Taking conditional expectation $\mathbb{E}_{k-1}[\cdot]$ on both sides and by noticing the fact that $\mathbb{E}_{k-1}[\tau_k] < 0$, $\mathbb{E}_{k-1}[\rho_k] = 0$, and $\mathbb{E}_{k-1}[\sigma_k] = 0$, we have

$$\mathbb{E}_{k-1}\left[ D_\psi\left( \xi^{k+1,x}, \xi^{k+1} \right) \right]$$
$$\leqslant (1 - \eta_k \varepsilon_k) D_\psi\left( \xi^{k,\star}, \xi^k \right) + \eta_k^2 \left( X^2A + Y^2B \right) + \mathbb{E}_{k-1}[\omega_k]$$
$$+ \eta_k^2 X A \varepsilon_k^2 \left( \log X + H \log (Ak) \right)^2 + \eta_k^2 Y B \varepsilon_k^2 \left( \log Y + H \log (Bk) \right)^2 .$$

Expanding the recursion in the above display leads to

$$\mathbb{E}\left[ D_\psi\left( \xi^{k+1,\star}, \xi^{k+1} \right) \right]$$
$$\leqslant \mathbb{E}\left[ \sum_{i=1}^k w_k^i \omega_i \right] + XA \left( \log X + H \log (Ak) \right)^2 \sum_{i=1}^k w_k^i \left( \eta_i \varepsilon_i \right)^2 + YB \left( \log Y + H \log (Bk) \right)^2 \sum_{i=1}^k w_k^i \left( \eta_i \varepsilon_i \right)^2$$
$$+ \sum_{i=1}^k w_k^i \eta_i^2 \left( X^2A + Y^2B \right)$$
$$\leqslant (X+Y)^{\frac{1}{2}} \left( H \log(Ak) + H \log(Bk) \right) \left( \log X + H \log(Ak) + \log Y + H \log(Bk) \right)$$
$$\quad \cdot \log(k) k^{-\min\{1, \frac{3}{2} - \frac{\alpha_\varepsilon}{2}\} - \alpha_\eta + \alpha_\varepsilon}$$
$$+ \left( XA \left( \log X + H \log (Ak) \right)^2 + YB \left( \log Y + H \log (Bk) \right)^2 \right) k^{-\alpha_\eta - \alpha_\varepsilon} + \left( X^2A + Y^2B \right) k^{-\alpha_\eta + \alpha_\varepsilon}$$
$$= \widetilde{\mathcal{O}}\left( (X+Y)^{\frac{1}{2}} H^2 k^{-\min\{1, \frac{3}{2} - \frac{\alpha_\varepsilon}{2}\} + \alpha_\eta + \alpha_\varepsilon} + (XA + YB) H^2 k^{-\alpha_\eta - \alpha_\varepsilon} + \left( X^2A + Y^2B \right) k^{-\alpha_\eta + \alpha_\varepsilon} \right) .$$

Hence,

$$\mathrm{NEGap}(\mu^k, \nu^k)$$
$$= \widetilde{\mathcal{O}}\Bigg( \varepsilon_k H + \frac{XAH}{k} + \frac{YBH}{k}$$
$$\quad + (X+Y) \left[ (X+Y)^{\frac{1}{4}} H k^{\left( -\min\{1, \frac{3}{2} - \frac{\alpha_\varepsilon}{2}\} + \alpha_\eta + \alpha_\varepsilon \right)/2} + \sqrt{(XA + YB)} + H k^{\frac{-\alpha_\eta - \alpha_\varepsilon}{2}} \right.$$
$$\quad + \left. \sqrt{(X^2A + Y^2B)} k^{\frac{-\alpha_\eta + \alpha_\varepsilon}{2}} \right] \Bigg)$$
$$= \widetilde{\mathcal{O}}\Bigg( k^{-\frac{1}{6}} H + \frac{XAH}{k} + \frac{YBH}{k} + (X+Y) \left[ (X+Y)^{\frac{1}{4}} H k^{-\frac{1}{6}} + \sqrt{(XA + YB)} H k^{-\frac{1}{3}} \right.$$

$$+ \sqrt{X^2 A + Y^2 B} k^{-\frac{1}{6}} \Big] \Big)$$

$$=\tilde{\mathcal{O}}\left((X + Y)\left[(X + Y)^{\frac{1}{4}} H + \sqrt{(X^2 A + Y^2 B)}\right] k^{-\frac{1}{6}}\right).$$

$\square$

## D  PROOF OF LOWER BOUND OF LAST-ITERATE CONVERGENCE

*Proof of Theorem 5.3.* Let $\mathrm{NEGap}_k := \mathrm{NEGap}\left(\mu^k, \nu^k\right)$ with $\left(\mu^k, \nu^k\right)$ as the policy profile generated by some algorithm Alg. Suppose that Alg leans the IIEFG with the last-iterate convergence rate of $\mathrm{NEGap}_k = \Theta\left(f(X, A) k^{-\alpha}\right)$ for some $\alpha \in (0, 1)$, where $f^{\mathrm{Alg}}(X, A)$ denotes the polynomial dependence on $X$ and $A$ of $\mathrm{NEGap}_k$.

Fix some $K \geqslant \max(XA, YB)$. Consider the regret defined as follows (Kozuno et al., 2021; Bai et al., 2022; Fiegel et al., 2023):

$$\mathrm{Reg}_K(\mathrm{Alg}) = \sup_{\mu \in \Pi_{\max}} \sum_{k=1}^{K} \left\langle \mu^k - \mu, \boldsymbol{G}\nu^k \right\rangle,$$

where $\{\nu^k\}_{k \in [K]}$ is potentially generated by an adversary. Then, one can deduce that

$$\mathrm{Reg}_K(\mathrm{Alg}) = \sup_{\mu \in \Pi_{\max}} \sum_{k=1}^{K} \left\langle \mu_k - \mu, \boldsymbol{G}\nu_k \right\rangle \tag{14}$$

$$\leqslant \sum_{k=1}^{K} \sup_{\mu \in \Pi_{\max}} \left\langle \mu_k - \mu, \boldsymbol{G}\nu_k \right\rangle$$

$$= \sum_{k=1}^{K} \sup_{\mu \in \Pi_{\max}} \mu_k^\top \boldsymbol{G}\nu_k - \mu^\top \boldsymbol{G}\nu_k$$

$$\leqslant \sum_{k=1}^{K} \sup_{\mu \in \Pi_{\max}, \nu \in \Pi_{\min}} \mu_k^\top \boldsymbol{G}\nu - \mu^\top \boldsymbol{G}\nu_k$$

$$= \sum_{k=1}^{K} \mathrm{NEGap}_k$$

$$= \Theta\left(f(X, A) \sum_{k=1}^{K} k^{-\alpha}\right)$$

$$= \Theta\left(f(X, A) K^{1-\alpha}\right). \tag{15}$$

On the other hand, by Theorem 6 of Bai et al. (2022) (see also Theorem 3.1 fo Fiegel et al. (2023)), we have

$$\mathrm{Reg}_K(\mathrm{Alg}) \geqslant \Omega(\sqrt{AXK}). \tag{16}$$

Combining Eq. (14) and Eq. (16), we have

$$\Omega(\sqrt{AXK}) \leqslant \Theta\left(f(X, A) K^{1-\alpha}\right).$$

We now further consider the following three cases:

- If $\alpha > \frac{1}{2}$, then $\sqrt{AX} \leqslant f(X, A) K^{\frac{1}{2} - \alpha}$. However, this does not hold for any $f$, when $K$ is large enough;

- If $\alpha = \frac{1}{2}$, it must hold that $\sqrt{AX} \leqslant f(X, A)$;

- If $\alpha < \frac{1}{2}$, then $\sqrt{AX} \leqslant f(X, A) K^{\frac{1}{2} - \alpha}$. This holds for all $f$, including $f(X, A) = 1$ when $K$ is large enough. In this case, the "minimal" $f$ is $f(X, A) = 1$, implying that the minimal possible convergence rate of $\mathrm{NEGap}_k$ in this case is $\mathrm{NEGap}_k = \Theta\left(k^{-\alpha}\right)$.

Taking the above three cases into account, the minimal possible convergence rate is

$$\min \left\{ \Theta \left( \sqrt{XA} k^{-\frac{1}{2}} \right), \Theta \left( k^{-\alpha} \right) \right\} \quad (\alpha > \frac{1}{2})$$
$$= \Theta \left( \sqrt{XA} k^{-\frac{1}{2}} \right).$$

Analogously, we can prove that $\text{NEGap}_k \geq \Theta(\sqrt{YB} k^{-\frac{1}{2}})$. Therefore, we have

$$\text{NEGap}_k \geq \Theta \left( \left( \sqrt{XA} + \sqrt{YB} \right) k^{-\frac{1}{2}} \right).$$

The proof is concluded by noticing that the above holds for all algorithms. $\square$

## E  AUXILIARY LEMMAS

**Lemma E.1** (Lemma 1 of Cai et al. (2023))**.** *Let* $0 < h < 1, 0 \leqslant k \leqslant 2$, *and let* $t \geqslant \left( \frac{24}{1-h} \ln \frac{12}{1-h} \right)^{\frac{1}{1-h}}$*. Then*

$$\sum_{i=1}^{t} \left( i^{-k} \prod_{j=i+1}^{t} \left( 1 - j^{-h} \right) \right) \leqslant 9 \ln(t) t^{-k+h}.$$

**Lemma E.2** (Lemma 2 of Cai et al. (2023))**.** *Let* $0 < h < 1, 0 \leqslant k \leqslant 2$, *and let* $t \geqslant \left( \frac{24}{1-h} \ln \frac{12}{1-h} \right)^{\frac{1}{1-h}}$*. Then*

$$\max_{1 \leqslant i \leqslant t} \left( i^{-k} \prod_{j=i+1}^{t} \left( 1 - j^{-h} \right) \right) \leqslant 4 t^{-k}.$$

**Lemma E.3** (Lemma 20 of Bai et al. (2020))**.** *Let* $c_1, c_2, \ldots, c_t$ *be fixed positive numbers. Then with probability at least* $1 - \delta$,

$$\sum_{i=1}^{t} c_i \left\langle x_i, \ell_i - \widehat{\ell}_i \right\rangle = \mathcal{O} \left( A \sum_{i=1}^{t} \beta_i c_i + \sqrt{\ln(A/\delta) \sum_{i=1}^{t} c_i^2} \right).$$

## F  OPTIMIZATION PROBLEM IN EQ. (3)

---

**Algorithm 3** Frank-Wolfe-type Algorithm for Solving Eq. (3) (max-player)

---

1: **Input:** Policy $\mu^k$ used in episode $k$, constrained policy space $\Pi_{\max}^{k+1}$, learning rate $\eta^{k+1}$, regularizer $\psi$, loss estimator $\widehat{\ell}^k$, number of iterations $T$.
2: **Initialize:** $\mu^{(1)} = \mu^k$, $\phi(\mu) = \eta^{k+1} \langle \mu, \widehat{\ell}^k \rangle + D_\psi(\mu, \mu^k)$.
3: **for** $t = 1, \ldots, T$ **do**
4:    Compute $g^{(t)} = \nabla \phi(\mu^{(t)})$.
5:    Compute $\widehat{\mu}^{(t)} = \arg\min_{\mu \in \Pi_{\max}^{k+1}} \langle \mu, g^{(t)} \rangle$ by Algorithm 4.
6:    Let $\delta = \frac{2}{1+t}$.
7:    Update $\mu^{(t+1)} = (1 - \delta)\mu^{(t)} + \delta \widehat{\mu}^{(t)}$.
8: **end for**
9: **Return** $\mu^{(T)}$.

---

In this section, we provide Algorithm 3 and Algorithm 4, which compute an approximate solution to Eq. (3).

---

**Algorithm 4** Computing Linear Minimizer in Algorithm 3 (max-player)

---

1: **Input:** $\Pi_{\max}^{k+1}, g^{(t)}$.
2: **Initialize:** $G^{(t)}(x_h, a_h) = 0, \mu(a_h|x_h) = 0, \forall(x_h, a_h) \in \mathcal{X}_h \times \mathcal{A}, \forall h \in [H]$.
3: **for** $h = H, \ldots, 1$ **do**
4:     **for** $x_h \in \mathcal{X}_H$ **do**
5:         Compute

$$G^{(t)}(x_h, a_h) = \sum_{x_{h+1} \in C(x_h, a_h), a_{h+1} \in \mathcal{A}} \mu(a_{h+1}|x_{h+1}) \left( g^{(t)}(x_{h+1}, a_{h+1}) + G^{(t)}(x_{h+1}, a_{h+1}) \right).$$

6:         Set $\mu(a_h|x_h) = \frac{1}{A(k+1)}, \forall a_h \in \mathcal{A}$.
7:         Set $\mu(a'_h|x_h) = 1 - \frac{A-1}{A(k+1)}$, where $a'_h = \arg\min_{a \in \mathcal{A}} g^{(t)}(x_h, a) + G^{(t)}(x_h, a)$.
8:     **end for**
9: **end for**
10: **Return** $\mu$.

---

**Computation Complexity** Suppose there are $K$ episodes. Let $w = \max_{h \in [H], (x_h, a_h) \in \mathcal{X}_h \times \mathcal{A}} |C(x_h, a_h)|$, where $C(x_h, a_h)$ is the set of immediate descendant in-fosets of $(x_h, a_h)$ as defined in Section 3. Then the computation complexity of our Algorithm 2 and Algorithm 1 will be of $\mathcal{O}(wXA)$ and of $\mathcal{O}(wXA + K(XA + \text{Oracle}))$, where Oracle denotes the computation complexity of an oracle algorithm to solve our Eq. (3). If Algorithm 3 and Algorithm 4 are adopted to solve an approximate solution to Eq. (3), then Oracle will be of $\mathcal{O}(wXAT)$ where $T$ is the number of iterations in Algorithm 3 and the total computation complexity of our Algorithm 1 will be of $\mathcal{O}(wXATK)$.

## G EXPERIMENTS

In this section, we present the empirical evaluations of our Algorithm 1. Since we are not aware of any other algorithm that can also learn the (approximate) NE policy profile in IIEFGs with provable *last-iterate convergence* guarantees under bandit feedback, we compare our algorithm against previous algorithms that converge to the (approximate) NE policy profile in IIEFGs with only *average-iterate convergence* guarantees including IXOMD (Kozuno et al., 2021), BalancedOMD (Bai et al., 2022) and BalancedFTRL (Fiegel et al., 2023). Since these algorithms are only devised to obtain the average-iterate convergence for learning IIEFGs, the last-iterate convergence of these algorithms for learning IIEFGs is not theoretically guaranteed.

**Environments** We consider four standard IIEFG instances including Lewis Signaling, Kuhn Poker (Kuhn, 1950), Leduc Poker (Southey et al., 2012) and Liars Dice. All the implementation of these games are from the OpenSpiel library (Lanctot et al., 2019).

**Implementation Details** For our algorithm, to save the computation costs, instead of using our Algorithm 3 and Algorithm 4 to solve Eq. (3) in Algorithm 1, we use a lazy update of our Algorithm 1, where only the policy of the experienced trajectory of infoset action pairs $\{(x_h^k, a_h^k)\}_{h \in [H]}$ in each episode $k$ are updated. For the remaining infoset action pairs that are not experienced by the max-player in episode $k$, the losses contributed by the entropy regularization (*i.e.*, the second term in our constructed entropy regularized loss estimator) of these infoset action pairs will be accumulated and will be used to update these infoset action pairs once they are experienced in some future episode, coming from the observation that the losses contributed by the entropy regularization are much smaller than the importance-weighted losses constructed using the rewards in the game (*i.e.*, the first term in our constructed entropy regularized loss estimator). In this way, the resulting computation complexity of our algorithm will only be of $\mathcal{O}(wXA + KXA)$ for running our algorithm in $K$ episodes where $w = \max_{h \in [H], (x_h, a_h) \in \mathcal{X}_h \times \mathcal{A}} |C(x_h, a_h)|$ ($C(x_h, a_h)$ is the set of immediate descendant infosets of $(x_h, a_h)$ as defined in Section 3). We adopt the implementation of all the baselines by Fiegel et al. (2023).[2] Besides, we consider a (logarithmic) grid search on the learning

---

[2]https://github.com/anon17893/IIG-tree-adaptation.

---

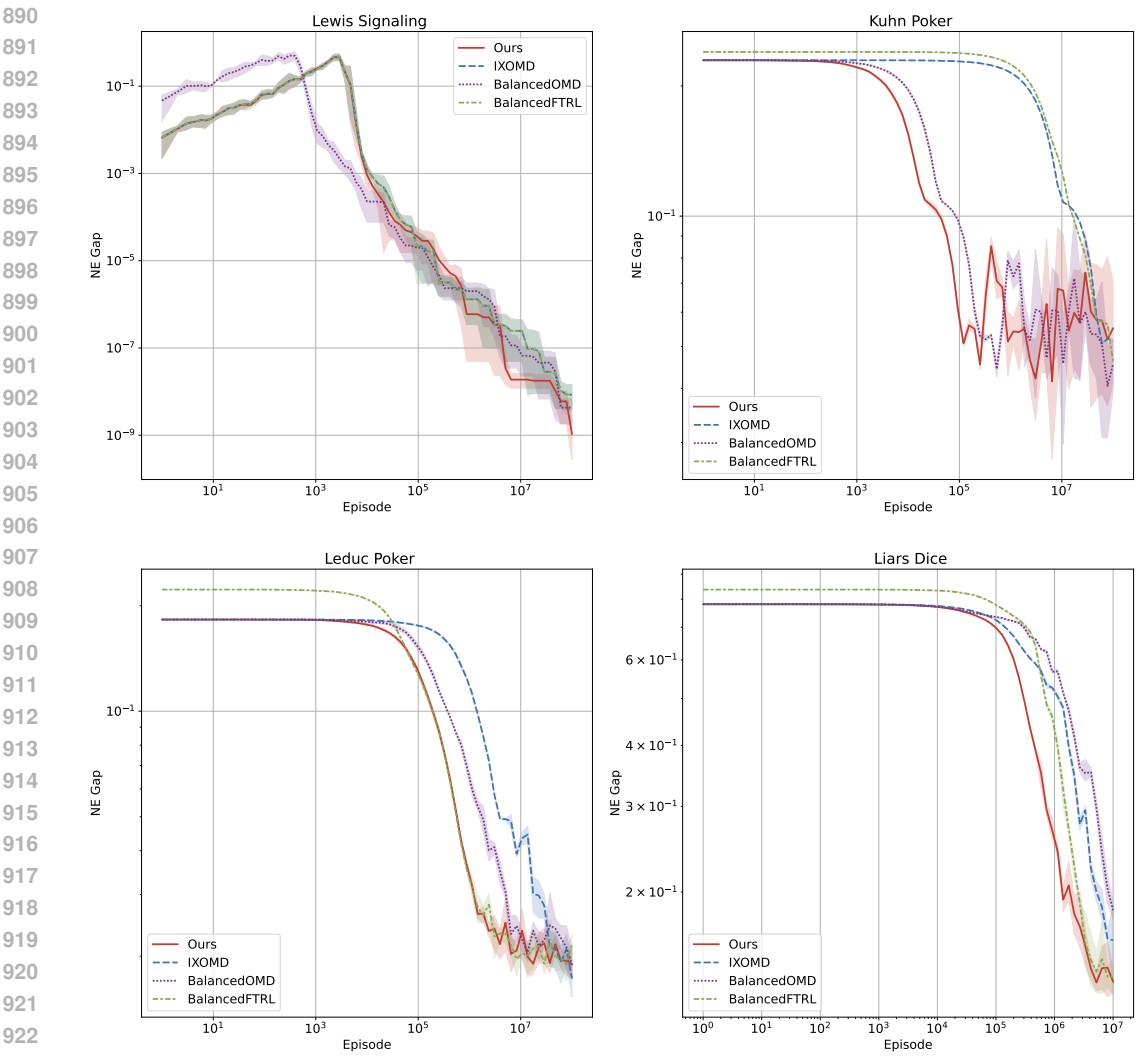

Figure 2: Experiment results of our Algorithm 1 against IXOMD (Kozuno et al., 2021), Balance-dOMD (Bai et al., 2022) and BalancedFTRL (Fiegel et al., 2023). The curves show the last-iterate convergence results of the NE gap defined in Eq. (2) against the number of episodes and are averaged over 5 different seeds.

rates for all the algorithms, following Fiegel et al. (2023). All the experiments are conducted on a server with an Intel Xeon Gold CPU and 251GiB system memory. The running of all the algorithms including our algorithm costs approximately 10 hours, 12 hours, 13 hours, and 16 hours on Lewis Signaling, Kuhn Poker, Leduc Poker, and Liars Dice, respectively.

**Results** The experimental results are shown in Figure 2. Our algorithm obtains the best or the competitive performance across all four IIEFG instances. In particular, our algorithm converges faster than all the baseline algorithms on Kuhn Poker and Liars Dice and also converges as fast as the empirically best baseline algorithm on Lewis Signaling and Leduc Poker. Though some baseline algorithms work relatively well on some game instances, we would like to note again that these algorithms are not theoretically guaranteed to converge to the NE policy profile with the last-iterate convergence. We speculate that this might also be the reason why some baseline algorithms perform relatively well in some instances but poorly in the remaining ones. For instance, the BalancedFTRL algorithm performs well on Leduc Poker while converging very slowly on Kuhn Poker. Analogously,

BalancedOMD converges relatively well on Kuhn Poker and Leduc Poker but converges the most slowly on Liars Dice.

Moreover, in general, it appears that the advantage of our algorithm becomes more pronounced in IIEFG instances with larger infoset spaces $\mathcal{X}$ (and action spaces $\mathcal{A}$) over previous algorithms. This observation aligns with the intuition that in such instances, the baseline algorithms, which solely have average-iterate convergence theoretical guarantees, face greater difficulty in achieving last-iterate convergence to the NE. This challenge may arise because these algorithms are more susceptible to getting stuck in suboptimal policy profiles, due to lack of the last-iterate convergence theoretical guarantees.