# OpenReview forum: "Learning Imperfect Information Extensive-form Games with Last-iterate Convergence under Bandit Feedback"
_ICLR.cc/2025/Conference — Submitted to ICLR 2025_

### Official Review · Reviewer_USyT · 2024-10-22

**Soundness:** 3
**Presentation:** 3
**Contribution:** 3
**Rating:** 5
**Confidence:** 4

**Summary:**

This paper establishes the first finite-time last-iterate convergence for learning an appeoximate NE of IIEFGs in the bandit feedback settting. They prove that their algorithm achieves the last-iterate convergence of order $O(k^{-1/8})$ with high probability and of order $O(k^{-1/6})$ in expectation. Also, they provide the lower bound of order $\Omega(k^{-1/2})$ for learning IIEFGs  with last-iterate convergence guarantee in the bandit feedback setting.

**Strengths:**

This paper presents the first finite-time last-iterate convergence results for learning an approximate NE of IIEFGs in the bandit feedback setting. The proposed algorithm achieves a last-iterate convergence rate of $O(k^{-1/8})$ with high probability, $O(k^{-1/6})$ in expectation, and establishes a lower bound of $\Omega(k^{-1/2})$.

**Weaknesses:**

In my view, the theoretical contributions of this paper are sound and without weakness. My concerns are primarily focused on the following aspects:

- The paper's analysis of the limitations of previous regularizers is overly obscure and difficult to comprehend. Firstly, I do not fully understand why the vanilla negentropy regularizer is generally difficult to control the NE gap, given that it directly regularizes the sequence-form representation policies. Secondly, the authors claim that bounding the stability term of OMD with dilated negentropy critically depends on its closed-form update solution. However, when solving Eq. (2), I find that no closed-form solution is provided.

- The benefits of the proposed regularizer are not clearly articulated. Based on the current version, I do not fully understand the advantages of the proposed regularizer, such as its impact on proving the convergence results of the proposed algorithm.

- The convergence results presented in the paper seem to rely on the exact solvability of Eq. (2). However, Eq. (2) only yields an approximate solution.

**Questions:**

- Could you explain the limitations of previous regularizers? It would be best if you could provide some simple examples.

- Could you elaborate on the contribution of your proposed regularizer to the convergence results of the algorithm?

- Could you provide the computational complexity for solving Eq. (2)?

- Could you offer a brief experimental analysis? For instance, discussing the convergence rate and runtime of the proposed algorithm in Kuhn Poker and Leduc Poker.

---

> ### Author Response · Authors · 2024-11-20
> **Author Response 1**
>
> We thank the reviewer for the valuable comments and suggestions. Our response to each question is provided in turn below.
>
> **Q1. "Firstly, I do not fully understand why ... the sequence-form representation policies."**
>
> We would like to remark that we believe it might be possible to obtain a convergence result of the policy profile $(\mu^k,\nu^k)$ in episode $k$ computed by the algorithm using vanilla negentropy regularizer to the NE policy profile $(\mu^{k,\star},\nu^{k,\star})$ of the regularized game in the sense that $\left\|\mu^k-\mu^{k, \star}\right\|\_1$ and $\left\|\nu^k-\nu^{k, \star}\right\|\_1$ are bounded. However, even if this is possible, it is still unclear (at least to us) how to convert this convergence of the policy profile to the convergence of the final NE gap, which is our ultimate learning objective. We refer the reviewer to our discussions in our response to **Q3** for details.
>
> **Q2. "Secondly, the authors claim that bounding the stability term of OMD with ... I find that no closed-form solution is provided."**
>
> Indeed, to our knowledge, bounding the stability term of OMD with Bregman divergence induced by the dilated negentropy regularizer depends on its closed-form update (please see our response to **Q3** of Reviewer xgRk for more detailed explanations on this). Nevertheless, we would like to note that our OMD-type Algorithm 1 utilizes Bregman divergence induced by a *virtual transition weighted negentropy* regularizer, instead of the *dilated negentropy* regularizer. Hence, the fact that bounding the stability term of our algorithm does not rely on the closed-form solution to Eq. (3) (in our revised paper) does not conflict with the fact that bounding the stability term of OMD with the dilated negentropy regularizer relies on their closed-form update.
>
> We will further clarify this point in the main body of the future version of our paper for better clarity.

---

> ### Author Response · Authors · 2024-11-20
> **Author Response 2**
>
> **Q3. "The benefits of the proposed regularizer are not clearly articulated. ..., such as its impact on proving the convergence results of the proposed algorithm."**
>
> On the positive side, the virtual transition weighted negentropy regularizer used in our Algorithm 1 mainly has the following advantages. First, with the leverage of such regularizer, as mentioned in Eq. (9) (in our revised paper), one is able to obtain an upper bound on $\operatorname{NEGap}$ by $\operatorname{NEGap}\left(\xi^k\right) \leqslant \operatorname{NEGap}\left(\xi^{k, \star}\right)+X\left\|p^x\left(\mu^k-\mu^{k, \star}\right)\right\|\_1+Y\left\|p^y\left(\nu^k-\nu^{k, \star}\right)\right\|\_1$, which involves the constructed virtual transition $p^x$ and $p^y$. Note that the upper bound $\left\|p^x\left(\mu^k-\mu^{k, \star}\right)\right\|\_1$ is tighter than $\left\|\mu^k-\mu^{k, \star}\right\|\_1$ because $p\_{1: h}^x\left(x\_h\right)\leq 1$ for all $x\_h\in\mathcal{X}$. More importantly, since $p\_{1: h}^x \cdot \mu\_{1: h}$ specifies a probability measure over infoset-action pairs as mentioned in Section 4.1, this enables us to further bound $\left\|p^x\left(\mu^k-\mu^{k, \star}\right)\right\|\_1$ by $\mathcal{O}\left(\sqrt{\mathrm{KL}\left(p^x \mu^{k, \star}, p^x \mu^k\right)}\right)$ via Pinsker’s inequality (which is not the case for the difference between the sequence-form representation of policies $\left\|\mu^k-\mu^{k, \star}\right\|\_1$). Second, also thanks to the leverage of the virtual transitions in our regularizer,  $\mathrm{KL}\left(p^x \mu^{k, \star}, p^x \mu^k\right)=D\_\psi\left(\mu^{k, \star}, \mu^k\right)$ if $\psi(\mu)=\sum\_{h, x\_h, a\_h} p\_{1: h}^x\left(x\_h\right) \mu\_{1: h}\left(x\_h, a\_h\right) \log \left(p\_{1: h}^x\left(x\_h\right) \mu\_{1: h}\left(x\_h, a\_h\right)\right)$ is the virtual transition weighted negentropy regularizer as defined in Section 4.1. Note that $D\_\psi\left(\mu^{k, \star}, \mu^k\right)$ is an (approximate) contraction mapping guaranteed by our algorithm. Therefore, intuitively, leveraging the virtual transition weighted negentropy regularizer facilitates the conversion from the convergence to the NE profile $\xi^{k,\star}$ to the final convergence of the NE gap, which serves as a key step in obtaining the last-iterate convergence of the NE gap.
>
> However, the potential downside of our virtual transition weighted negentropy regularizer is that the stability term of OMD is (approximately) enlarged by a factor inversely proportional to $\min\_{h\in[H],x\_h\in\mathcal{X}\_h} p^x(x\_h)$ as we discussed in Section 4.2 (please see Lemma B.1 in Appendix B of our revised paper). Moreover, the effect of $1/\min\_{h\in[H],x\_h\in\mathcal{X}\_h} p^x(x\_h)$ also appears in the analysis of bounding the contraction terms of the Bregman divergence (see, *e.g.*, the proof of Lemma B.7 for bounding Term 4 in Appendix B.2 of our revised paper) and the analysis when establishing the final convergence upper bound of the NE gap (please refer to the proof of Theorem 5.1 in Appendix B.4 of our revised paper). These potential negative impacts of the virtual transition necessitate designing a good virtual transition such that $1/\min\_{h\in[H],x\_h\in\mathcal{X}\_h} p^x(x\_h)$ is well upper bounded. This is directly controlled by our Algorithm 2, which constructs a special virtual transition $p^x$ and guarantees that $1/\min\_{h\in[H],x\_h\in\mathcal{X}\_h} p^x(x\_h)\le X$ for this special virtual transition $p^x$ (please see Lemma A.1 in our revised paper). This eventually guarantees a sharp dependence on the infoset space size $X$ of the final last-iterate convergence rate.
>
> **Q4. "The convergence results presented in the paper seem to rely on the exact solvability of Eq. (2). However, Eq. (2) only yields an approximate solution."**
>
> Please refer to our response to **Q2**.

---

> ### Author Response · Authors · 2024-11-20
> **Author Response 3**
>
> **Q5. "Could you explain the limitations of previous regularizers? It would be best if you could provide some simple examples."**
>
> The first regularizer that one might come up with is the dilated negentropy regularizer, which is commonly used in the analysis of establishing the average-iterate convergence results for learning IIEFGs [1,2]. The drawback of using such dilated negentropy regularizer is that it is highly unclear how to bound the stability term of the OMD with Bregman divergence induced by dilated negentropy regularizer when the closed-form update of OMD using dilated negentropy regularizer no longer holds (as mentioned in our response to **Q3** of Reviewer xgRk). This is exactly our case where we need to constrain the update of OMD onto a subset $\Pi\_{\max }^{k}$ of the full space of the sequence-form representation of policies $\Pi\_{\max }$ (mainly due to the leverage of the entropy regularization). The other potential regularizer might be the vanilla negentropy regularizer. However, as mentioned in our response to **Q1** and **Q3**, it is unclear (at least to us) whether it is possible to upper bound the final $\operatorname{NEGap}\left(\xi^k\right)$ as Pinsker’s inequality applies only to two probability measures and does not hold for two general sequence-form representation of policies $\mu\_1$ and $\mu\_2$ in IIEFGs.
>
> **Q6. "Could you elaborate on the contribution of your proposed regularizer to the convergence results of the algorithm?"**
>
> As we discussed in our response to **Q3**, the main benefit of our virtual transition weighted negentropy regularizer is that it facilitates the conversion from the convergence of the sequence-form representation of policy profiles to the final convergence of the NE gap via the leverage of the virtual transition. Moreover, we carefully design a virtual transition used in our virtual transition weighted negentropy regularizer, so that the minimum visitation probability contributed by the virtual transition $\min\_{h\in[H],x\_h\in\mathcal{X}\_h} p^x(x\_h)$ is well lower bounded, which is crucial to obtaining a sharp dependence on the infoset space size $X$ and $Y$ of the final convergence rate of the NE gap.
>
> **Q7. "Could you provide the computational complexity for solving Eq. (2)?"**
>
> Suppose there are $K$ episodes. Let $w=\max\_{h\in[H],(x\_h,a\_h)\in\mathcal{X}\_h\times\mathcal{A}}|C(x\_h,a\_h)|$, where $C(x\_h,a\_h)$ is the set of  immediate descendant infosets of $(x\_h,a\_h)$ as defined in Section 3. Then the computation complexity of our Algorithm 2 and Algorithm 1 will be of $\mathcal{O}(wXA)$ and of $\mathcal{O}(wXA+K(XA+\operatorname{Oracle}))$, where $\operatorname{Oracle}$ denotes the computation complexity of an oracle algorithm to solve our Eq. (3) (in the revised version of our paper). If  Algorithm 3 and Algorithm 4 are adopted to solve an approximate solution to Eq. (3) (in the revised version of our paper), then $\operatorname{Oracle}$ will be of $\mathcal{O}(wXAT)$ where $T$ is the number of iterations in Algorithm 3 and the total computation complexity of our Algorithm 1 will be of $\mathcal{O}(wXATK)$.
>
> We have now incorporated the above discussion of the computation complexity of our algorithm in Appendix F of our revised paper for better clarity (highlighted in green).

---

> ### Author Response · Authors · 2024-11-20
> **Author Response 4**
>
> **Q8. "Could you offer a brief experimental analysis? For instance, discussing the convergence rate and runtime of the proposed algorithm in Kuhn Poker and Leduc Poker."**
>
> Thanks for this comment. We now provide the empirical evaluations on Lewis Signaling, Kuhn Poker, Leduc Poker, and Liars Dice game instances. Since we are not aware of any other algorithm that can also learn the (approximate) NE policy profile in IIEFGs with provable *last-iterate convergence* guarantees under bandit feedback, we compare our algorithm against previous algorithms that converge to the (approximate) NE policy profile in IIEFGs with only *average-iterate convergence* guarantees on these game instances. In experiments, to save the computation costs, we use a lazy update of our Algorithm 1, where only the policy of the experienced trajectory of infoset action pairs $\set{(x^k\_h,a^k\_h)}\_{h\in[H]}$ in each episode $k$ are updated. For the remaining infoset action pairs that are not experienced by the learner in episode $k$, the losses contributed by the entropy regularization (*i.e.*, the second term in our constructed entropy regularized loss estimator) of these infoset action pairs will be accumulated and will be used to update these infoset action pairs once they are experienced in some future episode, coming from the observation that the losses contributed by the entropy regularization are much smaller than the importance-weighted losses constructed using the rewards in the game (*i.e.*, the first term in our constructed entropy regularized loss estimator). In this way, the resulting computation complexity of our algorithm will only be of $\mathcal{O}(wXA+KXA)$ for running our algorithm in $K$ episodes where $w=\max\_{h\in[H],(x\_h,a\_h)\in\mathcal{X}\_h\times\mathcal{A}}|C(x\_h,a\_h)|$. Our algorithm obtains the fastest convergence rates on Kuhn Poker and Liars Dice and the comparable performance on Lewis Signaling and Leduc Poker against baseline algorithms. In contrast, we observe that some baseline algorithms perform relatively well in some instances but poorly in the remaining ones. The runtime of all the algorithms, including ours, is approximately $10$ hours, $12$ hours, $13$ hours, and $16$ hours for Lewis Signaling, Kuhn Poker, Leduc Poker, and Liars Dice, respectively. Please see Appendix G in the revised version of our paper for detailed discussions on the experimental results (highlighted in green).
>
> ----
>
> We sincerely thank the reviewer again for the thoughtful feedback. We will also incorporate the additional discussions in our responses to **Q1**-**Q6** into the main body of the future version of our paper.
>
> [1] Kozuno et al. Model-Free Learning for Two-Player Zero-Sum Partially Observable Markov Games with Perfect Recall. NeurIPS, 21.
>
> [2] Bai et al. Near-Optimal Learning of Extensive-Form Games with Imperfect Information. ICML, 22.

---

> > ### Author Response · Authors · 2024-11-25
> > **Author Response 5**
> >
> > Dear Reviewer USyT,
> >
> > Thank you once again for generously dedicating your time to reviewing our paper and carefully considering our responses. As the author-reviewer discussion period is approaching its conclusion, we would greatly appreciate it if you could let us know if you have any remaining concerns or questions.
> >
> > We also noticed that you rated the **Presentation**, **Soundness**, and **Contribution** of our work each as 3, while providing an overall rating of 5. If there are any further points you would like us to clarify or discuss, we would be more than happy to address them within the remaining discussion period.
> >
> > Thank you again for your thoughtful feedback.
> >
> > Best regards,
> >
> > The Authors

---

> > > ### Comment · Reviewer_USyT · 2024-11-25
> > >
> > > Thank you for your responses, but I still have several points that remain unclear:
> > >
> > > - Regarding your reply to my Q1, I noticed that you stated the 1-norm in regularized games does not imply a reduction in the NE gap. However, this is not correct. For the NE of the original game, a well-known result is $NEgap(u, k) \leq NEgap(u^*, v^*) + \Vert u^* - u \Vert_1 + \Vert v^* - v \Vert$, where (u^*, v^*) is the NE of the regularized game. For the regularized game, the 1-norm is necessarily an upper bound on the NE gap, which makes it even simpler.
> > >
> > > - In your response to Q2, I do not understand what you mean by the "stability term of OMD." This term is quite uncommon and not widely recognized in the literature.
> > >
> > > - Regarding Q4, my question primarily concerns convergence. I do not understand why convergence holds if Eq. (2) cannot be solved. Without a closed-form solution, how can an exact solution be obtained? In prior works, I have not seen convergence results when there is no exact solution. Moreover, if you factor in the time required to solve Eq. (2), does your claim still hold? It seems the convergence rate of your algorithm would definitely decrease.

---

> > > > ### Author Response · Authors · 2024-11-25
> > > > **Author Response 6**
> > > >
> > > > Thanks for your replies. We provide our responses below.
> > > >
> > > > **Q1. "Regarding your reply to my Q1, I noticed that you stated the 1-norm in regularized games does not imply ..."**
> > > >
> > > > We apologize for the unclear parts in our previous response to Q1. There was a typo that $\left\\|\mu^k-\mu^{k, \star}\right\\|\_1$ and $\left\\|\nu^k-\nu^{k, \star}\right\\|\_1$ should be $D\_{\psi}(\mu^{k, \star},\mu^k)$ and $D\_{\psi}(\nu^{k, \star},\nu^k)$ for vanilla negentropy regularizer in our previous response to Q1. As mentioned in our response to Q3, the NE gap can be upper bounded by $\operatorname{NEGap}\left(\xi^k\right) \leqslant \operatorname{NEGap}\left(\xi^{k, \star}\right)+\left\\|\mu^k-\mu^{k, \star}\right\\|\_1+\left\\|\nu^k-\nu^{k, \star}\right\\|\_1$ (we adopt a slightly different form of $\operatorname{NEGap}\left(\xi^k\right) \leqslant \operatorname{NEGap}\left(\xi^{k, \star}\right)+X\left\\|p^x\left(\mu^k-\mu^{k, \star}\right)\right\\|\_1+Y\left\\|p^y\left(\nu^k-\nu^{k, \star}\right)\right\\|\_1$ that is better suited for the case when using our virtual transition weighted negentropy regularizer), which is also pointed out by the reviewer. Thus we originally did not mean that a convergence of $\left\\|\mu^k-\mu^{k, \star}\right\\|\_1$ does not imply a convergence on the NE gap. Instead, we originally intended to express that it seems unclear how to convert the convergence of $D\_{\psi}(\mu^{k, \star},\mu^k)$ for $\psi$ as the vanilla negentropy regularizer into the convergence of $\left\\|\mu^k-\mu^{k, \star}\right\\|\_1$ so as to obtain the final convergence of the NE gap. As illustrated in our response to Q3, if the vanilla negentropy regularizer is used, it is unclear how $\left\\|\mu^k-\mu^{k, \star}\right\\|\_1$ can be further upper bounded by $D\_{\psi}(\mu^{k, \star},\mu^k)$, as sequence-form representation of policies are generally not the probability measures over the infoset-action space. In contrast, by incorporating the virtual transition $p^x$, $p^x\mu$ will be a probability measure over the infoset-action space for any valid $\mu\in\Pi\_{\max}$ and thus $\left\\|p^x\left(\mu^k-\mu^{k, \star}\right)\right\\|\_1$ can be further bounded by $\mathcal{O}(\sqrt{\mathrm{KL}\left(p^x \mu^{k, \star}, p^x \mu^k\right)})=\mathcal{O}(\sqrt{D_\psi\left(\mu^{k, \star}, \mu^k\right)})$ via Pinsker’s inequality for $\psi$ as a virtual transition weighted negentropy regularizer.
> > > >
> > > > **Q2. "In your response to Q2, I do not understand what you mean by the "stability term of OMD." ..."**
> > > >
> > > > We are sorry for not explaining the terminology "stability term of OMD" explicitly. We originally used this terminology to denote
> > > > $$
> > > > \sum\_{k=1}^t\left\langle \mu^k-\mu^{k+1}, \widehat{\ell}^k\right\rangle-\frac{1}{\eta} \sum\_{k=1}^t D_{\psi}\left(\mu^{k+1}, \mu^k\right),
> > > > $$
> > > > which measures the part of the regret of OMD/FTRL contributed by "changing too much" between iterates (see, *e.g.*, Theorem 28.4 in [1]). The terminology "stability term of OMD" comes from the fact that if the prediction is completely stable ($i.e.$, $\mu^k=\mu^{k+1}$), $\left\langle \mu^k-\mu^{k+1}, \widehat{\ell}^k\right\rangle-\frac{1}{\eta}D_{\psi}\left(\mu^{k+1}, \mu^k\right)$ will be $0$. This terminology often appears in the online learning literature and can be found in, for example, [2,3].

---

> > > > > ### Author Response · Authors · 2024-11-25
> > > > > **Author Response 7**
> > > > >
> > > > > **Q3. "Regarding Q4, my question primarily concerns convergence. I do not understand why convergence holds if Eq. (2) cannot be solved ..."**
> > > > >
> > > > > Indeed, if a closed-form solution to the optimization problem does not exist, we generally can not guarantee the update can be exactly solved. However, we would like to note that "convergence rate" in our work denotes the sample complexity that measures using how many episodes the policy profile of the players will converge to an $\varepsilon$-NE policy profile. This metric is actually not related to the computation efficiency of the algorithm, which measures the computation complexity and the running time of the algorithm in each episode. More specifically, the metric of sample complexity is concerned with the  problem that if the policies $\set{\mu^1,\mu^2,\ldots}$ of OMD/FTRL can be exactly solved, using how many episodes will $\mu^k$ approach an $\varepsilon$-NE policy. In other words, the metric of sample complexity is not concerned with the computation complexity of computing $\mu^k$ in each episode $k$. Therefore, our claim will always hold, whether the time required to solve Eq. (2) is factored in or not.
> > > > >
> > > > > The sample complexity is guaranteed when the closed-form update of OMD/FTRL no longer holds can also be found in previous works studying the last-iterate convergence in matrix games and (fully-observable) Markov games in [4] and regret minimization in bandits/RL in *e.g.*, [5,6]. Even more broadly, there are many RL algorithms with optimal sample complexities but might require solving an NP-hard problem in the worst case in each episode (please see, for example, the discussions in Section 5 in [7]).
> > > > >
> > > > > ----
> > > > >
> > > > > We hope the above explanations would help to address the questions of the reviewer.
> > > > >
> > > > > [1] Lattimore et al. Bandit Algorithms. 2020.
> > > > >
> > > > > [2] [CSCI 659 Introduction to Online Optimization/Learning: Lecture 2](https://haipeng-luo.net/courses/CSCI659/2022_fall/lectures/lecture2.pdf)
> > > > >
> > > > > [3] Zimmert et al. Tsallis-INF: An Optimal Algorithm for Stochastic and Adversarial Bandits. JMLR, 21.
> > > > >
> > > > > [4] Cai et al. Uncoupled and Convergent Learning in Two-Player Zero-Sum Markov Games with Bandit Feedback. NeurIPS, 23.
> > > > >
> > > > > [5] Jin et al. Learning Adversarial Markov Decision Processes with Bandit Feedback and Unknown Transition. ICML, 20.
> > > > >
> > > > > [6] Liu et al. Towards Optimal Regret in Adversarial Linear MDPs with Bandit Feedback. ICLR, 24.
> > > > >
> > > > > [7] Jin et al. Bellman Eluder Dimension: New Rich Classes of RL Problems, and Sample-Efficient Algorithms. NeurIPS, 21.

---

> ### Comment · Reviewer_USyT · 2024-11-25
>
> Regarding your response to Q1, I respectfully disagree. Your statement holds true if the strategy space is not $\Pi_{max}$. However, since $\Pi_{max}$ is indeed under consideration, this can be formally proven. The existence of lower bounds for both $\log u$ and $\log v$ ensures the smoothness of the KL divergence.
>
> I also disagree with your response to Q3. If an algorithm is not implementable, it holds no practical value. This perspective might conflict with the views of certain researchers, but ultimately, algorithms are meant to be applied to solve real-world problems.

---

> ### Comment · Reviewer_USyT · 2024-11-25
>
> I notice that your statement seems to suggest that $\Vert \Vert^2_1 \leq KL()$ cannot be proven. However, this is essentially the definition of strong convexity, and I do not see any difficulty in this regard.

---

> > ### Author Response · Authors · 2024-11-25
> > **Author Response 8**
> >
> > Thanks for your replies. It is well known that $\operatorname{KL}$ is $1$-strongly convex w.r.t. $\ell_1$-norm on probability simplex and we would like to note that we are not saying that it is impossible to prove similar results on space of sequence-form representation of polices $\Pi_{\max}$. Instead, we believe our contribution lies in incorporating and designing a good virtual transition to leverage the strong convexity property of $\operatorname{KL}$ in a more convenient manner, by turning $\Pi_{\max}$ into the space of probability measures over infoset-action space and using the Bregman divergence of the virtual transition weighted regularizer as a surrogate.

---

### Official Review · Reviewer_fKCc · 2024-10-27

**Soundness:** 3
**Presentation:** 2
**Contribution:** 3
**Rating:** 6
**Confidence:** 2

**Summary:**

This paper studies learning in partially observable Markov games with bandit feedback. An algorithm is proposed that leverages the negentropy regularizer weighted by a virtual transition over infosetaction space. The proposed algorithm is proved to obtain last-iterate convergence. A lower bound for learning in imperfect-information extensive-form games under bandit feedback setting is also provided.

**Strengths:**

The considered partially observable setting is significant in practical.The results are well-structured, and the analysis is explained in a way that flows smoothly, making it easy to follow despite the technical complexity. The introduction of the virtual transition seems to be novel in the proof. Overall this paper is well-written.

**Weaknesses:**

My main concern is the lack of experiments to support the theoretical results. I would be particularly interested in seeing how the proposed algorithm performs in practical game scenarios.

**Questions:**

- What is the computational cost required in Algorithms 1 and 2?

- Is there any other method that can solve the game described in this paper? If so, how does the empirical performance compare?

Typo:
- In line 101, there are two 'to'.

---

> ### Author Response · Authors · 2024-11-20
> **Author Response 1**
>
> We thank the reviewer for the valuable comments and suggestions. Our response to each question is provided in turn below.
>
> **Q1. "What is the computational cost required in Algorithms 1 and 2?"**
>
> Suppose there are $K$ episodes. Let $w=\max_{h\in[H],(x_h,a_h)\in\mathcal{X}_h\times\mathcal{A}}|C(x_h,a_h)|$, where $C(x_h,a_h)$ is the set of immediate descendant infosets of $(x_h,a_h)$ as defined in Section 3. Then the computation complexity of our Algorithm 2 and Algorithm 1 will be of $\mathcal{O}(wXA)$ and of $\mathcal{O}(wXA+K(XA+\operatorname{Oracle}))$, where $\operatorname{Oracle}$ denotes the computation complexity of an oracle algorithm to solve our Eq. (3) (in the revised version of our paper). If  Algorithm 3 and Algorithm 4 are adopted to solve an approximate solution to Eq. (3) (in the revised version of our paper), then $\operatorname{Oracle}$ will be of $\mathcal{O}(wXAT)$ where $T$ is the number of iterations in Algorithm 3 and the total computation complexity of our Algorithm 1 will be of $\mathcal{O}(wXATK)$.
>
> We have now incorporated the above discussion of the computation complexity of our algorithm in Appendix F of our revised paper for better clarity (highlighted in green).
>
> **Q2. "My main concern is the lack of experiments to support the theoretical results. ... in practical game scenarios.""Is there any other method that can solve the game described in this paper? If so, how does the empirical performance compare?"**
>
> We would like to point out that we believe the main contribution of our work lies in devising the first algorithm for learning the (approximate) NE policy profile in IIEFGs with provable last-iterate convergence guarantees under bandit feedback. As mentioned in our paper, to the best of our knowledge, we are not aware of any other algorithm with similar theoretical guarantees in the same setting. Therefore, we compare our algorithm against previous algorithms that converge to the (approximate) NE policy profile in IIEFGs with only *average-iterate convergence* guarantees on Lewis Signaling, Kuhn Poker, Leduc Poker, and Liars Dice game instances. Our algorithm obtains the fastest convergence rates on Kuhn Poker and Liars Dice and the comparable performance on Lewis Signaling and Leduc Poker against baseline algorithms. In contrast, we observe that some baseline algorithms perform relatively well in some instances but poorly in the remaining ones. Please see Appendix G in the revised version of our paper for more detailed discussions on the experimental results (highlighted in green).
>
> **Q3. Typo.**
>
> Thanks for pointing this out. We have now corrected this typo in the revised version of our paper.

---

> > ### Author Response · Authors · 2024-11-25
> > **Author Response 2**
> >
> > Dear Reviewer fKCc,
> >
> > Thank you once again for dedicating your valuable time to reviewing our paper and considering our responses. As the author-reviewer discussion period is nearing its conclusion, we would greatly appreciate it if you could let us know whether you have any remaining concerns.
> >
> > In particular, we noticed that you rated the **Presentation** of our work as 2, while scoring both **Soundness** and **Contribution** as 3. We would be more than happy to address any remaining questions or clarify any aspects related to the presentation of our work.
> >
> > Thank you for your time and thoughtful feedback.
> >
> > Best regards,
> >
> > The Authors

---

> > > ### Comment · Reviewer_fKCc · 2024-11-26
> > >
> > > I thank the authors for detailed response and increase the score.

---

> > > > ### Author Response · Authors · 2024-11-26
> > > > **Author Response 3**
> > > >
> > > > Thank you for raising your rating for our work!

---

### Official Review · Reviewer_xwCS · 2024-11-05

**Soundness:** 4
**Presentation:** 4
**Contribution:** 4
**Rating:** 8
**Confidence:** 4

**Summary:**

This paper studies the problem of learning approximate NEs in IIEFGs, with the perfect-recall assumption and bandit feedback. They propose an algorithm based on a negentropy regularizer weighted by a virtual transition. The proposed algorithm achieves finite-time last-iterate convergence and is fully uncoupled between the two players involved.

**Strengths:**

This paper is in general well-written and smooth to follow. The authors did a great job in introducing the background literature, and in motivating the design of the proposed algorithm. The theoretical results, namely the finite-time last-iterate convergence for learning approximate NEs in IIEFGs, are of vital importance in the literature, and the theoretical proofs seem to be rigorous. The results of this paper serve as an important step towards improving the last-iterate convergence guarantee.

**Weaknesses:**

Although the theoretical side of this paper is strong, simulation parts are completely missing. I would appreciate the authors could simulate the proposed algorithm on several example settings to validate the last-iterate convergence and potentially the rate of convergence.

**Questions:**

- Intuitively, why do you choose subset $\Pi_{\max }^{k+1}$ such that  $\mu\left(a_h \mid x_h\right) \geqslant \frac{1}{A(k+1)}$, and why constraining this feasible set won't affect the convergence?
- It is stated that employing the virtual transition helps to operate OMD in the space of visitation measure instead of sequence-form representations of policies. However, as is shown in Eq (5), the virtual transition is directly related to a sequence-form representation. Can the authors discuss more on this? And why operating OMD in the space of visitation measure is important?
- It seems that the virtual transition is fixed at the beginning (once the game tree is fixed), could one potentially define an adaptive virtual transition, utilizing the historical bandit feedback up till now to further improve the convergence rate?

---

> ### Author Response · Authors · 2024-11-20
> **Author Response 1**
>
> We thank the reviewer for the valuable comments and suggestions. Our response to each question is provided in turn below.
>
> **Q1. "Although the theoretical side of this paper is strong, simulation parts are completely missing ... validate the last-iterate convergence and potentially the rate of convergence."**
>
> Thanks for this comment. We now provide the empirical evaluations on Lewis Signaling, Kuhn Poker, Leduc Poker, and Liars Dice game instances. Since we are not aware of any other algorithm that can also learn the (approximate) NE policy profile in IIEFGs with provable *last-iterate convergence* guarantees under bandit feedback, we compare our algorithm against previous algorithms that converge to the (approximate) NE policy profile in IIEFGs with only *average-iterate convergence* guarantees on these game instances. Our algorithm obtains the fastest last-iterate convergence rates on Kuhn Poker and Liars Dice and the comparable performance on Lewis Signaling and Leduc Poker against baseline algorithms. In contrast, we observe that some baseline algorithms perform relatively well in some instances but poorly in the remaining ones. Please see Appendix G in the revised version of our paper for more detailed discussions on the experimental results (highlighted in green).
>
> **Q2. "why do you choose subset $\Pi_{\max}^{k+1}$ such that $\mu\left(a_h \mid x_h\right) \geqslant \frac{1}{A(k+1)}$, and why constraining this feasible set won't affect the convergence?"**
>
> We consider constraining the optimization of OMD onto a subset $\Pi_{\max}^{k}$ of the full space of the sequence-form representation of policies $\Pi_{\max}$ mainly to prevent the second term in the constructed entropy regularized loss estimator from being prohibitively large (Line 6 of our Algorithm 1). Intuitively, there is a trade-off on the choice of the value of the lower bound of $\mu\left(a_h \mid x_h\right)$. If $\min_{a_h\in\mathcal{A}}\mu\left(a_h \mid x_h\right) $ is too small, then the NE gap contributed by using the entropy regularization will be prohibitively large and it is required to control the lower bound of $\min_{a_h\in\mathcal{A}}\mu\left(a_h \mid x_h\right)$ in, for example, our Lemma B.3 and Lemma B.12 (in the revised version of our paper). However, if $\min_{a_h\in\mathcal{A}}\mu\left(a_h \mid x_h\right)$ is too large (*i.e.*, $\Pi_{\max}^{k}$ is too "small"), the unique NE policy profile $\xi^{k, \star}=\left(\mu^{k, \star}, \nu^{k, \star}\right)$ in the regularized game $f_k$ might be too far from the NE policy profile $\xi^\star$ of the original game $f$, as noticed by the reviewer. This intuition is formalized in our Lemma B.2 (in our revised version), where the second term in Eq. (11) of Lemma B.2 is directly proportional to $\min_{a_h\in\mathcal{A}}\mu\left(a_h \mid x_h\right)$ and will be $0$ if $\Pi_{\max}^{k}=\Pi_{\max}$ (*i.e.*, $\min_{a_h\in\mathcal{A}}\mu\left(a_h \mid x_h\right)=0$). More specifically, choosing the constrained set $\Pi_{\max}^{k}$ such that $\mu\left(a_h \mid x_h\right)\geq \frac{1}{Ak^c}$ with $c\geq 1$ as some universal constant that is independent of all the parameters $X, Y, A, B, H$ of the problem will only make the convergence rate of our algorithm worse by a multiplicative constant $c$ and can be suppressed in big O notation. Therefore, we eventually choose our constraint set $\Pi_{\max}^{k}$ such that $\mu\left(a_h \mid x_h\right)\geq \frac{1}{Ak}$. Additionally, the factor of $A$ in the denominator of this constraint is to ensure that the constraint set $\Pi_{\max}^{k}$ is well-defined when $k=1$.

---

> ### Author Response · Authors · 2024-11-20
> **Author Response 2**
>
> **Q3. "It is stated that employing the virtual transition helps to ... Can the authors discuss more on this? And why operating OMD in the space of visitation measure is important?"**
>
> Indeed, the constructed virtual transition is utilized in our Algorithm 1 in its sequence-form representation. However, note that the construction of the sequence-form representation of the virtual transition is independent of any policy. Moreover, our algorithm is able to implicitly operate the update of OMD in the space of probability measures over infoset-action pairs precisely because $\left[p\_{1: h}^x \cdot \mu\_{1: h}\right]\left(\cdot, \cdot\right)$, where $\left[p\_{1: h}^x \cdot \mu\_{1: h}\right]\left(x\_h, a\_h\right)=p\_{1: h}^x\left(x\_h\right) \mu\_{1: h}\left(x\_h, a\_h\right)= p\_0^x\left(x\_1\right) \prod\_{h^{\prime}=1}^{h-1} p\_{h^{\prime}}^x\left(x\_{h^{\prime}+1} \mid x\_{h^{\prime}}, a\_{h^{\prime}}\right)\mu(a\_{h^{\prime}}\mid x\_{h^{\prime}})\cdot\mu(a\_{h}\mid x\_{h})$, is a probability measure over infoset-action space $\mathcal{X}\_h\times\mathcal{A}$ for all $h\in[H]$.
>
> Moreover, leveraging the virtual transition weighted negentropy and thus implicitly operating the update of OMD in the space of probability measures over infoset-action pairs is important mainly due to the following reasons. First, in this way, as mentioned in Eq. (9) (in the revised version of our paper), one is able to obtain an upper bound on $\operatorname{NEGap}$ by $\operatorname{NEGap}\left(\xi^k\right) \leqslant \operatorname{NEGap}\left(\xi^{k, \star}\right)+X\left\|p^x\left(\mu^k-\mu^{k, \star}\right)\right\|\_1+Y\left\|p^y\left(\nu^k-\nu^{k, \star}\right)\right\|\_1$, which involves the constructed virtual transition $p^x$ and $p^y$. Note that the upper bound $\left\|p^x\left(\mu^k-\mu^{k, \star}\right)\right\|\_1$ is tighter than $\left\|\mu^k-\mu^{k, \star}\right\|\_1$ because $p\_{1: h}^x\left(x\_h\right)\leq 1$ for all $x\_h\in\mathcal{X}$. More importantly, since $p\_{1: h}^x \cdot \mu\_{1: h}$ specifies a probability measure over infoset-action pairs as mentioned above (and in Section 4.1), this enables us to further bound $\left\|p^x\left(\mu^k-\mu^{k, \star}\right)\right\|\_1$ by $\mathcal{O}\left(\sqrt{\mathrm{KL}\left(p^x \mu^{k, \star}, p^x \mu^k\right)}\right)$ via Pinsker’s inequality (which is not the case for the difference between the sequence-form representation of policies $\left\|\mu^k-\mu^{k, \star}\right\|\_1$). Second, also thanks to the leverage of the virtual transitions,  $\mathrm{KL}\left(p^x \mu^{k, \star}, p^x \mu^k\right)=D\_\psi\left(\mu^{k, \star}, \mu^k\right)$ if $\psi(\mu)=\sum\_{h, x\_h, a\_h} p\_{1: h}^x\left(x\_h\right) \mu\_{1: h}\left(x\_h, a_h\right) \log \left(p\_{1: h}^x\left(x\_h\right) \mu\_{1: h}\left(x\_h, a\_h\right)\right)$ is the virtual transition weighted negentropy regularizer as defined in Section 4.1. Notice that $D\_\psi\left(\mu^{k, \star}, \mu^k\right)$ is an (approximate) contraction mapping guaranteed by our algorithm. Therefore, intuitively, operating the update of OMD in the space of probability measures over infoset-action pairs eventually facilitates the conversion from the convergence to the NE policy profile $\xi^{k,\star}$ to the final convergence of the NE gap, which is our ultimate learning objective.
>
> **Q4. "It seems that the virtual transition is fixed at the beginning (once the game tree is fixed), ... to further improve the convergence rate?"**
>
> Thanks for this comment. Currently, our virtual transition is constructed at the very beginning of the game because in our setting the game tree structure of the IIEFG instance is fixed and known (note that terminology of "bandit feedback" in our current work indicates that the reward functions $r$ and the transition probabilities over the underlying state space instead of the game tree structure are unknown). In cases where the game tree structures are unknown, we believe utilizing the historical information of the experienced infoset-actions to construct an importance-weighted virtual transition estimator is a promising approach, as the reviewer has pointed out. We leave this interesting extension as our future direction.

---

> > ### Comment · Reviewer_xwCS · 2024-11-29
> >
> > I appreciate the authors' detailed replies, which have addressed most of my problems. However, after reading Reviewer xgRk's comments, I do share the same concern regarding the necessity of including virtual transitions in the methodology. While I maintain my original score, I recommend that the authors provide a more comprehensive discussion on this.

---

### Official Review · Reviewer_xgRk · 2024-11-05

**Soundness:** 2
**Presentation:** 3
**Contribution:** 2
**Rating:** 5
**Confidence:** 3

**Summary:**

The paper proposes an algorithm for two-player zero-sum imperfect information extensive-form game with bandit feedback. The algorithm ensures that if both players use it simultaneously, the last iterate of their strategies converge to the Nash equilibrium in a rate of t^{-1/8} (or t^{-1/6} in expectation). The technique is based on negative entropy regularization that has been utilized in normal-form game / Markov game, with the additional technique called virtual transition.

**Strengths:**

- This works gives the first algorithm for 2p0s imperfect information extensive-form game with polynomial-time last-iterate convergence guarantee under bandit feedback.
- This work introduces a virtual transition technique designed to regularize policy updates in extensive-form games, driving last-iterate convergence. This approach adapts the commonly used entropy regularization technique from normal-form games to the extensive-form game setting.

**Weaknesses:**

- The work seems to be a straightforward extension from Cai et al. (2023), which studies last-iterate convergence in normal-form game with bandit feedback. This is not a major concern though, as it speaks more to the simplicity of the setting than to the insufficiency of the solution.
On the other hand, though the virtual transition seems to be the main innovation of this paper, I am not sure whether it is indeed needed (see related questions in the Questions section).

**Questions:**

- Can you explain why 1/X * 1/p^x_{1:h}(x_h) <= 1 (claimed in the last line of Page 23)?
- Does the algorithm still work if replacing the p^x(x_h) in Line 278 simply by 1/X (essentially making p^x a uniform distribution over infoset)?  If this is possible, then Line 278 is essentially not using virtual transition because you can absorb the factor 1/X into \epsilon_k.  If not, can you point out where the analysis might break?
- I do not quite understand the explanation in Line 340-344. Why is bounding the stability of dilated negentropy critically relies on its closed form? This is not the case for general OMD.  Stability term is related to how much the iterate moves between rounds, and the stability term in a "more constrained case" is smaller than that in a "less constrained case". So why not the stability term in \Pi_max smaller than that in the entire space, and existing analysis suffices?

---

> ### Author Response · Authors · 2024-11-20
> **Author Response 1**
>
> We thank the reviewer for the valuable comments and suggestions. Our response to each question is provided in turn below.
>
> **Q1. "though the virtual transition seems to be the main innovation of this paper, I am not sure whether it is indeed needed (see related questions in the Questions section).";"Does the algorithm still work if replacing the p^x(x_h) in Line 278 simply by 1/X ... If not, can you point out where the analysis might break?"**
>
> We now provide further explanations on why we need the proposed virtual transition as well as an illustrative example, the figure of which is shown in Appendix A.1 in the revision of our paper. For ease of discussion, there is only one action $a$ and $H=4$ in this example. Each infoset $x$ in the game tree of this example satisfies $|C(x,a)|=2$ except for infoset $x\_{2,1}$, which is such that $|C(x\_{2,1},a)|=n$ with some $n\geq 2$. Now suppose we use the uniform distribution $p$ as a virtual transition over infoset-action spaces. Then for all the descendants $\set{x\_{4,i}}\_{i=1}^{2n}$ on step $h=4$ of infoset $x\_{2,1}$,  one can see that $p\_{1:H}(x\_{H,i})=\frac{1}{2} \cdot \frac{1}{n} \cdot \frac{1}{2}=\frac{1}{4n}$, while there are only $X=9+3n$ infosets in total. As such, it will happen that $p\_{1:H}(x\_{H,i})< \frac{1}{X}$ when $n> 9$. Actually, one can easily construct an IIEFG instance such that $\min_{x\_H\in\mathcal{X}\_H}p\_{1:H}(x\_{H})\leq \mathcal{O}(\frac{1}{n^m})$ and $X=\mathcal{O}(mn+c)$ with $c$ as a parameter that depends on $m$ but not $n$ for uniform virtual transition $p$. Therefore, when using uniform distribution $p$ as a virtual transition, $\max\_{x\_H\in\mathcal{X}\_H} 1/p\_{1:H}(x\_{H})$ might be prohibitively large and lead to a convergence rate with much worse dependence on $X$ than the virtual transition constructed in our Algorithm 2. This is the reason why we do not simply choose the uniform transition as the virtual transition in our algorithm and analysis.
>
> **Q2. "Can you explain why 1/X * 1/p^x_{1:h}(x_h) <= 1 (claimed in the last line of Page 23)?"**
>
> This question is directly related to the last one. This is due to that our constructed virtual transition $p^x$ satisfies $\min\_{x\_h\in\mathcal{X}\_h,h\in[H]}p^x\_{1:h}(x\_h)\ge \frac{1}{X}$, guaranteed by the following lemma.
>
> **Lemma**. For any $ h\in[H]$ and $x\_h\in\mathcal{X}\_h$, the constructed virtual transition $p^x$ guarantees that $1/p^x\_{1:h}(x_h)\leq X$.
>
> **Proof**. Clearly, $p^x\_{1:h}(\cdot)$ is minimized at $h=H$ for some $x\_H\in\mathcal{X}\_H$ by the definition of the sequence-form representation of virtual transition. By the construction of $p^x\_{1:h}(\cdot)$ in Algorithm 2, one can deduce that $\forall x\_H\in\mathcal{X}\_H$, it holds that (understanding $\set{(x\_h,a\_h)}\_{h\in[H-1]}$ as the unique trajectory leading to $x_H$ below)
> $$
> \begin{align*}
> \notag p^x\_{1:H}(x_H)&=q[x\_H]\\\\
> &=q\left[x\_{H-1}\right] \cdot \frac{c\left[x\_{H}\right]}{\sum\_{x\_{H}^\prime \in C\left(x\_{H-1}, a\_{H-1}\right)} c\left[x\_{H}^\prime \right]}\\\\
> &=q\left[x\_{H-2}\right] \cdot \frac{c\left[x\_{H-1}\right]}{\sum\_{x\_{H-1}^\prime \in C\left(x\_{H-2}, a\_{H-2}\right)} c\left[x\_{H-1}^\prime \right]}\cdot \frac{c\left[x\_{H}\right]}{\sum\_{x\_{H}^\prime \in C\left(x\_{H-1}, a\_{H-1}\right)} c\left[x\_{H}^\prime \right]}\\\\
> &=q\left[x\_{H-2}\right] \cdot \frac{c\left[x\_{H-1}\right]}{\sum\_{x\_{H-1}^\prime \in C\left(x\_{H-2}, a\_{H-2}\right)} c\left[x\_{H-1}^\prime \right]}\cdot \frac{c\left[x\_{H}\right]}{ d\left[x\_{H-1}, a\_{H-1}\right]}\\\\
> &\overset{(i)}{\geq} q\left[x\_{H-2}\right] \cdot \frac{c\left[x\_{H-1}\right]}{\sum\_{x\_{H-1}^\prime \in C\left(x\_{H-2}, a\_{H-2}\right)} c\left[x\_{H-1}^\prime \right]}\cdot \frac{c\left[x\_{H}\right]}{ c[x\_{H-1}]}\\\\
> &= q\left[x\_{H-2}\right] \cdot \frac{c\left[x\_{H}\right]}{\sum\_{x\_{H-1}^\prime \in C\left(x\_{H-2}, a\_{H-2}\right)} c\left[x\_{H-1}^\prime \right]}\\\\
> &\geq\ldots\\\\
> &\geq \frac{c\left[x\_H\right]}{\sum\_{x\_1 \in \mathcal{X}\_1} c\left[x\_1\right]}\\\\
> &\geq \frac{c\left[x\_H\right]}{X\_H}\\\\
> &\geq \frac{c\left[x\_H\right]}{X}\\\\
> &=\frac{1}{X},
> \end{align*}
> $$
> where $c[\cdot]$, $q[\cdot]$, and $d[\cdot,\cdot]$ are defined in our Algorithm 2; and $(i)$ is due to $c[x\_{H-1}]=\max\_{a\in\mathcal{A}} d[x\_{H-1},a]\geq d[x\_{H-1},a\_{H-1}]$.
>
> **Q.E.D.**
>
> This property of our constructed virtual transition $p^x$ serves as a key ingredient in the analysis (*e.g.*, when bounding our Term 4 as noticed by the reviewer and when establishing the final convergence upper bound of the NE gap in the proof of Theorem 5.1 in our Appendix B.4) and we have now incorporated this lemma together with the discussions above into Appendix A of our revised paper for better clarity (highlighted in green).

---

> ### Author Response · Authors · 2024-11-20
> **Author Response 2**
>
> **Q3. "Why is bounding the stability of dilated negentropy critically relies on its closed form? This is not the case for general OMD."**
>
> Indeed, in general, for OMD with Bregman divergence induced by any regularizer $\psi$, one can always bound the stability term as follows (consider the time-invariant learning rate $\eta$ for simplicity) [1]:
>
> $$
> \begin{align*}
> \operatorname{Stability Term}\lesssim&\sum\_{k=1}^t\left\langle \mu^k-\mu^{k+1}, \widehat{\ell}^k\right\rangle-\frac{1}{\eta} \sum\_{k=1}^t D\_{\psi}\left(\mu^{k+1}, \mu^k\right)\quad (\*)\\\\
> \le& \frac{1}{\eta}\sum\_{k=1}^t D\_{\psi}\left(\mu^k, \widetilde{\mu}^{k+1}\right)\\,,\quad (\*\*)
> \end{align*}
> $$
>
> where $\widetilde{\mu}\_{k+1}=\operatorname{argmin}\_{\mu \in\mathbb{R}^{A}\_{\ge 0}} \eta\left\langle \mu, \widehat{\ell}^k\right\rangle+D\_{\psi}\left(\mu, \mu^k\right)$. To further bound the above display, a typical method to consider using Young-Fenchel inequality and Taylor’s theorem to upper bound $(\*)$ and $(\*\*)$ as local norms:
> $$
> \begin{align*}
> (\*)&\le\sum_{k=1}^t \frac{\eta}{2}\|\widehat{\ell}^k\|\_{\nabla^{-2}\psi(z^k)}^2\quad\text{and}\\\\
> (\*\*)&\le\sum_{k=1}^t \frac{\eta}{2}\|\widehat{\ell}^k\|\_{\nabla^{-2}\psi(\widetilde{z}^k)}^2,
> \end{align*}
> $$
> where $z^k=\alpha\mu^{k+1}+(1-\alpha)\mu^k$ and $\widetilde{z}^k=\beta\widetilde{\mu}^{k+1}+(1-\beta)\mu^k$ for some $\alpha,\beta\in(0,1)$. When $\psi$ is chosen as a common regularizer such as negentropy, $\alpha$-Tsallis entropy or log-barrier regularizer, it usually happens that $\widetilde{\mu}^{k+1}(a)\lesssim\mu^k(a)$ and we can obtain an explicit form of the inverse of the Hessian $\nabla^{-2}\psi(\cdot)$ as the Hessian $\nabla^{2}\psi$ is usually a diagonal matrix. Therefore, even when we constrain the optimization of OMD/FTRL onto a smaller subset of the probability simplex and $\mu^{k+1}$ does not admit a closed-form update, we can still bound the stability term by $O(\eta\sum\_{k=1}^t\sum\_{a\in\mathcal{A}}\widehat{\ell}^k(a)^2\mu^k(a)^\alpha)$ ($\alpha=1$ for negentropy, $\alpha=2$ for log-barrier and $\alpha\in(0,1)$ for general $\alpha$-Tsallis entropy).
>
> However, when it comes to the case of using the dilated negentropy regularizer in IIEFGs, the situation becomes more complicated. Recall the dilated negentropy is defined as $\psi(\mu)=\sum\_{h, x\_h, a\_h} \mu\_{1: h}\left(x\_h, a\_h\right) \log \left(\frac{\mu\_{1: h}\left(x\_h, a\_h\right)}{\mu\_{1: h}\left(x\_h\right)}\right)$. For any $\mu\in\mathbb{R}^{XA}\_{> 0}$, $\mu\_{1: h}\left(x\_h\right)$ can be regarded as either $\mu\_{1: h+1}\left(x\_{h+1}\right)=\mu\_{1:h}(x\_h,a\_h)$ (here $(x\_h,a\_h)$ is such that $x\_{h+1}\in C(x\_h,a\_h)$) or $\mu\_{1: h}\left(x\_{h}\right)=\sum\_{a\_h\in\mathcal{A}}\mu\_{1:h}(x\_h,a\_h)$. Note that if these two conditions simultaneously hold, then $\mu\in\Pi\_{\max}$ is a valid sequence-form representation of policy. If we constrain the optimization of OMD onto the subset $\Pi\_{\max}^{k}\subset \Pi\_{\max}$, we can still bound the stability term as $O(\sum\_{k=1}^t \eta\|\widehat{\ell}^k\|\_{\nabla^{-2}\psi(\widetilde{z}^k)}^2)$. However, it is highly unclear (at least to us) how to further bound this local norm for two reasons.
>
> * First, $\eta\widehat{\ell}^k+\nabla \psi(\widetilde{\mu}^{k+1})-\nabla \psi({\mu}^{k})=0$ still holds, but it is unclear whether $\widetilde{\mu}^{k+1}\_{1:h}(x\_h,a\_h)\lesssim\mu^k\_{1:h}(x\_h,a\_h)$ still holds. If we regard $\mu\_{1: h+1}\left(x\_{h+1}\right)=\mu\_{1:h}(x\_h,a\_h)$ in dilated negentropy, the gradient will be such that
> $$
> \frac{\partial\psi(\mu)}{\partial \mu\_{1:h}(x\_h,a\_h)}=\log \frac{\mu\_{1:h} (x\_h,a\_h)}{\mu\_{1:h-1}(x\_{h-1},a\_{h-1})}+1-\sum\_{\left(x\_{h+1}, a\_{h+1}\right) \in C\left(x\_h, a\_h\right) \times \mathcal{A}} \frac{\mu\_{1: h+1}\left(x\_{h+1}, a\_{h+1}\right)}{\mu\_{1: h}\left(x\_h, a\_h\right)}\,.
> $$
> ​If we regard $\mu\_{1: h}\left(x\_{h}\right)=\sum\_{a\_h\in\mathcal{A}}\mu\_{1:h}(x\_h,a\_h)$ in dilated negentropy, the gradient will be such that
> $$
> \frac{\partial\psi(\mu)}{\partial \mu\_{1:h}(x\_h,a\_h)}=\log \frac{\mu\_{1:h} (x\_h,a\_h)}{\sum\_{a\in\mathcal{A}}\mu\_{1:h}(x\_{h},a\_{h})}\,.
> $$
> In either way, different $\mu\_{1:h}(x\_h,a\_h)$'s are coupled with each other, and thus it is unclear whether $\widetilde{\mu}^{k+1}\_{1:h}(x\_h,a\_h)\lesssim\mu^k\_{1:h}(x\_h,a\_h)$ still holds.
>
> * Further, even if it can be shown that $\widetilde{\mu}^{k+1}\_{1:h}(x\_h,a\_h)\lesssim\mu^k\_{1:h}(x\_h,a\_h)$, the Hessian $\nabla^{2}\psi$ is non-diagnoal regardless whether we regard $\mu\_{1: h+1}\left(x\_{h+1}\right)=\mu\_{1:h}(x\_h,a\_h)$ or $\mu\_{1: h}\left(x\_{h}\right)=\sum\_{a\_h\in\mathcal{A}}\mu\_{1:h}(x\_h,a\_h)$. As such, it is unclear how to obtain the explicit form (or at least an upper bound) of the inverse of the Hessian $\nabla^{-2}\psi(\cdot)$. Therefore, it is unclear how to evaluate and upper bound this local norm.

---

> ### Author Response · Authors · 2024-11-20
> **Author Response 3**
>
> **Q3. "Why is bounding the stability of dilated negentropy critically relies on its closed form? This is not the case for general OMD." (Cont.)**
>
> Fortunately, when the feasible set is the full $\Pi_{\max}$, it has been observed that the stability term of OMD with dilated negentropy regularizer can be bounded by
> $$
> \frac{1}{\eta}\sum\_{k=1}^t D\_{\psi}\left(\mu^k, {\mu}^{k+1}\right)\quad(\*\*\*)
> $$
> in contrast to the general bound of $\frac{1}{\eta}\sum_{k=1}^t D\_{\psi}\left(\mu^k, \widetilde{\mu}^{k+1}\right)$ in $(\*\*)$. Notice that this is slightly tighter than $\frac{1}{\eta}\sum\_{k=1}^t D_{\psi}\left(\mu^k, \widetilde{\mu}^{k+1}\right)$ as ${\mu}^{k+1}$ is the Bregman projection of $\widetilde{\mu}^{k+1}$ onto $\Pi\_{\max}$ and $D\_{\psi}\left(\mu^k, {\mu}^{k+1}\right)+D\_{\psi}\left( {\mu}^{k+1},\widetilde{\mu}^{k+1}\right)\le D\_{\psi}\left(\mu^k, \widetilde{\mu}^{k+1}\right)$ (see, *e.g.*, Lemma 8 in [2]). And $(\*\*\*)$ can be eventually upper bounded as $\sum\_{k=1}^t\sum\_{h=1}^H\mu^k\_{1:h}(x^k\_h,a^k\_h)\widehat{\ell}^k\_h(x^k\_h,a^k\_h)^2$ [2]. However, we note that obtaining the upper bound of the stability term in $(\*\*\*)$ relies on the closed-form update of $\mu^{k+1}$ when the feasible set is the full $\Pi\_{\max}$ and we believe this closed-form update of $\mu^{k+1}$ no longer holds in our case where we need to constrain the feasible set to a subset $\Pi\_{\max}^{k}\subset \Pi\_{\max}$. Therefore, it is unclear (at least to us) whether the stability term of OMD with dilated negentropy regularizer can still be upper bounded by $(***)$ in our case.
>
> **Q4. "Stability term is related to ... So why not the stability term in \Pi_max smaller than that in the entire space, and existing analysis suffices?"**
>
> Yes, we fully agree that the stability term of OMD/FTRL actually evaluates how much the iterate moves between rounds. However, as aforementioned, when $\psi$ is the dilated negentropy regularizer, the stability term in our constrained case can indeed be upper bounded by $\frac{1}{\eta}\sum_{k=1}^t D_{\psi}\left(\mu^k, \widetilde{\mu}^{k+1}\right)$, as in the case where the feasible set is the full space of sequence-form representation of policies $\Pi_{\max}$. Nevertheless, this seems difficult (at least for us) to further upper bound it as illustrated above. On the other hand, we believe the closed-form update of $\mu^{k+1}$ in our constrained case no longer holds, thus it is also unclear whether the stability term in our case can be upper bounded by $\frac{1}{\eta}\sum_{k=1}^t D_{\psi}\left(\mu^k, {\mu}^{k+1}\right)$. If there is truly a way to upper bound the stability term in our case by $\frac{1}{\eta}\sum_{k=1}^t D_{\psi}\left(\mu^k, {\mu}^{k+1}\right)$, we believe previous analysis will suffice since $D_{\psi}\left(\mu^k, {\mu}^{k+1}\right)\le D_{\psi}\left(\mu^k, \widehat{\mu}^{k+1}\right)$ (let $\widehat{\mu}^{k+1}=\arg\min_{\mu\in\Pi_{\max}}D_{\psi}(\mu,\widetilde{\mu}^{k+1})$ be the Bregman projection of $\widetilde{\mu}^{k+1}$ onto the full $\Pi_{\max}$) and $D_{\psi}\left(\mu^k, \widehat{\mu}^{k+1}\right)$ can be upper bounded by $\sum_{k=1}^t\sum_{h=1}^H\mu^k_{1:h}(x^k_h,a^k_h)\widehat{\ell}^k_h(x^k_h,a^k_h)^2$ as discussed above. Therefore, as it is unclear whether the stability term in our case can be upper bounded by $\frac{1}{\eta}\sum_{k=1}^t D_{\psi}\left(\mu^k, {\mu}^{k+1}\right)$, it seems not feasible to compare the stability term in our case with that in the case where the feasible set is the full $\Pi_{\max}$ and leverage existing analysis.
>
>
> ----
>
> We would like to thank the reviewer once again for the insightful feedback. We will also incorporate the discussions in our responses to **Q3** and **Q4** into the main body of the future version of our paper for better clarity.
>
> [1] Lattimore et al. Bandit Algorithms. 2020.
>
> [2] Kozuno et al. Model-Free Learning for Two-Player Zero-Sum Partially Observable Markov Games with Perfect Recall. NeurIPS, 21.

---

> ### Comment · Reviewer_xgRk · 2024-11-20
> **Reviewer Response for Q3**
>
> Thanks for the detailed feedback. I may have to take a more careful look at the calculation later.  But in your response for Q3 you mentioned several issues like whether $\tilde{\mu}^{k+1}\leq \mu^k$. Shouldn't these also the case in the analysis for non-clipped policy space (like in Kozuno et al)? They should have dealt with this, right?
>
> But actually, my original suggestion is simpler than that in your response.  As in your response, we have
>
> $\text{Stability} \leq \sum_k \left( \langle \mu^k - \mu^{k+1},\hat{\ell}^k  \rangle - \frac{1}{\eta} D_\psi(\mu^{k+1}, \mu^k)\right) $
>
> $= \sum_k  \max_{\mu\in \Pi_{max}^{k+1}} \left( \langle \mu^k - \mu,\hat{\ell}^k  \rangle - \frac{1}{\eta} D_\psi(\mu, \mu^k)\right)$
> // You can also use "$\leq $" in this line, but it's actually an equality based on Eq.(3) in the paper
>
> $\leq \sum_k  \max_{\mu\in \Pi_{max}} \left( \langle \mu^k - \mu,\hat{\ell}^k  \rangle - \frac{1}{\eta} D_\psi(\mu, \mu^k)\right)$
>
>
> Then in the last line, the max solution for $\mu$ is the "$\mu^{k+1}$ in the non-clipped policy space $\Pi_{max}$".  Then you can use the closed form solution to bound this stability term?

---

> > ### Author Response · Authors · 2024-11-21
> > **Author Response 4**
> >
> > Thank you for your quick response! Our responses to each point are provided below.
> >
> > ----
> >
> > **Q1. "you mentioned several issues like whether $\tilde{\mu}^{k+1} \leq \mu^k$. Shouldn't these also the case in the analysis for non-clipped policy space (like in Kozuno et al)? They should have dealt with this, right?"**
> >
> > We would like to note that even in the case where the feasible set is the full $\Pi\_{\max}$, previous work [1] does not seem to deal with the issues we mentioned in our response to **Q3** as they actually do not consider using $\textcolor{green}{\widetilde{\mu}^{k+1}}=\operatorname{argmin}\_{\mu \in\mathbb{R}^{XA}\_{\ge 0}} \eta\left\langle \mu, \widehat{\ell}^k\right\rangle+D\_{\psi}\left(\mu, \textcolor{blue}{\mu^k}\right)$ to bound the stability term. Instead, they only show that bounding the stability term using $\frac{1}{\eta}\sum\_{k=1}^t D\_{\psi}\left(\textcolor{blue}{\mu^k}, \textcolor{blue}{\mu^{k+1}}\right)$ is possible. Also, as mentioned in our response to **Q3**, it is unclear how to address these issues if following the typical analysis of OMD/FTRL to bound the stability term as local norms.
> >
> > **Q2. "But actually, my original suggestion is simpler than that in your response. As in your response, we have ...  Then you can use the closed form solution to bound this stability term?"**
> >
> > Thanks for this point! However, we would like to note that this idea still seems insufficient to solve our problem, mainly due to the following two reasons. First, please notice that [1] does not directly bound the stability term of $\sum\_k \max \_{\mu \in \Pi\_{\max }}\left(\left\langle\mu^k-\mu, \hat{\ell}^k\right\rangle-\frac{1}{\eta} D\_\psi\left(\mu, \mu^k\right)\right)$ for an algorithm using dilated negentropy as regularizer $\psi$. Instead, they consider bounding the stability term using the closed-form update of $\mu$. Particularly, applying Lemma 8 of [1] in our case shows that, $\forall \mu\in\Pi\_{\max}$,
> >
> > $$
> > \mathrm{D}\_{\psi}\left(\mu, \textcolor{red}{\widehat{\mu}^{k+1}}\right)-\mathrm{D}\_{\psi}\left(\mu,\textcolor{blue}{\mu^k}\right)=\eta\left\langle\mu, \widehat{\ell}^k\right\rangle+\log \hat{Z}\_1^k,\quad (\*)\notag
> > $$
> > where $\textcolor{blue}{\mu^k}$ is the sequence-form representation of policy computed by an algorithm using dilated negentropy on subset $\Pi_{\max}^k$ ,  $\textcolor{red}{\widehat{\mu}^{k+1}}=\operatorname{argmin}\_{\mu \in\Pi\_{\max}} \eta\left\langle \mu, \widehat{\ell}^k\right\rangle+D\_{\psi}\left(\mu, \textcolor{blue}{\mu^k}\right)$ is the sequence-form representation of policy computed by the same algorithm on the full $\Pi\_{\max}$, and $\hat{Z}\_1^k$ is a quantity depends on $\textcolor{blue}{\mu^k}$ and $\widehat{\ell}^k$. Note that this equality holds due to the closed-form update relation between $\textcolor{blue}{\mu^k}$ and $\textcolor{red}{\widehat{\mu}^{k+1}}$. Then the cumulative regret of this algorithm can be upper bounded by
> > $$
> > \begin{align*}
> > \sum\_{k=1}^K\left\langle\textcolor{blue}{\mu^k}-\mu^{\dagger}, \widehat{\ell}^k\right\rangle=&\sum\_{k=1}^K\left\langle\textcolor{blue}{\mu^k}, \widehat{\ell}^k\right\rangle+\frac{1}{\eta}\log \hat{Z}\_1^k-\left(\left\langle\mu^{\dagger}, \widehat{\ell}^k\right\rangle+\frac{1}{\eta}\log \hat{Z}\_1^k\right)\\\\
> > =&\sum\_{k=1}^K\frac{1}{\eta}\left(\mathrm{D}\_{\psi}\left(\textcolor{blue}{\mu^k}, \textcolor{red}{\widehat{\mu}^{k+1}}\right)-\mathrm{D}\_{\psi}\left(\textcolor{blue}{\mu^k},\textcolor{blue}{\mu^k}\right)\right)-\frac{1}{\eta}\left(\mathrm{D}\_{\psi}\left(\mu^{\dagger}, \textcolor{red}{\widehat{\mu}^{k+1}}\right)-\mathrm{D}\_{\psi}\left(\mu^{\dagger},\textcolor{blue}{\mu^k}\right)\right)\\\\
> > =&\frac{1}{\eta}\sum\_{k=1}^K \mathrm{D}\_{\psi}\left(\textcolor{blue}{\mu^k}, \textcolor{red}{\widehat{\mu}^{k+1}}\right)+\frac{1}{\eta}\sum\_{k=1}^K\left(\mathrm{D}\_{\psi}\left(\mu^{\dagger},\textcolor{blue}{\mu^k}\right)-\mathrm{D}\_{\psi}\left(\mu^{\dagger}, \textcolor{red}{\widehat{\mu}^{k+1}}\right)\right)\\\\
> > =&\frac{1}{\eta}\sum\_{k=1}^K \mathrm{D}\_{\psi}\left(\textcolor{blue}{\mu^k}, \textcolor{red}{\widehat{\mu}^{k+1}}\right)+\mathrm{D}\_{\psi}\left(\mu^{\dagger},\textcolor{blue}{\mu^1}\right)-\mathrm{D}\_{\psi}\left(\mu^{\dagger}, \textcolor{red}{\widehat{\mu}^{K+1}}\right)+ \frac{1}{\eta}\sum\_{k=2}^K\left(\mathrm{D}\_{\psi}\left(\mu^{\dagger},\textcolor{blue}{\mu^k}\right)-\mathrm{D}\_{\psi}\left(\mu^{\dagger}, \textcolor{red}{\widehat{\mu}^{k}}\right)\right)\\\\
> > \le&\frac{1}{\eta}\sum\_{k=1}^K \mathrm{D}\_{\psi}\left(\textcolor{blue}{\mu^k}, \textcolor{red}{\widehat{\mu}^{k+1}}\right)+\mathrm{D}\_{\psi}\left(\mu^{\dagger},\textcolor{blue}{\mu^1}\right)+ \frac{1}{\eta}\sum\_{k=2}^K\left(\mathrm{D}\_{\psi}\left(\mu^{\dagger},\textcolor{blue}{\mu^k}\right)-\mathrm{D}\_{\psi}\left(\mu^{\dagger}, \textcolor{red}{\widehat{\mu}^{k}}\right)\right).\quad (\*\*)
> > \end{align*}
> > $$

---

> ### Author Response · Authors · 2024-11-21
> **Author Response 5**
>
> **Q2. "But actually, my original suggestion is simpler than that in your response. As in your response, we have ...  Then you can use the closed form solution to bound this stability term?" (Cont.)**
>
> Nevertheless, it is difficult (at least for us) to further upper bound the above display mainly due to the following two reasons:
>
> * Compared with the bound in the proof of Lemma 7 in [1], there is an additional term $\frac{1}{\eta}\sum\_{k=2}^K\left(\mathrm{D}\_{\psi}\left(\mu^{\dagger},\textcolor{blue}{\mu^k}\right)-\mathrm{D}\_{\psi}\left(\mu^{\dagger}, \textcolor{red}{\widehat{\mu}^{k}}\right)\right)$, coming from the fact that $\textcolor{blue}{\mu^k}$ and $\textcolor{red}{\widehat{\mu}^{k}}$ are now not equal with each other in our case. In general, it is unclear whether this term can be further upper bounded.
> * Further, it is also unclear how to bound the first term $\frac{1}{\eta}\sum\_{k=1}^K \mathrm{D}\_{\psi}\left(\textcolor{blue}{\mu^k}, \textcolor{red}{\widehat{\mu}^{k+1}}\right)$ in the last line of $(\*\*)$ since $\textcolor{blue}{\mu^k}$ now no longer admits a closed-form update (note that [1] only shows that the stability term $\frac{1}{\eta}\sum\_{k=1}^K \mathrm{D}\_{\psi}\left(\textcolor{blue}{\mu^k}, \textcolor{red}{\widehat{\mu}^{k+1}}\right)$ can be well upper bounded by $\sum_{k=1}^K\sum\_{h=1}^H\textcolor{blue}{\mu^k\_{1:h}}(x^k\_h,a^k\_h)\widehat{\ell}^k\_h(x^k\_h,a^k\_h)^2$ when $\textcolor{blue}{\mu^k}$ is updated on the full $\Pi\_{\max}$ as discussed in our response to **Q3**). More importantly, to achieve $(\*)$, the analysis of [1] also relies on the closed-form update of $\textcolor{blue}{\mu^k}$, in addition to the closed-form update relation between $\textcolor{blue}{\mu^k}$ and $\textcolor{red}{\widehat{\mu}^{k+1}}$. This means that $\hat{Z}\_1^k$ in our case should be different from the original ${Z}\_1^k$ defined in [1], which depends on the closed-form update of $\textcolor{blue}{\mu^k}$. Currently, it is unclear to us whether such a $\hat{Z}\_1^k$ exists, and if it does, what its form should be. If such $\hat{Z}\_1^k$ does not exist, it even seems impossible to obtain the upper bound in $(\*\*)$.
>
> As such, following the method in [1] to bound the stability term of the cumulative regret in our case seems not applicable. Moreover, please note that our aim is to establish an algorithm for learning IIEFGs with provable last-iterate convergence guarantees, the case of which is relatively more complicated than simply achieving the sublinear regret to guarantee the average-iterate convergence. Therefore, it seems more difficult (at least for us) to utilize such a regularizer and leverage the analysis in [1] in our problem. Besides, even if it is possible to use an algorithm with a dilated negentropy to achieve the last-iterate convergence guarantee, based on the above discussions,  we believe the analysis of our algorithm will be neater and cleaner than that of such an algorithm, due to the use of our virtual transition weighted negentropy. In this sense, we believe our algorithm, especially the design of our virtual transition, is valuable and contributes positively to the understanding of obtaining the last-iterate convergence guarantee in IIEFGs.
>
> We hope the above explanations would help to address the concerns of the reviewer. We are also more than happy to discuss any other questions with the reviewer.
>
> ----
>
> [1] Kozuno et al. Model-Free Learning for Two-Player Zero-Sum Partially Observable Markov Games with Perfect Recall. NeurIPS, 21.

---

> ### Comment · Reviewer_xgRk · 2024-11-22
> **Reviewer Response**
>
> First, the analysis in [1] should be able to bound $D_\psi(\mu^k, \hat{\mu}^{k+1})$ in your case. If you read their Lemma 7 and the part in Lemma 6 that bounds $D_\psi(\mu^k, \hat{\mu}^{k+1})$ carefully, you can find that they hold for **any** $\mu^k\in\Pi_{\max}$.  The only requirement is that $\hat{\mu}^{k+1}$ is one-step update from $\mu^k$, that is, $\hat{\mu}^{k+1}=\text{arg} \min_{\mu\in\Pi_{\max}} \left(\eta \langle \mu,\hat{\ell}^k  \rangle + D_\psi(\mu, \mu^k)\right)$.
>
> Below, I continue my previous calculation to show that the stability term is upper bounded by $\frac{1}{\eta}D_\psi(\mu^k ,\hat{\mu}^{k+1})$.   The calculation below are essentially the same as in [1] for the $\Pi_{\max}$ space.
>
> $\text{Stability} \leq \sum_k  \max_{\mu\in \Pi_{max}} \left( \langle \mu^k - \mu,\hat{\ell}^k  \rangle - \frac{1}{\eta} D_\psi(\mu, \mu^k)\right)$
>
> $= \sum_k  \left( \langle \mu^k - \hat{\mu}^{k+1},\hat{\ell}^k  \rangle - \frac{1}{\eta} D_\psi(\hat{\mu}^{k+1}, \mu^k)\right)$.
>
> Next, we show that the right-hand side above is equal to $\sum_k  \frac{1}{\eta} D_\psi(\mu^k, \hat{\mu}^{k+1})$.
> This is shown below:
>
> $D_\psi(\mu^k ,\hat{\mu}^{k+1}) + D_\psi(\hat{\mu}^{k+1}, \mu^k)$
>
> $=\sum_{h}\sum_{x_h,a_h} (\mu_{1:h}^k(x_h,a_h) - \hat{\mu}^{k+1}_{1:h}(x_h, a_h)) \log \frac{\mu_h^{k}(a_h | x_h) } {\hat{\mu}^{k+1}_h(a_h|x_h) }$
>
> $=\sum_{h}\sum_{x_h,a_h} (\mu_{1:h}^k(x_h,a_h) - \hat{\mu}^{k+1}_{1:h}(x_h, a_h))$  $\left( \mathbf{1} _{(x_h, a_h)=(x_h^k, a_h^k)}\left(\eta \hat{\ell}^k(x_h,a_h) - \log Z^k _{h+1}\right)+  \mathbf{1} _{x_h=x_h^k}\left(\log Z^k_h\right) \right)\ \ \ $   // By Eq.(7) in [1]
>
> $=\eta \langle \mu^k - \hat{\mu}^{k+1}, \hat{\ell}^k \rangle - \sum_{h=1}^H  \left(\mu_{1:h}^k(x_h^k,a_h^k) - \hat{\mu}^{k+1}_{1:h}(x_h^k, a_h^k)\right)\log Z^k _{h+1}$  $+ \sum _{h=1}^H  \left(\mu _{1:h}^k(x_h^k) - \hat{\mu}^{k+1} _{1:h}(x _h^k)\right)\log Z^k _{h}$
>
> $=\eta  \langle \mu^k - \hat{\mu}^{k+1}, \hat{\ell}^k \rangle - \sum_{h=2}^H  \left(\mu_{1:h}^k(x_h^k) - \hat{\mu}^{k+1}_{1:h}(x_h^k)\right)\log Z^k _{h} + \sum _{h=1}^H  \left(\mu _{1:h}^k(x_h^k) - \hat{\mu}^{k+1} _{1:h}(x _h^k)\right)\log Z^k _{h}$
>
> $= \eta \langle \mu^k - \hat{\mu}^{k+1}, \hat{\ell}^k \rangle + \left(\mu _{1}^k(x_1^k) - \hat{\mu}^{k+1} _{1}(x _1^k)\right)\log Z^k _{1}$
>
> $= \eta \langle \mu^k - \hat{\mu}^{k+1}, \hat{\ell}^k \rangle\ \ \ \ \ $   // because $\mu _{1}^k(x_1^k) =  \hat{\mu}^{k+1} _{1}(x _1^k) = 1$

---

> > ### Author Response · Authors · 2024-11-22
> > **Author Response 6**
> >
> > We sincerely thank the reviewer for their response, and we agree with the points raised. Upon careful reflection, we recognize that the statement regarding $Z^k_1$ depending on the closed-form update of $\mu^k$ in our previous response, as well as the statement that the prior analysis cannot be used to bound the stability term in our case, are inaccurate. We will make sure to clarify and correct these points in the main body of the paper in its future version.
> >
> > That said, we would like to note that, while it turns out that the stability term of OMD with a dilated negentropy regularizer in the constrained case can indeed be bounded, there might remain other technical challenges that need to be addressed to establish the last-iterate convergence guarantee for learning in IIEFGs. For instance, in Eq. (9) of our paper, the NE gap is currently bounded using $\left\\|p^x\left(\mu^k-\mu^{k, \star}\right)\right\\|_1$ (and $\left\\|p^y\left(\nu^k-\nu^{k, \star}\right)\right\\|_1$), given our use of a virtual transition weighted negentropy regularizer. With the leverage of the virtual transition, this term represents the difference between probability measures over infoset action space and can be further bounded using Pinsker’s inequality. However, with the dilated negentropy regularizer, even if the NE gap could be expressed in terms of $\left\\|\mu^k-\mu^{k, \star}\right\\|_1$ (and $\left\\|\nu^k-\nu^{k, \star}\right\\|_1$), it appears infeasible to bound these terms via Pinsker’s inequality, as sequence-form representations of policies are generally not probability measures over infoset-action space, except in the special case where the IIEFG reduces to a matrix game.
> >
> > Furthermore, even if last-iterate convergence for learning in IIEFGs can eventually be achieved using a dilated negentropy regularizer, we believe that our algorithm, particularly the design of the virtual transition, offers valuable insights since it provides an alternative perspective for understanding how last-iterate convergence guarantees in IIEFGs can be obtained. We believe this is especially meaningful considering the fact that there are no existing results established for our problem to our knowledge.
> >
> > We deeply appreciate the reviewer’s thoughtful comments, and we are also more than willing to answer any further questions.

---

> > > ### Comment · Reviewer_xgRk · 2024-11-22
> > > **Reviewer Response**
> > >
> > > Still, it looks like the inclusion of virtual transition is redundant and only introduces unnecessary complication.  So let's go back to Q1.  Maybe I was not clear in my original review, but I actually mean "what if you replace all $p_{1:h}^x(x_h)$ simply by 1/X" --- I don't mean replacing decendants distribution to uniform, as indicated in your response. So the issue $p_{1:h}^x(x_h)\leq \frac{1}{n^m}$ you pointed out won't happen.  Notice that your regularizer or your loss estimator construction only depends on $p_{1:h}^x(x)$ for any $x$.  Again, please point out where the proof might break.

---

> > > > ### Author Response · Authors · 2024-11-23
> > > > **Author Response 7**
> > > >
> > > > We thank the reviewer for clarifying this point in the initial review. If we were to substitute $p\_{1: h}^x\left(x\_h\right)$ with $1/X$, at least one part of the analysis would become more complicated, and another part might fail to hold.
> > > > * Upon substituting $p\_{1: h}^x\left(x\_h\right)$ with $1/X$, the Bregman divergence induced by $\psi\_{\frac{1}{X}}(\mu)=\sum_{h, x\_h, a\_h} \frac{1}{X} \mu\_{1: h}\left(x\_h, a\_h\right) \log \left(\frac{1}{X} \mu\_{1: h}\left(x\_h, a\_h\right)\right)$ becomes
> > > > $$
> > > > D\_{\psi\_{\frac{1}{X}}}(\mu^1,\mu^2)=\frac{1}{X}\sum\_{h, x\_h, a\_h}\mu\_{1: h}^1\left(x\_h, a\_h\right)\log \frac{\mu\_{1: h}^1\left(x\_h, a\_h\right)}{\mu\_{1: h}^2\left(x\_h, a\_h\right)}-\frac{1}{X}\sum\_{h, x\_h, a\_h}\left(\mu\_{1: h}^1\left(x\_h, a\_h\right)-\mu\_{1: h}^2\left(x\_h, a\_h\right)\right)\notag.
> > > > $$
> > > > ​In this expression, the second term of the RHS may be non-zero even for two valid sequence-form representations $\mu^1$, $\mu^2\in\Pi_{\max}$. In contrast, when using a virtual transition $p^x$ instead of a constant $\frac{1}{X}$, the second term will become $0$. Accounting for this additional term would result in an additional Term 7 in the analysis when establishing the (approximate) contraction mapping property for $D_{\psi_{\frac{1}{X}}}(\cdot,\cdot)$, specifically related to $\varepsilon_k\cdot\frac{1}{X}(\mu^{k,\star}-\mu^k)$. While it seems possible to upper bound this term within the contraction mapping analysis, we are uncertain whether it would enlarge the final dependence on $X$ and $A$. And we believe this will at least make the analysis more complicated than ours.
> > > >
> > > > * More importantly, using $\psi_{\frac{1}{X}}(\cdot)$, it becomes unclear how to derive a final convergence upper bound for the NE gap. In our current analysis, the NE gap is first bounded as
> > > > $$
> > > > \operatorname{NEGap}\left(\xi^k\right) \leqslant \operatorname{NEGap}\left(\xi^{k, \star}\right)+X\left\\|p^x\left(\mu^k-\mu^{k, \star}\right)\right\\|_1+Y\left\\|p^y\left(\nu^k-\nu^{k, \star}\right)\right\\|_1\notag.
> > > > $$
> > > > ​The second term on the RHS is then bounded via Pinsker’s inequality as $\mathcal{O}\left(\sqrt{\mathrm{KL}\left(p^x \mu^{k, \star}, p^x \mu^k\right)}\right)$, since $p^x\left(\mu^k-\mu^{k, \star}\right)$ represents the difference between probability measures over the infoset-action space.
> > > > Also, with the observation that
> > > > $\operatorname{KL}\left(p^x \mu^{k, \star}, p^x \mu^k\right)=D\_{\psi\_{p^x}}\left(\mu^{k, \star}, \mu^k\right)$,
> > > > this analysis converts the convergence of NE policy profile into the convergence of NE gap. However, if $\psi\_{\frac{1}{X}}(\cdot)$ is used, we can only bound the NE gap as
> > > > $$
> > > > \operatorname{NEGap}\left(\xi^k\right) \leqslant \operatorname{NEGap}\left(\xi^{k, \star}\right)+X\left\\|\frac{1}{X}\left(\mu^k-\mu^{k, \star}\right)\right\\|\_1+Y\left\\|\frac{1}{Y}\left(\nu^k-\nu^{k, \star}\right)\right\\|\_1\notag.
> > > > $$
> > > > ​It is unclear how to further bound this expression using Pinsker’s inequality, as sequence-form representations of policies are generally not probability measures over the infoset-action space. Since our constructed virtual transition $p^x$ satisfies $p^x\_{1:h}(x_h)\ge \frac{1}{X}$, one might attempt to bound the second term on the RHS as
> > > > $$
> > > > X\left\\|\frac{1}{X}\left(\mu^k-\mu^{k, \star}\right)\right\\|\_1\le X\left\\|p^x\left(\mu^k-\mu^{k, \star}\right)\right\\|\_1=\mathcal{O}\left(X\sqrt{\mathrm{KL}\left(p^x \mu^{k, \star}, p^x \mu^k\right)}\right)=\mathcal{O}\left(X\sqrt{D\_{\psi\_{p^x}}\left(\mu^{k, \star}, \mu^k\right)}\right)\notag.
> > > > $$
> > > > ​However, this approach is problematic, as the algorithm uses $D\_{\psi\_{\frac{1}{X}}}\left(\mu^{k, \star}, \mu^k\right)$ and thus can only guarantee an upper bound on $D\_{\psi\_{\frac{1}{X}}}\left(\mu^{k, \star}, \mu^k\right)$, and $D\_{\psi\_{p^x}}\left(\mu^{k, \star}, \mu^k\right)$ is generally incomparable with $D\_{\psi\_{\frac{1}{X}}}\left(\mu^{k, \star}, \mu^k\right)$. Even if the second term $\frac{1}{X}\sum\_{h, x\_h, a\_h}\left(\mu\_{1: h}^1\left(x\_h, a\_h\right)-\mu\_{1: h}^2\left(x\_h, a\_h\right)\right)$ in $D\_{\psi\_{\frac{1}{X}}}(\cdot,\cdot)$ were zero, $D\_{\psi\_{\frac{1}{X}}}(\cdot,\cdot)$ would remain smaller than $D\_{\psi\_{p^x}}\left(\cdot, \cdot\right)$, as $p^x\_{1:h}(x\_h)\ge \frac{1}{X}$.
> > > >
> > > > Given these challenges, we believe substituting $p_{1: h}^x\left(x_h\right)$ with $1/X$ would not simplify the analysis. On the contrary, it would likely complicate the analysis and even render it infeasible. Besides, algorithmically, we would humbly note that the computation of our virtual transition $p^x$ is also not complicated and can be done in $\mathcal{O}(wXA)$ time with $w=\max_{h\in[H],(x_h,a_h)\in\mathcal{X}_h\times\mathcal{A}}|C(x_h,a_h)|$ as shown in our Algorithm 2. Therefore, we believe the design of our proposed virtual transition is neither redundant nor unnecessarily complex.
> > > >
> > > > We thank the reviewer again for the time and we hope the above explanations will help address the concerns of the reviewer.

---

> > > > > ### Comment · Reviewer_xgRk · 2024-11-23
> > > > > **Reviewer Response**
> > > > >
> > > > > I have to first understand why $[p^x\mu]$ is a distribution over $X\times A$.  Where did you prove $\sum_h \sum_{x_h, a_h} p^x _{1:h}(x_h) \mu _{1:h}(x_h, a_h)=1$?

---

> > > > > > ### Author Response · Authors · 2024-11-23
> > > > > > **Author Response 8**
> > > > > >
> > > > > > Thanks for your quick response! Actually, this holds for any valid sequence-form representation of virtual transitions and similar results can be found in existing works that also leverage similar ideas of the virtual transition weighted regularizer (say, [1,2]). We provide a formal proof here for convenience. Formally, we prove that the virtual transition $p^x$ computed by our Algorithm 2 satisfies $\[p^x\_{1:h}\mu\_{1:h}\](\cdot,\cdot)$ is a probability measure over the infoset-action space on step $h$ for all $h\in[H]$ and $\mu\in\Pi\_{\max}$.
> > > > > >
> > > > > > **Lemma**. For any $h\in[H]$ and $\mu\in\Pi\_{\max}$, $p^x$ computed by our Algorithm 2 satisfies $\sum\_{(x\_h,a\_h)\in\mathcal{X}\_h\times\mathcal{A}}\[p^x\_{1:h}\mu\_{1:h}\](x\_h,a\_h)=1$.
> > > > > >
> > > > > > **Proof**. Fix $h\in[H]$ in what follows. We first prove that $p^x\_h(\cdot\mid x\_h,a\_h )$ computed by our Algorithm 2 is indeed a valid transition probability over $C(x\_h,a\_h )$. To see this,  since $p^x\_h(x\_{h+1}\mid x\_h,a\_h )=\frac{c[x\_{h+1}]}{d\left[x\_h, a\_h\right]}=\frac{c[x\_{h+1}]}{\sum\_{x_{h+1}^\prime \in C\left(x\_h, a\_h\right)} c\left[x\_{h+1}^\prime\right]}$ ($c[\cdot]$ and $d[\cdot,\cdot]$ are defined in Algorithm 2), we have
> > > > > > $$
> > > > > > \sum\_{x_{h+1} \in C\left(x\_h, a\_h\right)}p^x\_h(x\_{h+1}\mid x\_h,a\_h )=\sum\_{x\_{h+1} \in C\left(x\_h, a\_h\right)}\frac{c[x\_{h+1}]}{\sum\_{x\_{h+1}^\prime \in C\left(x\_h, a\_h\right)} c\left[x\_{h+1}^\prime\right]}=1\notag.
> > > > > > $$
> > > > > > Now suppose there is a layered Markov decision process (MDP) with $H$ steps, "state space" as $\{\mathcal{X}\_h\}\_{h\in[H]}$, action space as $\mathcal{A}$,  and "transition dynamics" in step $h$ determined by $p^x\_h(\cdot|\cdot,\cdot)$. For any valid (Markov) policy $\mu$, we denote by $\mathbb{P}(x\_h,a\_h)$ the probability that the "state-action" pair $(x\_h,a\_h)$ will be visited under policy $\mu$ in this MDP. Specifically, for any $(x\_h,a\_h)\in\mathcal{X}\_h\times\mathcal{A}$, one can deduce that (understanding $\set{(x\_i,a\_i)}\_{i\in[h]}$ as the unique trajectory leading to $x\_h$ below)
> > > > > > $$
> > > > > > \begin{align*}
> > > > > > \mathbb{P}(x\_h,a\_h)=&\mathbb{P}(x\_1,a\_1,\ldots,x\_h,a\_h)\\\\
> > > > > > =&\mathbb{P}(x\_1,a\_1,\ldots,x\_h)\mathbb{P}(a\_h\mid x\_1,a\_1,\ldots,x\_h)\\\\
> > > > > > =&\mathbb{P}(x\_1,a\_1,\ldots,x\_h)\mathbb{P}(a\_h\mid x\_h)\\\\
> > > > > > =&\mathbb{P}(x\_1,a\_1,\ldots,x\_h)\mu(a\_h\mid x\_h)\\\\
> > > > > > =&\mathbb{P}(x\_1,a\_1,\ldots,x\_{h-1},a\_{h-1})\mathbb{P}(x\_h\mid x\_1,a\_1,\ldots,x\_{h-1},a\_{h-1})\mu(a\_h\mid x\_h)\\\\
> > > > > > =&\mathbb{P}(x\_1,a\_1,\ldots,x\_{h-1},a\_{h-1})\mathbb{P}(x\_h\mid x\_{h-1},a\_{h-1})\mu(a\_h\mid x\_h)\\\\
> > > > > > =&\mathbb{P}(x\_1,a\_1,\ldots,x\_{h-1},a\_{h-1})p^x\_{h}(x\_h\mid x\_{h-1},a\_{h-1})\mu(a\_h\mid x\_h)\\\\
> > > > > > =&\ldots\\\\
> > > > > > =&p^x\_1(x\_1)\prod\_{i=2}^{h}p^x\_{i}(x\_i\mid x\_{i-1},a\_{i-1})\prod\_{i=1}^{h}\mu(a\_i\mid x\_i)\\\\
> > > > > > =&p^x\_{1:h}(x\_h)\mu\_{1:h}(x\_h,a\_h),
> > > > > > \end{align*}
> > > > > > $$
> > > > > >
> > > > > > where the last equality follows from the definition of the sequence-form representation of virtual transitions and policies. Notice that there must be one and only one $(x\_h^\prime,a\_h^\prime)\in\mathcal{X}\_h\times\mathcal{A}$ that is visited at step $h$. Thus it holds that
> > > > > > $$
> > > > > > \sum\_{(x\_h,a\_h)\in\mathcal{X}\_h\times\mathcal{A}}\mathbb{P}(x\_h,a\_h)=\sum\_{(x\_h,a\_h)\in\mathcal{X}\_h\times\mathcal{A}}p^x\_{1:h}(x\_h)\mu\_{1:h}(x\_h,a\_h)=1.
> > > > > > $$
> > > > > > **Q.E.D.**
> > > > > >
> > > > > > ----
> > > > > >
> > > > > > [1] Bai et al. Near-Optimal Learning of Extensive-Form Games with Imperfect Information. ICML, 22.
> > > > > >
> > > > > > [2] Fiegel et al. Adapting to game trees in zero-sum imperfect information games. ICML, 23.

---

> > > > > > > ### Comment · Reviewer_xgRk · 2024-11-23
> > > > > > > **Reviewer Response**
> > > > > > >
> > > > > > > Still, I don't see the necessity of virtual transition and how it simplifies the analysis. Instead, it seems to complicate both the analysis and the algorithm.   I think it's not ideal that the element whose importance is emphasized a lot in the paper can be completely removed.
> > > > > > >
> > > > > > > Since for any $h$,
> > > > > > >
> > > > > > > $\sum_{x_h, a_h} \mu_{1:h}(x_h,a_h)$
> > > > > > >
> > > > > > > $= \sum_{x_1, a_1, x_2, a_2, \ldots, x_h, a_h} \prod_{i=1}^h \mu(a_i|x_i) $
> > > > > > >
> > > > > > > $= \sum_{x_1, x_2, \ldots, x_h} 1$
> > > > > > >
> > > > > > > $= X_h\ \ \ \ $   // $X_h$ is the number of infosets on layer $h$
> > > > > > >
> > > > > > >
> > > > > > >
> > > > > > > Following arguments similar to yours, to bound $|| \mu^k - \mu^{k,\star} ||_1$ by $D _\psi(\mu^k, \mu^{k,\star})$, one only needs to split $|| \mu^k - \mu^{k,\star} ||_1$ into layers, scaling layer $h$ by $\frac{1}{X_h}$ (to make it a distribution), and applying Pinsker's inequality on each layer.

---

> > > > > > > > ### Author Response · Authors · 2024-11-24
> > > > > > > > **Author Response 9**
> > > > > > > >
> > > > > > > > Thanks for your response. However, we believe that your derivation seems problematic from Line 3 and does not hold for a valid $\mu\in\Pi\_{\max}$. We just need to think about a simple example. Consider an IIEFG instance with $A=2$ and $H=2$. On step $h=1$, there is only one infoset $x\_1$ such that $C(x\_1,a\_1)=\set{x\_{2,1},x\_{2,2}}$ and $C(x\_1,a\_2)=\set{x\_{2,3}}$. Then for a valid policy $\mu$ such that $\mu(a\_1\mid x)=1$ while $\mu(a\_2\mid x)=0$ for any $x\in\mathcal{X}$, it is clear that
> > > > > > > > $$
> > > > > > > > \sum\_{(x,a)\in\mathcal{X}\_2\times\mathcal{A}}\mu\_{1:2}(x,a)=\mu\_{1:2}(x\_{2,1},a\_1)+\mu\_{1:2}(x\_{2,2},a\_1)=2\neq X\_2.
> > > > > > > > $$
> > > > > > > > Similarly, for a uniform policy $\mu^\prime$ such that $\mu^\prime(a\_1\mid x)=\mu^\prime(a\_2\mid x)=1/2$ for any $x\in\mathcal{X}$, it is also clear that
> > > > > > > > $$
> > > > > > > > \sum\_{(x,a)\in\mathcal{X}\_2\times\mathcal{A}}\mu^\prime\_{1:2}(x,a)=\frac{1}{4}\cdot 6=\frac{3}{2}\neq X\_2.
> > > > > > > > $$
> > > > > > > > We hope the above clarifications will help to address the concerns of the reviewer.

---

> ### Comment · Reviewer_xgRk · 2024-11-24
> **Reviewer Response**
>
> Yes, my previous calculation has some mistake. Sorry about that.
>
> Yet, the reason to include virtual transition remains uncompelling, especially when the reasons are all about low-level difficulties in the proofs instead of high-level algorithmic necessity.
>
> The inclusion of virtual transition makes the algorithm heavily depends on the learner's prior knowledge on the set of infosets. In contrast, the version by Kozuno et al. only makes update on the infosets the learner visits.
>
> I also point out the "Pinksker's inequality" for unnormalized measures: Theorem 5 in  https://www.cs.cmu.edu/~yaoliang/mynotes/sc.pdf, which allows you to connect 1-norm and $D_\psi$, but I cannot spend time on this anymore.

---

> > ### Author Response · Authors · 2024-11-25
> > **Author Response 10**
> >
> > Thank you for raising your score for our work!
> >
> > Currently, the construction of our proposed virtual transition indeed needs knowledge of the game tree structure, whereas the approach in [1] does not. However, we also would like to note that knowing the game tree structure to construct the virtual transition is also required by the Balanced OMD algorithm of [2] and the Balanced FTRL algorithm of [3] to achieve shaper dependence on $X$ (and $H$). Moreover, as noted by [2], the requirement of knowledge of the game tree structure is relatively mild, as the structure can be extracted by just one tree traversal. Additionally, it is also possible to construct an empirical virtual transition to serve as a surrogate using the importance-weighted virtual transition estimator for infoset-action pairs, as shown by the Adaptive FTRL algorithm in [3]. Extending our algorithm to the case with unknown game tree structures seems an interesting direction, but it is currently out of the scope of the focus of our current work and we leave this as our future work.
> >
> > Regarding the point noted by the reviewer, we agree with the reviewer that this is the other potential approach to solve our problem. Nevertheless, this seems to require computing a policy $\mu^h=\arg\max\_{\mu\in\Pi\_{\max}}\sum\_{(x\_h,a\_h)\in\mathcal{X}\_h\times\mathcal{A}} \mu\_{1:h}(x\_h,a\_h)$ and use the value $\sum\_{(x\_h,a\_h)\in\mathcal{X}\_h\times\mathcal{A}} \mu\_{1:h}^h(x\_h,a\_h)$ of this $\mu^h$ as a normalizing factor in regularizer, for each $h\in[H]$. Computing such $\mu^h$ seems to also require knowledge of the game tree structure and it is currently not very clear (at least to us) whether and how such $\mu^h$ can be efficiently computed on general IIEFG instances, in contrast to our proposed virtual transition. Overall, we believe this approach might be applicable to our problem, but it appears that it may not result in a simpler solution than ours. We leave the investigation in this direction also as our future works.
> >
> > Again, we would like to express our heartfelt gratitude to you for your time and your thoughtful comments.
> >
> > ----
> >
> > [1] Kozuno et al. Model-Free Learning for Two-Player Zero-Sum Partially Observable Markov Games with Perfect Recall. NeurIPS, 21.
> >
> > [2] Bai et al. Near-Optimal Learning of Extensive-Form Games with Imperfect Information. ICML, 22.
> >
> > [3] Fiegel et al. Adapting to game trees in zero-sum imperfect information games. ICML, 23.

---

### Meta-Review · Area_Chair_MrZD · 2024-12-18

**Metareview:**

This paper examines the problem of learning approximate Nash equilibria in two-player zero-sum imperfect information extensive-form games (IIEFGs). The authors model IIEFGs as partially observable Markov games (POMGs) with perfect recall and bandit feedback, i.e., the game’s structure is a priori unknown, and only the rewards of experienced information set-action pairs are observed in each episode. They propose an algorithm that employs a negentropy regularizer weighted by a virtual transition over the information set-action space. By carefully designing this virtual transition and leveraging entropy regularization techniques, the authors prove that the proposed algorithm converges to Nash equilibrium with a finite-time (last-iterate) convergence rate of $\widetilde{O}(k^{-\frac{1}{8}})$ with high probability under bandit feedback.

This paper was discussed extensively between authors and reviewers, and Reviewer xgRk in particular went into great depth discussing both technical and conceptual details. One of the main concerns raised was that the paper seems, in certain regards, incremental over previous work by Cai et al., and the reason to include the virtual transitions is somewhat uncompelling - especially because it seems motivated by low-level proof details as opposed to conceptual reasons. This reliance on virtual transitions means that the algorithm requires full prior knowledge of the game's infosets, in contrast to prior work by, e.g., Kozuno et al., which only requires updates on the infosets that a player actually visits during play. Reviewer USyT also pointed out several computational complexity issues with the authors' approach which were not adequately addressed by the authors during the discussion phase.

In the end, despite some partial improvement in the reviewers' assessment of this work, it was not possible to make a clear case that the paper meets the (admittedly high) acceptance criteria of ICLR, so a decision was reached to make a reject recommendation to the program chairs.

**Additional Comments On Reviewer Discussion:**

To conform to ICLR policy, I am repeating here the relevant part of the metareview considering the reviewer discussion.

> This paper was discussed extensively between authors and reviewers, and Reviewer xgRk in particular went into great depth discussing both technical and conceptual details. One of the main concerns raised was that the paper seems, in certain regards, incremental over previous work by Cai et al., and the reason to include the virtual transitions is somewhat uncompelling - especially because it seems motivated by low-level proof details as opposed to conceptual reasons. This reliance on virtual transitions means that the algorithm requires full prior knowledge of the game's infosets, in contrast to prior work by, e.g., Kozuno et al., which only requires updates on the infosets that a player actually visits during play. Reviewer USyT also pointed out several computational complexity issues with the authors' approach which were not adequately addressed by the authors during the discussion phase. In the end, despite some partial improvement in the reviewers' assessment of this work, it was not possible to make a clear case that the paper meets the (admittedly high) acceptance criteria of ICLR.

---

### Decision · Program_Chairs · 2025-01-22

Reject